

# The fraternal birth-order effect as a statistical artefact: convergent evidence from probability calculus, simulated data, and multiverse meta-analysis

Johannes K. Vilsmeier, Michael Kossmeier, Martin Voracek and Ulrich S. Tran

Department of Cognition, Emotion, and Methods in Psychology, Faculty of Psychology, University of Vienna, Vienna, Austria

Corresponding author
Ulrich S. Tran,
ulrich.tran@univie.ac.at

## ABSTRACT

The fraternal-birth order effect (FBOE) is a research claim which states that each older brother increases the odds of homosexual orientation in men *via* an immunoreactivity process known as the maternal immune hypothesis. Importantly, older sisters supposedly either do not affect these odds, or affect them to a lesser extent. Consequently, the fraternal birth-order effect predicts that the association between the number of older brothers and homosexual orientation in men is greater in magnitude than any association between the number of older sisters and homosexual orientation. This difference in magnitude represents the main theoretical estimand of the FBOE. In addition, no comparable effects should be observable among homosexual *vs* heterosexual women. Here, we triangulate the empirical foundations of the FBOE from three distinct, informative perspectives, complementing each other: first, drawing on basic probability calculus, we deduce mathematically that the body of statistical evidence used to make inferences about the main theoretical estimand of the FBOE rests on incorrect statistical reasoning. In particular, we show that throughout the literature researchers ascribe to the false assumptions that effects of family size should be adjusted for and that this could be achieved through the use of ratio variables. Second, using a data-simulation approach, we demonstrate that by using currently recommended statistical practices, researchers are bound to frequently draw incorrect conclusions. And third, we re-examine the empirical evidence of the fraternal birth-order effect in men and women by using a novel specification-curve and multiverse approach to meta-analysis (64 male and 17 female samples, $N = 2,778,998$). When analyzed correctly, the specific association between the number of older brothers and homosexual orientation is small, heterogenous in magnitude, and apparently not specific to men. In addition, existing research evidence seems to be exaggerated by small-study effects.

## INTRODUCTION

The research claim that each older brother increases the odds of homosexual orientation in later-born males is one of the oldest and most widely accepted ideas in the literature on human sexuality (*e.g.*, *Balthazart, 2018*; *Blanchard & Bogaert, 1996a*, *1996b*) and is referred to as the *fraternal birth-order effect* (FBOE). Its precise formulation requires the additional qualifications that other sibling types, namely "older sisters, younger brothers and younger sisters have no effect on these odds" (*Blanchard, 2018a*).[1] As of now, proponents of the FBOE seem to be certain that the association between the number of older brothers and male sexual orientation reflects a causal biological mechanism known as the *maternal immune hypothesis* (MIH; *e.g.*, *Blanchard, 2001*; *Bogaert et al., 2018*; *Bogaert & Skorska, 2011*). The MIH states that Y-linked proteins of XY-male foetuses may enter the maternal system prenatally or perinatally. The maternal immune system supposedly reacts to this "alien" tissue by producing antibodies. These antibodies are released, once Y-linked proteins originating from subsequent XY-male foetuses are detected. Further, these antibodies may enter the circulation of the foetus, where they are hypothesised to modulate the proliferation of "sex-dimorphic brain regions", thus contributing to the development of homosexual orientation. It is important to emphasize that the MIH has little direct (immunological or antibody) evidence to show for and is better described as speculation rather than a fact. As of this writing, only a single study on the MIH has been published (*Bogaert et al., 2018*).

Numerous sources report an estimated increase in the odds of homosexual orientation of approximately 33% per older brother (*Blanchard, 2004*, *2018a*; *Blanchard & Bogaert, 1996b*), with a recent publication on the effect (*Blanchard et al., 2021*) purporting a range of 30–40%. To put this in perspective, suppose that a man who has no older brothers is homosexual with probability 0.02 (an often-repeated estimate in the FBOE literature; *e.g.*, *Cantor et al., 2002*). This is equivalent to 0.02/(1−0.02) = 0.0204 odds of being homosexual. Then if each older brother increases the odds of being homosexual by 33%, the odds that a man who has one older brother is homosexual are 0.0204·1.33 = 0.027, or equivalently, the probability of that man being homosexual should now be 0.027/(1 + 0.027) = 0.026. Generalizing this calculation, the odds that a man with $x$ older brothers is homosexual are given by 0.0204·1.33$^x$. Consequently, the probabilities are 0.03, 0.04, and 0.06 for men with two, three or four older brothers, respectively.

The reason why the FBOE is interpreted causally seems to go back to the types of statistical analyses used in the context of FBOE research. Many studies, especially the early ones (*Blanchard & Bogaert, 1996b*), used logistic regression models to analyze the association between the variables "number of older brothers" (as a predictor variable) and "homosexual orientation" (as an outcome variable). The exponentiated regression coefficients of a logistic regression model can be interpreted as odds ratios. Odds ratios are often briefly (and perhaps insufficiently) described as reflecting the *increase* (or decrease) in the odds per one-unit difference in the predictor variable. This does not mean that regression coefficients from logistic regression models can be interpreted causally as perhaps the everyday language meaning of the phrase "increase in odds" would suggest.

[1] *Blanchard & Lippa (2021*, p. 802*)* recently seemed to have suggested that this definition should be amended, adding that "older sisters also increase these odds, although to a lesser extent." This additional qualifier was added upon observing a difference in the number of older sisters between homosexual and heterosexual men in one sample.

It would be more accurate to describe these odds ratios in terms of comparing the odds of homosexual orientation between two groups: The individuals of the first group all share the same value $x$ and the individuals in the second group all share the value $x + 1$ in the predictor variable of interest, while all individuals (irrespective of the group that they belong to) share the same values in all of the remaining predictor variables in the model. The exponentiated regression coefficient of the predictor variable of interest is given by the odds of homosexual orientation in the second group divided by those same odds in the first group.

Regardless of whether the causal model put forth by Blanchard and colleagues has much evidence to show for, if it is assumed as given, it necessitates a few key observations (*e.g.*, *Blanchard, 2004*; *Bogaert & Skorska, 2011*; *Blanchard, 2018c*): First, homosexual men should have more older brothers than heterosexual men on average or, equivalently, a positive association between homosexual orientation and the number of older brothers among men should be observable. Second, homosexual men should also have slightly more older sisters than heterosexual men (*i.e.*, a positive association between the number of older sisters and homosexual orientation among men). Importantly, this latter difference (or association) should be weaker than the difference (or association) with respect to older brothers; this represents the main theoretical estimand of the FBOE. The weaker association between older sisters and homosexual orientation is claimed to arise due to a positive correlation between the number of older brothers and older sisters (*e.g.*, *Blanchard, 2014*)[2]. Of course, these two observations necessitate that all relevant confounding variables are adjusted for in a statistical model (*e.g.*, *Blanchard, 2014*), which we will assume as given throughout. Third, no (comparable) difference in the number of older brothers should be observable between homosexual and heterosexual women. Not only has this null effect among women been asserted (*e.g.*, *Bogaert & Skorska, 2011*), but it seems also to follow from the absence of Y-chromosomes in XX-female foetuses, and hence the absence of Y-linked proteins which could enter the maternal system and elicit an immune response. Arguably, a similarly-sized effect in women would be incompatible with the current formulation of the MIH. While there are well over 50 reports claiming to support the observation that homosexual men have more older brothers but not more (or only slightly more) older sisters than heterosexual men, only a few studies have compared these variables in homosexual *vs.* heterosexual women (see Part III, below).

The biological explanation and framework involving the MIH was formulated *post-hoc*; that is, only after a greater number of older brothers in homosexual *vs* heterosexual men and the lack of such a difference in homosexual *vs* heterosexual women had been observed (*Blanchard & Bogaert, 1996a*, *1996b*). Moreover, the claim that it is solely the number of older brothers and not the number of older siblings (*i.e.*, older brothers *and* sisters) in general, which increases the odds of homosexual orientation, seems to rest on a misinterpretation of statistical significance (see *Gelman & Stern, 2006*; Part I, below). For instance, *Blanchard & Bogaert (1996a)* concluded that the presence of a statistically significant association between sexual orientation and the number of older brothers, and the simultaneous absence of statistical significance for the association between sexual orientation and the number of older sisters, is compatible with a model of only older

---

[2] Only recently (*Blanchard & Lippa, 2021*) have researchers begun to ascribe group differences in the number of older siblings to substantive biological factors. This also prompted the formulation of an auxiliary to the MIH, purporting that mothers of homosexual men must have had more miscarriages of XY-male fetuses, as opposed to mothers of heterosexual men, and thus greater exposure to Y-linked proteins (*Blanchard et al., 2021*; *Blanchard & Lippa, 2021*).

brothers increasing the odds of homosexual orientation. However (*Gelman & Stern, 2006*), their results were also compatible with a model of older brothers *and* older sisters increasing these odds. As we show below, most of the literature on the FBOE failed to address this conflict between these two competing models (a specific older brother effect *vs* a more general older sibling effect) by making inferences about the main theoretical estimand of a specific older brother effect, based on results compatible with both models.

In addition to the collection of primary studies, there are by now seven meta-analyses of this research literature (*Blanchard, 2004*, *2018a*, *2018b*; *Blanchard et al., 2020*, *2021*; *Blanchard & VanderLaan, 2015*; *Jones & Blanchard, 1998*), every single one of which concluded that homosexual men had more older brothers on average, but not more older sisters. While all of these meta-analyses were conducted by the same group of researchers, they differ considerably with respect to their primary goals stated, the subsets of included samples or studies, and the type of meta-analytic model fitted to the data. An overview of these previous meta-analyses can be found in Table 1. Below, we focus primarily on the fourth (*Blanchard, 2018a*), fifth (*Blanchard, 2018b*), and seventh (*Blanchard et al., 2021*) of these meta-analyses for the following two reasons: First, there is a substantial overlap of the samples included in the different meta-analyses listed in Table 1, with a total of 54 unique samples. The three meta-analyses we focus on included a total of 45 of these unique samples. Second, and most relevant to the discussion in Part I below, these three meta-analyses employed a common effect-size metric, namely the *older brothers odds ratio* (*OBOR*; Table 2), whilst two of the existing seven meta-analyses did not report any effect size at all (*Blanchard, 2004*; *Blanchard & VanderLaan, 2015*). Moreover, the set of samples comprising the observations for the remaining two meta-analyses (*Blanchard et al., 2020*; *Jones & Blanchard, 1998*) was fully contained within (*i.e.*, merely representing a subset of) the combined set of samples considered in the meta-analyses by *Blanchard (2018a*, *2018c*) and *Blanchard et al. (2021)*. We review and evaluate the first (*Jones & Blanchard, 1998*), second (*Blanchard, 2004*), third (*Blanchard & VanderLaan, 2015*), and sixth (*Blanchard et al., 2020*) of these meta-analyses in detail in Supplement S1.

In this article, we adopt the framework laid out by Blanchard and other researchers on the FBOE and foremost focus on the necessary observation of homosexual men having more older brothers relative to older sisters than heterosexual men (*i.e.*, the main theoretical estimand) and that there is no comparable observation in homosexual *vs* heterosexual women. In the following, we show converging evidence that all of the evidence in favor of the first key observation (solely the number of older brothers is associated with homosexual orientation in men) is based on incorrect statistical reasoning and that the specific association between the number of older brothers and homosexual orientation is (a) much smaller than previously claimed, (b) highly variable across different samples, (c) not specific to men, and (d) possibly exaggerated due to the influence of small-study effects.

The following three parts scrutinize the statistical and empirical foundations of the key observations necessitated by the FBOE and MIH from different angles: In Part I, we show mathematically that currently recommended effect sizes and variable transformations (*Blanchard, 2014*, *2018a*, *2020*) all are falsely advertised as quantifying the main theoretical
**Table 1 Overview of previous meta-analyses on the greater number of older brothers among homosexual *vs* heterosexual men.**

| Study | #Samples | Group | N | Effect size | Goal |
|---|---|---|---|---|---|
| *Jones & Blanchard (1998)* | 9 | Homo<br>Hetero | 827<br>2,115 | Fraternal- and sororal indices | Primary goal was to determine whether older sisters showed any association with homosexual orientation in men. |
| *Blanchard (2004)* | 14 | Homo<br>Hetero | 3,181<br>6,962 | none | Each of 28 homo- and heterosexual groups (14 samples) was treated as an independent observation. *P* value of change in logistic regression's deviance associated with the removal of the samples' average number of older brothers from the list of predictors was statistically significant and therefore interpreted as supporting the FBOE. |
| *Blanchard & VanderLaan (2015)* | 14 | Homo<br>Hetero | -<br>- | none | Sign-test meta-analysis over selection of samples not collected by Blanchard or VanderLaan or any other frequent collaborators to mitigate the potential of experimenter (or lab) bias. |
| *Blanchard (2018a)* | 30 | Homo<br>Hetero | 7,140<br>12,837 | *OBOR* | Three primary stated goals: (1) Assess the effect of family size on the FBOE, (2) assess whether the magnitude of the FBOE is stronger in "feminine" as opposed to more "masculine" samples, (3) update previous meta-analyses and examine the reliability of the FBOE. |
| *Blanchard (2018b)* | 6 | Homo<br>Hetero | 3,386<br>445,301 | *OBOR* | Primary goal was to respond to *Zietsch (2018)* who questioned the in- and exclusion criteria in *Blanchard (2018a)*, resulting in the exclusion of all available probability samples. *Blanchard (2018b)* thus conducted a meta-analysis over six probability samples. |
| *Blanchard et al. (2020)* | 14 | Homo<br>Hetero | 823<br>1,885 | *OR* of second- *vs* first-born sons | Primary goal was to assess the performance of the *OR* of second- *vs* firts-born sons. This *OR* is computed by restricting the sampling space to individuals who reported exactly one brother but any number of sisters and comparing the ratios of second- *vs* first-born sons in homo- *vs* heterosexual men. In light of Parts I and II below it is easy to see that, just like the *OBOR*, this *OR* fails to account for a more general excess of older siblings. See Supplemental Material for a detailed account of *Blanchard et al. (2020)*. |
| *Blanchard et al. (2021)* | 24 | Homo<br>Hetero | 5,963<br>12,250 | *OBOR* | The primary stated goals of this meta-analysis were (1) "to examine the evidence for the FBOE in pedophiles," (2) to compare its strength to that of the FBOE in individuals attracted to mature adults ("teleiophiles"), and (3) to determine if an excess of older sisters could be detected in these two groups. |

**Note:**
#Samples, number of samples included in the meta-analysis; Group, sexual orientation (Homo, homosexual; Hetero, heterosexual); *N*, number of participants; "-", not reported; *OBOR*, older brothers odds ratio; *OR*, odds ratio.

estimand, while adjusting for relevant confounders. Moreover, the shortcomings of these methods are based on the unfounded claims that (a) statistical models must adjust for a confounding effect of overall family size (*Blanchard, 2014*, *2018a*) and that (b) this could be achieved through the use of ratio variables. As a result, existing claims about the magnitude, consistency, and specificity of the association between the number of older brothers based on these methods are methodological artefacts and thus spurious. In Part II, we illustrate the inadequacy of these methods using simulated data. We show that in combination with null-hypothesis significance testing (*e.g.*, *Gigerenzer, 2018*) researchers are bound to draw incorrect conclusions about the magnitude of the difference in the association between homosexual orientation and older brothers at a rate much greater than expected. Having shown that these methods are incapable of quantifying the association between the number of older brothers and homosexual orientation in isolation, we conduct a new meta-analysis (the eighth one about the FBOE literature), thereby correcting previous estimates of this association. More precisely, in Part III, we present

**Table 2 Equations of currently recommended measures for quantifying the greater number of older brothers in homosexual as compared to heterosexual men.**

| Introduced by | Measure | Equation |
|---|---|---|
| *Blanchard (2018a)* | Older brothers odds ratio | $OBOR = \dfrac{\#\text{OB}_{\text{Hom}}/\#\text{Other}_{\text{Hom}}}{\#\text{OB}_{\text{Het}}/\#\text{Other}_{\text{Het}}}$ |
| *Blanchard (2018c)* | Older sisters odds ratio | $OSOR = \dfrac{\#\text{OS}_{\text{Hom}}/\#\text{Other}_{\text{Hom}}}{\#\text{OS}_{\text{Het}}/\#\text{Other}_{\text{Het}}}$ |
| *Blanchard (2014)* | Modified ratio of older brothers | $MROB_i = \dfrac{\#\text{OB}_i + 0.33}{\#\text{Other}_i + 1}$ |
| | Modified ratio of older sisters | $MROS_i = \dfrac{\#\text{OS}_i + 0.33}{\#\text{Other}_i + 1}$ |
| | Modified proportion of older brothers | $MPOB_i = \dfrac{\#\text{OB}_i + 0.25}{\#\text{All}_i + 1}$ |
| | Modified proportion of older sisters | $MPOS_i = \dfrac{\#\text{OS}_i + 0.25}{\#\text{All}_i + 1}$ |

Note:
#OB, number of older brothers; #OS, number of older sisters; #YB, number of younger brothers; #YS, number of younger sisters; $i$, indexes the observations (participants).

re-analyses of the three previous meta-analyses, which employed ambiguous effect-size metrics and provide a glimpse at the multiverse of possible meta-analyses about this literature, using a specification-curve meta-analytic approach (*Pietschnig et al., 2022*; *Simonsohn, Simmons & Nelson, 2020*; *Steegen et al., 2016*; *Voracek, Kossmeier & Tran, 2019*). In addition, we present the first set of meta-analyses on the association between the number of older brothers and homosexual *vs* heterosexual orientation in women, as well as meta-analytic syntheses on the difference between men and women regarding the magnitude of this association (see Fig. 1 for a PRISMA flow diagram; *Page et al., 2021*). The results of Part III converge with the key findings from Parts I and II, in that there appears to be little, if any, prior empirical foundation in the FBOE and the MIH.

During the process of peer review two article were published, which are of direct relevance for the present study. First, *Blanchard & Skorska (2022)* published a comment to a preprint of the present study. Second, *Ablaza, Kabátek & Perales (2022)* published administrative population-level register data from the Netherlands ($N = 9{,}073{,}496$), providing compelling evidence for the FBOE among both men and women. They used data of formally recognized same-sex unions (*i.e.*, marriages or registered partnerships), which (even though previously denied by *Blanchard (2018a)*; see Part III) obviously is one of the most valid indicators of homosexual orientation currently available.

Concerning the first article, we would like to note that a journal response to a preprint appears highly unusual to us. We further want to highlight that *Blanchard & Skorska (2022)* completely misconstrued our work by claiming that we wrote there is no evidence for the FBOE in men or women. This is not what we claim, neither in the present study, nor in the preprint. Moreover, *Blanchard & Skorska (2022)* now conclude that there is evidence for the FBOE in both men and women, which, prior to our preprint and the *Ablaza, Kabátek & Perales (2022)* study, was adamantly denied by this group of
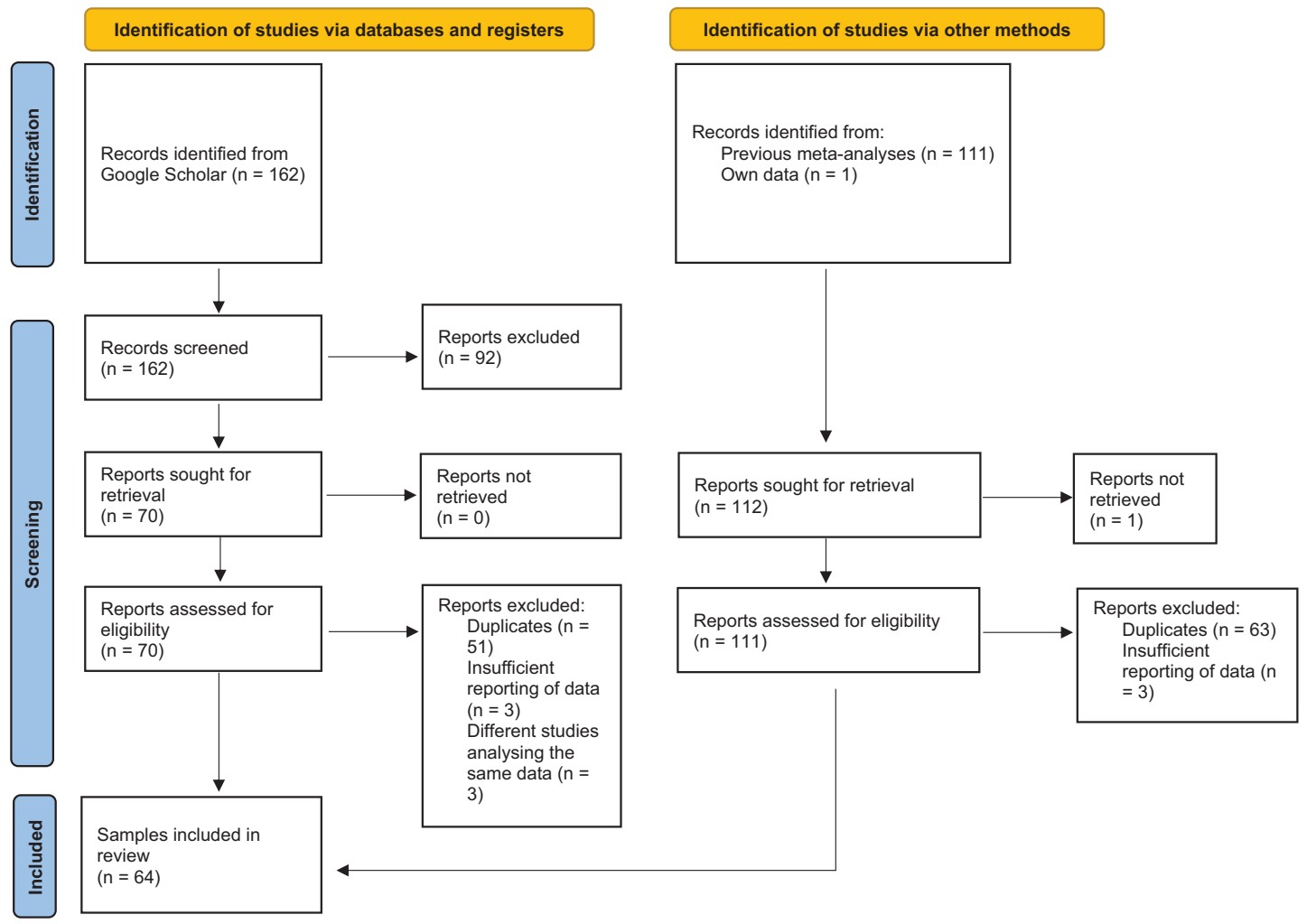

**Figure 1 PRISMA flow diagram.** The PRISMA flow diagram details the process of our literature search in Part III.

researchers. All data provided in *Blanchard & Skorska (2022)* was also already included in our previous analysis.

Concerning the second article, there is now robust, independent, and high-quality evidence for the FBOE among both men and women. The FBOE, hence, appears not specific to men only, as was previously assumed. Yet, as the sample size of *Ablaza, Kabátek & Perales (2022)* is more than thrice the total sample size of all previously available data (see Part III), we chose not to include this study into the meta-analyses of the present study. Including this study in our meta-analytic computations would have rendered them mostly superfluous, as the results of *Ablaza, Kabátek & Perales (2022)* would have driven almost any meta-analytic result. Hence, we used this study as independent external evidence to compare our results with. Thus, as we show in the following, the results of Part III not only converge with the findings of Parts I and II, but also with recent, empirical, and independent evidence provided by *Ablaza, Kabátek & Perales (2022)*.

# PART I. CURRENT APPROACHES DO NOT QUANTIFY THE THEORETICAL ESTIMAND OF INTEREST: INSIGHTS FROM PROBABILITY CALCULUS

To save space and provide a much-needed level of clarity and rigor, we present our arguments using technical terms and basic mathematical formulations found in any introductory textbook on the subject of probability (*e.g.*, *Blitzstein & Hwang, 2019*). We provide more lengthy explanations and intuitive rephrasing of Eqs. (1–4) in Supplement S2.

## Notation

The most widely used effect size for quantifying the number of older brothers among homosexual men in comparison to heterosexual men, the *older brothers odds ratio (OBOR)*, is defined at the level of siblings, as reported by study participants. Hence, much of the observational evidence for the FBOE (including most of the meta-analyses) rests on a rather unusual approach for comparing the number of older brothers between groups. Intuitively, birth-order effects are analyzed by considering the sampled individuals as the units of analysis (*e.g.*, *Rohrer, Egloff & Schmukle, 2015*). For instance, to quantify the association between an individual's sexuality and their number of older brothers, one could use either of these variables as the outcome and the other as one of the predictors in a (generalized) linear model. However, the *OBOR* (but also other recommended statistical approaches) does not build on this intuition and treats the reported siblings as the units of analysis. We thus need to introduce some notation to be able to show how precisely the *OBOR* fails to address the effect of interest.

If the reported siblings are regarded as the sample, the following events can be defined: A given sibling can be either an older brother, an older sister, a younger brother, or a younger sister of the study participant who reported him or her. These events are denoted by OB, OS, YB and YS, respectively. The complement of any event is denoted by the superscript $c$ (as in "complement"). For instance, $OB^c$ denotes the event that a given sibling is either an older sister, a younger brother, or a younger sister. The event $OB^c$ will mostly be referred to as Other, meaning that the sibling is not an older brother, but any of the other three possible sibling types. There are instances in this work, where Other refers to siblings who are not older sisters, which should then be clear from the given context. Furthermore, we denote the event that a given sibling is an older sibling (*i.e.*, the union of OB and OS) by Older. Its complement $Older^c$ denotes the event that a given sibling is a younger sibling (for the sake of simplicity, twins, triplets, *etc.*, are not accounted for).

A sibling can either be reported by a homosexual or by a heterosexual study participant. These events are denoted by Hom and Het, respectively, and are regarded as complementary. To aid readability, we use Het instead of $Hom^c$, as the events Hom and Het are extensively referred to in the following.

The probability that an event A occurs, is denoted by P(A). The probability of an intersection of two events A and B is denoted by P(A, B). The probability of an event A

conditional on an event B (*i.e.*, the conditional probability of A given B) is denoted by P(A|B).

In order to distinguish the events which pertain to individual siblings from the number of times these events occur in a sample, we use the number sign (#), which in this context should be read as "number of." Hence, #OB simply denotes the number of all reported older brothers in a sample. The subscripts Het and Hom are used to denote whether the total number of a given sibling type refers to that reported by homosexual or heterosexual participants. For instance, the number of older brothers reported by homosexual participants is denoted by $\#OB_{Hom}$.

We treat the terms "probability of an event" and "proportion of times an event occurred" interchangeably, with the latter being an estimate of the former.

*Blanchard (2014, 2018a)* and *Blanchard et al. (2020, 2021)* warned that the specific (average) difference in the number of older brothers between homosexual and heterosexual individuals due to the FBOE may go undetected (or be obscured), if the mean number of all siblings reported by the group of homosexual men is appreciably smaller than the mean number of all siblings reported by the group of heterosexual men.[3]

To counter potential mitigation of the relationship between homosexual orientation and the number of older brothers due to the suspected confounding variable "differences in family size between groups", Blanchard suggested that it is necessary to adjust for either the number of other siblings (#Other; *Blanchard, 2014, 2018a, 2020*) or the number of all siblings (#All; *Blanchard, 2014*) in a statistical model (*i.e.*, test) comparing these two groups.[4]

Over the years, three new methods, designed to achieve this goal, have been developed, and increasingly are used in primary publications and meta-analyses on the FBOE. These are the already mentioned *OBOR* (*Apostolou, 2020*; *Blanchard, 2018a, 2018b*; *Skorska & Bogaert, 2020*), the *modified ratio of older brothers* (*MROB*; *Blanchard, 2014*), and the *modified proportion of older brothers* (*MPOB*; *Blanchard, 2014*; *e.g.*, *Skorska & Bogaert, 2020*).

For now, we show that the *OBOR*, *MROB*, and *MPOB* quantify the more general average difference (or association) in the number of older siblings (*i.e.*, brothers and sisters) between homosexual and heterosexual men, rather than the more specific difference in the number of older brothers only (or the association between the number of brothers and sexual orientation, only).

## The *OBOR*

As the name implies, the *OBOR* (see Table 2) is an odds ratio, and it can thus be expressed in terms of conditional probabilities:

$$OBOR = \frac{P(OB|Hom)/[1 - P(OB|Hom)]}{P(OB|Het)/[1 - P(OB|Het)]}. \tag{1}$$

Now suppose we observe an *OBOR* > 1, which, according to the interpretation of *Blanchard (2018a, 2018b, 2018c)* and *Blanchard et al. (2020, 2021)*, should be regarded as evidence for the FBOE.

[3] The term "undetected" refers to the lack of statistical significance of a size-$\alpha$-test of a respective effect size. Per common practice, $\alpha$ is chosen to be 0.05, and there seems to be some allowable margin of freedom or variation as to whether the alternative hypothesis should be one-sided or two-sided (*Blanchard & VanderLaan, 2015*).

[4] This necessity is claimed to have been demonstrated by *Blanchard (2014)*, which upon closer examination, however, seems to be based on circular reasoning. *Blanchard (2014)* appears to have assumed that a set of real participant data contained evidence for a greater number of older brothers in homosexual men, as opposed to heterosexual men, and then concluded that the data showed evidence of a greater number of older brothers among homosexual men, when using a transformation of predictor variables which supposedly adjusts for #All or #Other. These transformations are known as the *Modified Ratio of Older Brothers* (*MROB*) and the *Modified Proportion of Older Brothers* (*MPOB*; Table 2) and are discussed in detail below.

For the *OBOR* to become greater than unity, the numerator in Eq. (1) must be greater than the denominator. Formally, this relationship can be stated as

$$OBOR > 1 \Leftrightarrow P(OB|Hom) > P(OB|Het), \tag{2}$$

Equation (2) is just an algebraic transformation of Eq. (1) (see Supplement S3 for a derivation).

By conditioning on the event Older, the *law of total probability* (*e.g.*, *Blitzstein & Hwang, 2019*) allows for the factorisation of both P(OB|Hom) and P(OB|Het), as follows:

$$P(OB|Hom) = P(OB|Older, Hom)P(Older|Hom)$$
$$+ P(OB|Older^c, Hom)P(Older^c|Hom) \text{ and,}$$
$$P(OB|Het) = P(OB|Older, Het)P(Older|Het)$$
$$+ P(OB|Older^c, Het)P(Older^c|Het).$$

It is impossible for a sibling to be both an older brother and a younger sibling (*i.e.*, the event Older$^c$ *and* OB is impossible). Therefore, P(OB|Older$^c$,Hom) and P(OB|Older$^c$,Het) are both zero, reducing the above factorisations to

$$P(OB|Hom) = P(OB|Older, Hom)P(Older|Hom), \text{ and}$$
$$P(OB|Het) = P(OB|Older, Het)P(Older|Het). \tag{3}$$

One can think of P(OB|Older,Hom) and P(OB|Older,Het) as first restricting the sampling space from the set of all reported siblings to the subset which contains only older siblings and then computing the proportion (*i.e.*, probability) of brothers within this subset of older siblings for the homosexual and heterosexual group, respectively.

In combining Eqs. (2 and 3), we obtain the following equivalent inequalities:

$$OBOR > 1$$
$$\Leftrightarrow P(OB|Hom) > P(OB|Het) \tag{4}$$
$$\Leftrightarrow P(OB|Hom, Older)P(Older|Hom) > P(OB|Het, Older)P(Older|Het).$$

Notice that Eq. (4) (*i.e.*, an *OBOR* larger than unity) can hold even if P(OB|Hom,Older) ≤ P(OB|Het,Older). That is, one may observe that the proportion (or probability) of older brothers in the subset of older siblings is smaller in the homosexual group, as opposed to the heterosexual group. However, the *OBOR* (considering the entire set of reported siblings) may still be greater than unity, due to the proportion (or probability) of older siblings being sufficiently greater in the homosexual group (*i.e.*, P(Older|Hom) > P(Older|Het)).

This contradicts the key necessary observation of the FBOE, as the homosexual group must show a greater proportion (or probability) of older brothers among older siblings. This observation is required by the claim that older brothers increase the odds of homosexual orientation in men, but older sisters do not (or to a lesser extent). The *OBOR* thus insufficiently accounts for a relevant confounder (here: Older) and could potentially lead to the conclusion that homosexual men have more older brothers relative to older sisters, when in fact the opposite is true. In other words, the *OBOR* is ambiguous: It is affected by both, the proportions of older brothers among older siblings as well as the proportions of older siblings among all siblings.

It follows that–contrary to previous claims (*Blanchard, 2018a*, *2020*; *Blanchard et al., 2021*)–for an *OBOR* of one to be considered to adequately represent the null hypothesis of no group difference in the number of older brothers relative to older sisters, the proportion (or probability) of observing an older sibling must be identical in both groups (this is equivalent to the requirement that the number of older siblings be equal in both groups).

To see this, suppose that P(OB|Older,Het) and P(OB|Older,Hom) are equal to 0.515, which is a widely agreed upon population estimate of the proportion of male births, and hence for the probability of being born male (*e.g.*, *Grech & Mamo, 2020*). Further assume that homosexual men have more older brothers and sisters compared to heterosexual men; that is, there is no specific difference in the number older brothers that exceeds the difference in the number of older siblings. This can be achieved by requiring that the probabilities of observing an older sibling in the homosexual and heterosexual groups are equal to the median proportions of older siblings of P(Older|Hom) = 0.58 and P(Older|Het) = 0.48 reported across the 45 samples included in the three meta-analyses using the *OBOR* (*Blanchard, 2018a*, *2018b*; *Blanchard et al., 2021*). The *OBOR* under the null hypothesis is then given by

$$OBOR_{H_0} = \frac{(0.515 \times 0.58)/(1 - 0.515 \times 0.58)}{(0.515 \times .48)/(1 - 0.515 \times 0.48)} = 1.30. \tag{5}$$

Equation (5) states a more adequate value for a null hypothesis, which would be expected if the probability of observing an older brother among the older siblings of homosexual and heterosexual men were equal to the population probability of being born male (0.515), with the additional assumption that the probability of observing an older sibling (brother or sister) is greater (0.58) in the homosexual as compared to the heterosexual (0.48) group. That is, this null model assumes that the two groups differ only with respect to the proportion of older siblings, but that there is no specific (or additional) difference in the proportion of older brothers in the homosexual group (as the FBOE requires).

The fourth meta-analysis (*Blanchard, 2018a*) reported an estimated random-effects mean *OBOR* of 1.47, 95% *CI* [1.33–1.62], combining a total of 30 (31; see Part III) samples. For a distinct subset of 18 samples denoted "non-feminine/cisgender" men, *Blanchard (2018a)* reported a mean *OBOR* of 1.27, 95% *CI* [1.20–1.35]. Both of these estimates can be regarded as more or less compatible with the value of 1.30 derived in Eq. (5). In a commentary to *Blanchard (2018a)*, *Zietsch (2018)* cautioned about the evidence in favor of the FBOE contained in this meta-analysis, mainly due to concerns about questionable and arbitrary inclusion criteria used by *Blanchard (2018a)*, which led to the exclusion of all probability samples known at that time.[5]

*Blanchard (2018b)* replied by putting forth new evidence for the specific difference in the number of older brothers between homosexual and heterosexual men in a fifth meta-analysis, this time amalgamating probability samples only. The estimated mean *OBOR* over six such probability samples was 1.21, 95% *CI* [1.13–1.30], and was thus judged to be incompatible with the null hypothesis of *OBOR* = 1.

[5] The term "probability sample" was defined by *Zietsch (2018*, p. 1) as "samples selected randomly with respect to the independent variable–sexual orientation in this case."

However, in light of the above Eq. (5), this estimated *OBOR* appears compatible with a model of an unspecific group difference in older brothers *and* sisters. That is, the observed *OBOR* > 1 may not reflect the claimed specific difference in the number of older brothers. It goes without saying that a more general phenomenon like a larger number of older siblings in homosexual men as opposed to heterosexual men might be seen as a theoretically interesting observation on its own. The claims surrounding the FBOE, however, are highly specific and the necessary observational consequence (conditional on the assumptions underlying the FBOE) of a surplus of older brothers relative to older sisters cannot be quantified using the *OBOR*.

## Addressing group differences with respect to the number of older sisters

It appears that Blanchard and colleagues were aware of the *OBOR*'s inability to distinguish between a specific difference in the number of older brothers and a more general difference in the number older siblings between homosexual and heterosexual men, as shortly after the fifth meta-analysis, the *older sisters odds ratio* (*OSOR*; see Table 2) was introduced (*Blanchard, 2018c*). Analogous to the *OBOR*, the *OSOR* is simply the odds ratio of older sisters among all the siblings reported by homosexual and heterosexual men. *Blanchard (2018c)* reported that for 29 out of the 36 samples included in the fourth and fifth meta-analyses, the *OBOR* was numerically greater than the *OSOR*, concluding that the group difference with respect to the number of older brothers is greater than the difference with respect to the number of older sisters as predicted by the FBOE. However, this result reveals nothing about the size of the difference in the *OBOR* and the *OSOR* and might have led to a different conclusion if these differences had been weighted and combined in a meta-analysis. In similar vein, the seventh meta-analysis (*Blanchard et al., 2021*) considered both the *OBOR* and the *OSOR* in a set of 24 samples and estimated a random-effects mean *OBOR* of 1.28, 95% *CI* [1.22–1.35] for the entire set (which again is an estimate very compatible with the value derived in Eq. (5)). In addition, *Blanchard et al. (2021)* reported estimated random-effects mean *OSOR*s of 1.11, 95% *CI* [1.05–1.17], for what they denoted the "teleiophiles" subgroup and 1.15, 95% *CI* [0.99–1.34], for the "pedophiles and hebephiles" subgroup. *Blanchard et al. (2021)* describe pedophiles as men predominantly attracted to children before puberty, hebephiles as men predominantly attracted to children in puberty, and teleiophiles as men attracted to postpubertal, mature individuals.

The implication of the above seems to be that, in order to ensure that an observed group difference in the number of older brothers *via* the *OBOR* is not an artefact of a more general group difference in the number of older siblings (*i.e.*, older brothers and sisters), one should interpret the *OBOR* and the *OSOR* in tandem.

However, having to interpret two separate, but correlated, statistics introduces an unnecessary degree of complexity. Moreover, the apparent need to report both the *OBOR* and the *OSOR*, as conveyed by *Blanchard et al. (2021)*, illustrates that it is the number (or equivalently, the proportion) of older siblings that needs to be adjusted for–not total family

size or number of "other" siblings, as has repeatedly been asserted (*Blanchard, 2014*, *2018a*, *2018b*).

If researchers want to use an odds ratio in order to quantify the difference in the number of older brothers relative to older sisters between homosexual and heterosexual men, then the *OBOR* can be fixed by simply omitting the younger siblings from the denominator of the *OBOR*. Then, only the relevant subset of older siblings is considered. The odds of observing an older brother in either group are now given by the ratio of older brothers to older sisters, and consequently the odds ratio (*OR*) is given by

$$OR = \frac{\#\text{OB}_{\text{Hom}}/\#\text{OS}_{\text{Hom}}}{\#\text{OB}_{\text{Het}}/\#\text{OS}_{\text{Het}}}. \tag{6}$$

This *OR* adjust for the confounding effect of any differences in the number of older siblings in either group. Alternatively, one could also use the difference in the proportion of older brothers among older siblings as an effect size (see supplemental materials to Part III, below).

The numerator and denominator of Eq. (6) may be conceived as the male-to-female sex ratio within the subset of older siblings among homosexual and heterosexual participants, respectively. Under the assumption of the FBOE and the implied greater number of older brothers among older siblings of homosexual men, one would expect this ratio of sex ratios (*i.e.*, an odds ratio) to be greater than unity.

We note again that the use of *OR*s (the *OBOR* and the *OR* in Eq. (6)) implies a shift from the level of participants to the siblings reported by these participants. For instance, suppose participant *i* reports to have three siblings, one older brother and two older sisters. Use of the *OBOR* or *OR* implies that the number of older brothers is compared to the number of other sibling types. In the example just given, each of the three siblings provides one observation of the binary (or Bernoulli) variable "older brother" (1 = sibling is older brother, 0 = sibling is not older brother). It follows that in this example, a single participant provides multiple (three) data points and thus naturally the units of analysis (*i.e.*, siblings) are nested, or grouped, within participants.

## The modified ratio and proportion of older brothers

The *MROB* (see Table 2 for a definition) can be regarded as an estimator of the odds of observing an older brother among all the siblings of the *i*th participant, whereas the *MPOB* (see Table 2 for a definition) can be regarded as an estimator of the probability of this event. Hence, given the one-to-one relation between odds and probabilities, the *MROB* and the *MPOB* are related as follows:[6]

$$MROB \approx \frac{MPOB}{1 - MPOB} \quad \text{and} \quad MPOB \approx \frac{MROB}{1 + MROB}.$$

Keeping in mind that we use proportions and probabilities interchangeably, the *MPOB* of the *i*th participant is defined as the probability of observing an older brother within the sibship of the *i*th participant, which can be written as

[6] The reason for that the relation between the *MROB* and the *MPOB* can only be stated approximately is due to the constants (0.33 and 0.25) which are added to both the numerator and the denominator of the *MROB* and *MPOB* (see Table 2), in order to have these indices defined for only children (*Blanchard, 2014*).

$$MPOB_i \approx P(OB|i) = P(OB|Older, i)P(Older|i). \tag{7}$$

Applying once more the law of total probability, it follows that the *MROB* for the *i*th participant is given by:

$$MROB_i \approx \frac{P(OB|i)}{1 - P(OB|i)} = \frac{P(OB|Older, i)P(Older|i)}{1 - P(OB|Older, i)P(Older|i)}. \tag{8}$$

*Blanchard (2014, 2020)* recommended either using the *MROB* or the *MPOB* as predictors in a logistic regression model of participants' sexual orientation, as opposed to using the raw number of older brothers. Equations (7 and 8) suggest that it would not be difficult to come up with a scenario in which, on average, homosexual participants report just as many (or fewer) older brothers among their older siblings as heterosexual participants do, yet, due to more general group differences in the proportion or number of older siblings, the odds (or the probability) of observing an older brother still are greater for homosexual participants. Consequently, any positive association between the probability of homosexual orientation and the odds (or probability) of observing an older brother among all of the siblings of a participant may be compatible with both models: a specific group difference in the number of older brothers (conditional on the number of older siblings), but also with a more general difference in the number of older brothers and sisters.

*Blanchard (2014)* defined two complementary indices for older sisters, the *modified proportion of older sisters* (*MPOS*) and the *modified ratio of older sisters* (*MROS*; see Table 2 for definitions), and interpreted the presence of a statistically significant coefficient for the *MROB* (*MPOB*) and the simultaneous lack of a statistically significant coefficient for the *MROS* (*MPOS*) as evidence for the hypothesis that the number of older brothers, but not the number of older sisters, are related to the probability of homosexual orientation in men. Declaring a difference between two effects (that of the *MROB/MPOB* and that of the *MROS/MPOS*) significant, based on the observation that one effect statistically is significantly different from the null hypothesis, whereas the other is not, is a well-known fallacy in statistical significance testing (*Gelman & Stern, 2006*). A pattern of statistically significant *vs* statistically nonsignificant coefficients does not inform about whether the *MROB*'s (*MPOB*'s) association with sexual orientation can be taken to be greater than that of the *MROS* (*MPOS*).

As discussed next, the rationale for using the *MPOB* and *MROS* seems to be based on a common misconception about ratios, namely, that these would adjust for the variable in their denominator (*Sollberger & Ehlert, 2016*), which was the stated purpose for introducing these ratios into the FBOE literature in the first place (*Blanchard, 2014*).

## Ratios do not adjust for confounding variables

As the sole predictor in a linear model, the *MROB* (and/or the *MROS*) is equivalent to including only the interaction term between the number of older brothers (+0.33) and the reciprocal of the number of all other siblings (+1) into the model (*Kronmal, 1993*; see the corresponding equation in Table 2). However, the constituent variables of the interaction,

(#OB + 0.33) and 1/(#OS + #YB + #YS + 1), are omitted from the model, which implies that the regression coefficients for #OB + 0.33 and 1/(#OS + #YB + #YS + 1) are both set to zero. That is, when using the *MROB* as a predictor, it is not clear whether the statistical effect is driven by the number of older brothers (numerator), or the reciprocal of the number of other siblings (denominator), or some interaction between these two. Similar considerations apply to the use of the *MPOB* and *MPOS*.

Ratios (often referred to as "indices") are ubiquitous in many areas in the social and behavioural sciences, and often their substantive interpretation appears straightforward and meaningful. The statistical analysis of ratios, however, is not at all straightforward (*Wiseman, 2009*; see also *Kronmal, 1993*; *Sollberger & Ehlert, 2016*). Most importantly, it is not the case that ratios adjust for the variable in the denominator (*e.g.*, *Kronmal, 1993*; *Wiseman, 2009*). Instead, in order to adjust for the influence of an assumed confounding variable, one could simply add the constituents of a ratio as a predictor variable of its own to the statistical model.

For instance, a researcher might posit that the relationship between homosexual orientation and the number of older brothers should be positive, but that the number of other siblings (#Other) could attenuate the estimate of this relationship (as suggested by *Blanchard (2014)*). Regressing sexual orientation (*via* an appropriate link function) on #OB and #Other would adjust for the confounding effect of #Other. This regression model is a simple representation of the assumed theoretical model. In using the *MROB* (*MPOB*) for modelling this relationship, the associated regression coefficient does not correspond to the regression coefficient of #OB in the more adequate model, and it is generally not clear how to interpret this regression coefficient. At worst, a spurious relationship between the predictor and outcome variables is introduced (*Kronmal, 1993*; *Wiseman, 2009*).

Nevertheless, in trying to interpret the regression coefficient for #OB, it becomes clear that adjusting for the number of other siblings does not rule out the possibility of an older-sibling effect (*i.e.*, an effect of older brothers *and* sisters). In a logistic regression (as recommended by *Blanchard (2014)*), a positive regression coefficient for #OB would indicate that–while holding the number of other siblings constant–the logit of the probability of homosexual orientation increases as a function of the number of older brothers. The problem with this model lies in holding #Other constant, as there are numerous combinations of its constituent variables #OS, #YB, and #YS, which all could add up to one and the same value for #Other. That is, if the regression coefficient for #OB were in fact driven by a greater number of older brothers *and* sisters, participants who have more older brothers would also have more older sisters, whereas fewer younger brothers and younger sisters. In adjusting for #Other, this information is lost. By analogy, the variable #All (the total number of siblings) is equally unfit for ruling out an older sibling effect, since numerous combinations of #OB, #OS, #YB and #YS would sum up to identical values of #All.

Thus, there is no reasonable justification for adjusting for #Other and #All (see also *Zietsch, 2018*). However, for the goal of quantifying the specific association between the number of older brothers and homosexual orientation, one could adjust for the

confounding effect of the number of older siblings, #Older (or, equivalently, the proportion of older siblings; *Frisch & Hviid, 2006*; *Gelman & Stern, 2006*; *Zietsch, 2018*).

To this end, *Gelman & Stern (2006*, p. 330) suggested the difference between #OB and #OS as one predictor and #Older as a second predictor (in a logistic regression model). If solely the number of older brothers, but not the number of older sisters (or the number of older sisters to a lesser extent than the number of older brothers), were associated with homosexual orientation, then a positive regression coefficient for the difference between #OB and #OS should be observed (the number of older siblings). Alternatively, and to obtain an estimate of the increase in the odds of homosexual orientation, a model with the predictors #OB and #Older could be fitted to the data.

### Part I conclusions

Our findings in Part I boil down to two overarching themes. First, assuming the claims made by Blanchard et al. about which relationships between variables should be observable as given, the *OBOR*, the *MROB*, and the *MPOB* are all intended to adjust for the confounding effect of total family (or sibship) size in statistical analysis. Yet, it is the number of older siblings that should be adjusted for instead, as has already been pointed out before (*Frisch & Hviid, 2006*; *Gelman & Stern, 2006*; *Zietsch, 2018*). Second, ratios do not adjust for the variable in the denominator, which is a common misconception surrounding the use of ratio variables (*e.g.*, *Wiseman, 2009*). Using basic probability calculus, the statistical clarification provided here has thus shown that ratios better should not be used in the way they are used in extant research on the FBOE, and that analyses need to adjust for the number of older siblings, but not for total family (or sibship) size.

## PART II: ASSESSING AND COMPARING THE PERFORMANCE OF RECOMMENDED AND ALTERNATIVE MEASURES USING SIMULATED DATA

Next, we assess the performance of the statistical models and practices recommended and used by *Blanchard (2014*, *2018a*, *2020)* to models, which appropriately adjust for #Older. To this end, we simulated data in *R* (*R Core Team, 2021*), and assessed the frequency with which researchers would falsely conclude that solely the number of older brothers is greater among homosexual men, when in fact, it is the number of older brothers *and* sisters with respect to which the two groups differ (*i.e.*, no specific older brother effect). Equivalently, and in line with the terminology used in the FBOE literature, a greater number of specifically older brothers (not sisters) in homosexual men can be interpreted as "solely older brothers increase the odds of homosexual orientation." We do not use this latter interpretation to mean that older brothers "cause" the odds of homosexual orientation to increase (as usage of the term "increase" might implicate).

### Methods

A detailed description of the simulation study is provided in Supplement S5.
We investigated scenarios of (1) a 33% increase in the odds of homosexual orientation per older brother ($\theta = 0.33$), as is reported throughout the literature (*Blanchard, 2001*;

*Blanchard et al., 1996*); (2) no distinct association between the number of older brothers and sexual orientation ($\theta = 0$); and (3) a 33% decrease in the odds of homosexual orientation per older brother ($\theta = -0.33$). Each older brother increased these odds by a factor of $(1 + \theta)$. For the homosexual sample, the study employed three different values for the proportion of older siblings in a given sample, $\pi$, namely (1) 0.5, (2) 0.6, and (3) 0.7, while for the heterosexual sample $\pi$ was fixed at 0.5. That is, we simulated differences in the number (or proportion) of older siblings, assuming that the proportion of older siblings is equal to the proportion of younger siblings among homosexual individuals, but increasing in the homosexual group.

The median of the mean numbers of all siblings for homosexual participants, $\mu$, across the 45 samples in *Blanchard (2018a*, *2018b)*, and *Blanchard et al. (2021)* was 2.45. In the *equal* condition (1) of the simulation study, this value served as the mean number of all siblings for both the homosexual and heterosexual group. In the *unequal* condition (2), the mean number of siblings for each group were taken from the *Mismatch 2* sample in *Blanchard (2014)*, where the mean number of siblings in the homosexual group was 2.19, and the mean number of siblings in the heterosexual group was 3.31. Blanchard used the *Mismatch 2* sample to demonstrate the inability of tests for mean differences and logistic regression to detect an older brother effect and to promote the use of the *MROB* and *MPOB*.

Thus, there were $3 \times 3 \times 2 = 18$ possible combinations of conditions in this simulation study. For each combination, we fitted 10 different models (see Table 3).

## Models and evaluation

Table 3 lists the equations of the models we fitted to the simulated data. For each model and each combination of $\mu$, $\pi$, and $\theta$, 1,000 replications with a sample size of 700 participants per group each were carried out (*i.e.*, 18,000 replications in total).

The performance of the models was primarily assessed with respect to inferring the state of $\theta$ with null-hypothesis significance testing (NHST), as it is the case throughout the FBOE literature. Only Models 1, 6, 7, and 9 in Table 3 could be interpreted as providing an estimate of $\theta$ (by exponentiating the estimates and subtracting (1)). However, in the FBOE literature, studies rarely report estimates of $\theta$. Instead, all of the currently advocated practices involving the *OBOR*, the *MROB*, and *MPOB* infer about $\theta \neq 0$ given a test of a parameter $\beta$, which does not correspond to $\theta$. $\beta$ represents any of the regression coefficients of interest in Table 3. It is evident that the hypothesis of $\beta = 0$ *vs.* $\beta \neq 0$ need not correspond to a hypothesis test about $\theta = 0$ *vs.* $\theta \neq 0$. Inferences about $\theta$ are nevertheless made in the FBOE literature based on inferences about $\beta$, using $\alpha = 0.05$ (two-sided, with only a few exceptions). The simulation aims to show that relying on NHST of the wrong model inevitably leads to biased conclusions about the true state of $\theta$.

## RESULTS

The results of the simulation study are displayed as error plots in Fig. 2 (see online supplement for code, https://osf.io/3wnhu/). Each plot corresponds to one of the models in

**Table 3 Regression models fitted to simulated data.**

| Model | Equation | Description |
|-------|----------|-------------|
| **Models warned against by *Blanchard (2014)*** | | |
| Model 1 | $\text{logit}[P(\text{Hom}_i)] = \beta_0 + \beta_{\#OB}\#OB_i + \beta_{\#OS}\#OS_i$ | Predicts the logit of the probability of homosexual orientation from the number of older brothers and older sisters. Effect of interest: $\beta_{\#OB}$. |
| **Models recommended by *Blanchard (2018a, 2020, 2014)*** | | |
| Model 2 | $\text{logit}[P(\text{Hom}_i)] = \beta_0 + \beta_{MROB}MROB_i + \beta_{MROS}MROS_i$ | Predicts the logit of the probability of homosexual orientation from the modified ratio of older brothers (*MROB*) and modified ratio of older sisters (*MROS*). Effect of interest: $\beta_{MROB}$. |
| Model 3 | $\text{logit}[P(\text{Hom}_i)] = \beta_0 + \beta_{MPOB}MPOB_i + \beta_{MPOS}MPOS_i$ | Predicts the logit of the probability of homosexual orientation from the *MPOB* and *MPOS*. Effect of interest: $\beta_{MROB}$. |
| Model 4 | $\text{logit}\big[P(OB_{ij})\big] = \beta_0 + \beta_{\text{Hom}}\#\text{Hom}_i$ | Predicts the logit of the probability of observing an older brother among all siblings. Effect of interest: $\ln OBOR = \beta_{\text{Hom}}$. |
| Model 5 | $\text{logit}\big[P(OS_{ij})\big] = \beta_0 + \beta_{\text{Hom}}\#\text{Hom}_i$ | Predicts the logit of the probability of observing an older sister among all siblings. Effect of interest: $\ln OSOR = \beta_{\text{Hom}}$. |
| **Models implied by *Blanchard (2014)*** | | |
| Model 6 | $\text{logit}[P(\text{Hom}_i)] = \beta_0 + \beta_{\#OB}\#OB_i + \beta_{\#All}\#All_i$ | Predicts the logit of the probability of homosexual orientation from the number of older brothers and the number of all siblings. Effect of interest: $\beta_{\#OB}$. |
| Model 7 | $\text{logit}[P(\text{Hom}_i)] = \beta_0 + \beta_{\#OB}\#OB_i + \beta_{\#Other}\#Other_i$ | Predicts the logit of the probability of homosexual orientation from the number of older brothers and the number of other siblings (*i.e.*, siblings who are not older brothers). Effect of interest: $\beta_{\#OB}$. |
| **Models controlling for the number of older siblings** | | |
| Model 8 | $\text{logit}[P(\text{Hom}_i)] = \beta_0 + \beta_{\#OB-\#OS}(\#OB_i - \#OS_i) + \beta_{\#Older}\#Older_i$ | Predicts the logit of the probability of homosexual orientation from the difference of the number of older brothers and older sisters and the sum of older brothers and older sisters (*Gelman & Stern, 2006*). Effect of interest: $\beta_{\#OB-\#OS}$. |
| Model 9 | $\text{logit}[P(\text{Hom}_i)] = \beta_0 + \beta_{\#OB}\#OB_i + \beta_{\#Older}\#Older_i$ | Predicts the logit of the probability of homosexual orientation from the the number of older brothers and the sum of older brothers and older sisters. Effect of interest: $\beta_{\#OB}$. |
| Model 10 | $\text{logit}\big[P(OB_{ij})\big] = \beta_0 + \beta_{\text{Hom}}\#\text{Hom}_i$ | Predicts the logit of the probability of observing an older brother among older siblings (*i.e.*, the sampling space is restricted to older siblings). Effect of interest: $\ln OR = \beta_{\text{Hom}}$. |

**Note:**
The subscripts $i$ and $j$ index the participants and reported siblings, respectively. #OB, #OS, #Older, #All, and #Other refer to the number of older brothers, the number of older sisters, the number of older siblings, the number of all siblings and the number of other siblings (*i.e.*, siblings who are not older brothers), respectively. Sexual orientation is indicated by binary variable Hom with values 0 = heterosexual and 1 = homosexual. OB$ij$, OS$ij$ and Older$ij$ are binary variables with values 0 and 1, indicating the absence and presence of the event that the $j$th sibling is an older brother, an older sister, or an older sibling of the participant $i$ who reported him/her. *MROB*, *MROS*, *MPOB* and *MPOS* refer to the modified ratio of older brothers, the modified ratio of older sisters, the modified proportion of older brothers and the modified proportion of older sisters.

Table 3. These results describe the consequences of employing the models in Table 3 together with the NHST decision rule about the theoretical estimand θ.

Model 1 performed as described by *Blanchard (2014)*. Given a difference in the mean number of all siblings (*i.e.*, the μ-unequal condition), a positive effect of older brothers on homosexual orientation (θ = 0.33) would have frequently been misidentified as a negative or no effect of older brothers, as indicated by the negative values of the estimated regression coefficients ($\hat{\beta}_{\#OB}$) for the variable #OB.

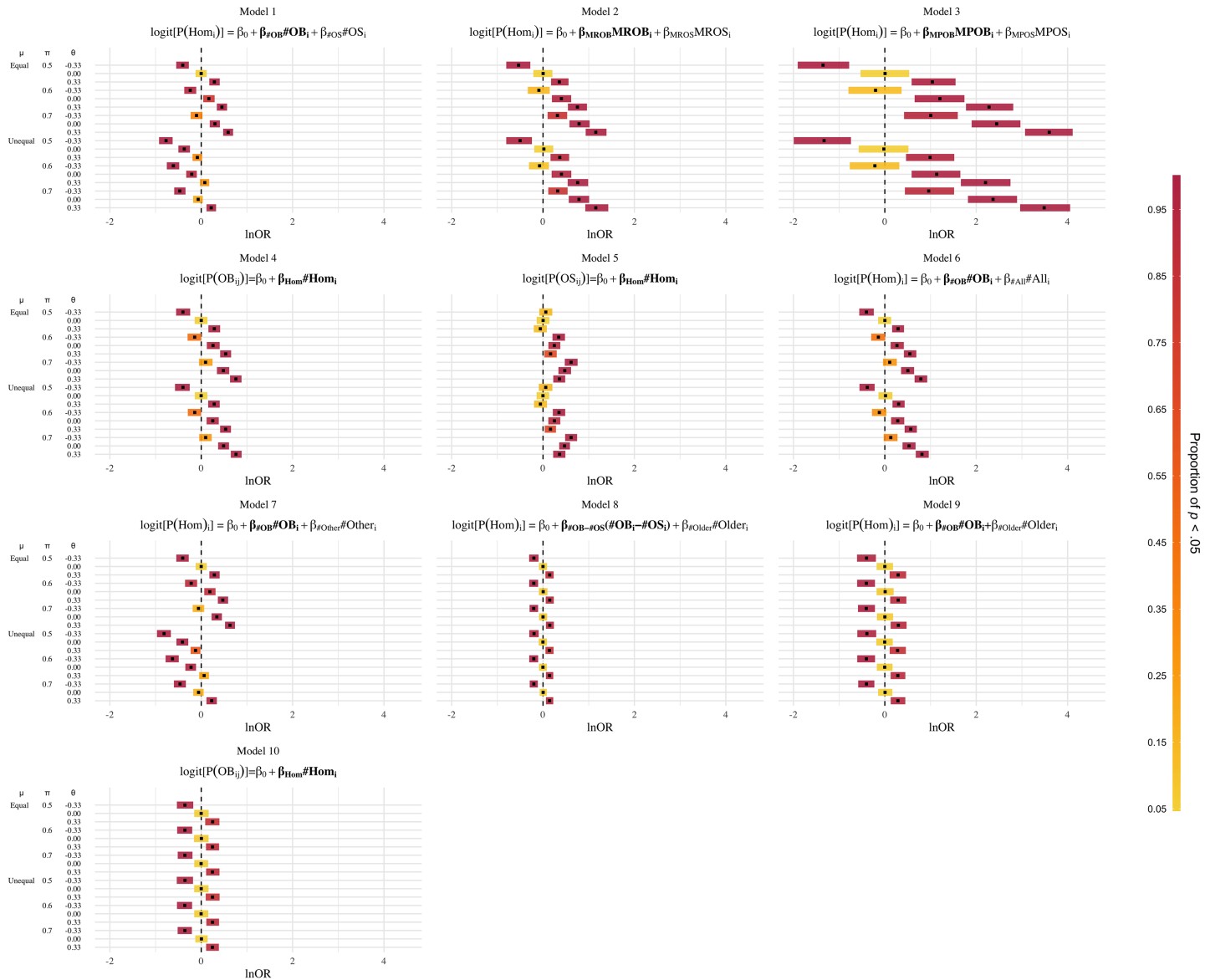

**Figure 2 Results of the simulation study for Models 1 to 10.** Error plots for the estimates of the regression coefficient for the predictor variables of interest (boldfaced) by model and conditions. Predictor variables in boldface were HOM, Homosexual (1 = yes, 0 = no); *MROB*, number of older brothers; #OS, number of older sisters; #OLDER, number of older siblings; #OTHER, number of other siblings (not older brothers); *MROB*, modified ratio of older brothers; *MROB*, modified ratio of older sisters; *MPOB*, modified proportion of older brothers; *MPOS*, modified proportion of older sisters; OB, older brother (1 = yes, 0 = no); OS, older brothers (1 = yes, 0 = no). Note that in Models 4, 5, and 10 the siblings of individuals rather than the individuals themselves are regarded as the observations and that in model 10 (as opposed to Model 4) the sampling space is restricted to older siblings (as opposed to all siblings). Black squares depict average estimates of the effect of interest in ln*OR* units over 1,000 replicates for each combination of μ, π, and θ (vertical axis). The column μ indicates whether in the homosexual and heterosexual groups, the mean number (or rate) of all siblings was equal (μ = 2.45 in both samples) or unequal (μ = 2.19 for the homosexual group and μ = 3.31 for the heterosexual group). The column π indicates the average proportion of older siblings in the homosexual group. In the heterosexual group this proportion is always equal to 0.5. The column θ indicates whether a positive, a negative, or no increase in the odds of homosexual orientation per older brother was present. θ = 0.33 denotes a 33% increase in these odds, θ = −0.33 denotes a 33% decrease and θ = 0 denotes no association. The width of a given rectangle corresponds to the interval between the 0.025 and the 0.975 quantile of the distribution of the 1,000 estimated regression coefficients. The color of the rectangle indicates the proportion out of the 1,000 replications that returned $p < 0.05$ (two-tailed).

[7] As shown in Supplement S5, Subsection 4, when an older-brother effect was present (*i.e.*, θ = 0.33), the regression coefficients for the *MROB* and *MPOB* were consistently greater in magnitude than the corresponding regression coefficients for the *MROS* and *MPOS*. This suggests that interpreting the two conjointly (for instance, by considering their difference) may yield decisions with a false-positive rate of 5%. This, however, just demonstrates the necessity of adjusting for the number of older siblings, not the number of other (or all) siblings.

The plots for Models 2 and 3 show that interpreting the coefficient estimates of the *MROB* or the *MPOB* in a logistic regression model (see Table 3) as corresponding to the specific effect of older brothers on homosexual orientation would frequently lead to false conclusions about the presence of such an older-brother effect. Furthermore, in following this line of inferring from these regression coefficients the theoretical estimand, researchers would quite often misidentify a negative older-brother effect (*i.e.*, the odds of homosexual orientation decreasing as a function of the number of older brothers due to θ = −0.33) as a positive older-brother effect. The magnitudes of the *MROB* and *MPOB* increased, as the difference in the proportion of older siblings increased as well.[7]

Similar considerations hold for the *OBOR* (Model 4) and the *OSOR* (Model 5). Use of the *OBOR* (in combination with a test of $H_0$: *OBOR* = 1) would frequently have led to the declaration of an older-brother effect (*i.e.*, θ ≠ 0), when in fact a more unspecific greater proportion of older siblings (π > 0.5) in homosexual men drove the increase in the *OBOR*. In some conditions, even negative older-brother effects (*i.e.*, θ = −0.33) would have been declared as positive older-brother effects. Another noticeable pattern in Models 4 and 5 is that under equal proportions of older siblings in both groups (*i.e.*, π = 0.5), the mean *OSOR* went in the opposite direction of the *OBOR* slightly more often than would be expected. This is interesting, as *Blanchard et al. (2021)* reported a negative correlation between the *OBOR* and the *OSOR*, but did not further elaborate on it, as it was not statistically significant. Here, we observed that the *OBOR* and *OSOR* were negatively associated under conditions of equal proportions of older siblings between groups (which once again demonstrates the necessity of adjusting for the number of older siblings).

The use of models, which adjust for #All (Model 6) and #Other (Model 7), would have also resulted in the false detection of an older-brother effect, way beyond the 5% rate. With respect to the qualitative assessment of the older-brother effect, Model 6 (*i.e.*, adjusting for #All) behaved almost exactly as the *MROB* and *MPOB* in Models 2 and 3. Adjusting for #Other (Model 7) would have frequently led to the conclusion that a negative older-brother effect is a positive older-brother effect under conditions of equal mean numbers of siblings and vice versa under conditions of unequal mean numbers of siblings.

The results of Models 8 through 10 are straightforward. The corresponding plots in Fig. 2 reveal that employing these models would have led to correct decisions about both the presence and the direction of an older-brother effect (*i.e.*, about θ) below the claimed false positive rate of 5%. Most importantly, these models contained meaningful (*i.e.*, interpretable) regression coefficients, which remained unaffected across all combinations of μ and π.

## Summary of Parts I and II

The results of Parts I and II demonstrate that currently recommended effect sizes, variable transformations, and their implementation in statistical models in the FBOE literature neither are needed nor adequate for quantifying and distinguishing the specific association between the number of older brothers and homosexual orientation (*i.e.*, the theoretical estimand) from a more general association of older siblings and sexual orientation.

Furthermore, models adjusting for #Other or #All (Models 6 and 7) can be regarded as misspecifications and thus are prone to lead to false or biased conclusions.

We also demonstrated the necessity of adjusting for the number of older siblings, as group differences in the proportion of older siblings are bound to lead to the false decision that solely the number of older brothers is associated with sexual orientation when using the *OBOR*, *MROB*, or *MPOB*. It is quite plausible to suppose such group differences, since the difference in the median proportions of older siblings across the 45 homosexual and heterosexual samples in *Blanchard (2018a*, *2018b)*, and *Blanchard et al. (2021)* amounted to a difference of 10 percentage points. We note that such differences in the number of older siblings are not independent of differences in the number of older brothers between groups, and it would therefore be too simplistic to conclude that such differences solely are due to the associations between the number of older sisters and homosexual orientation and the number of older brothers and homosexual orientation being of equal magnitude.

The *OBOR*, *MROB*, and *MPOB* fail to quantify the very thing, which they are advertised as quantifying. These indices and their recommended usage are not rigorous enough in exposing the proclaimed specific association between the number of older brothers and sexual orientation to conditions under which observations could indicate the absence of the specificity of such an association.

Given that the most extensive meta-analyses to date (totaling 45 samples of homo- and heterosexual males; *Blanchard, 2018a*, *2018b*; *Blanchard et al., 2021*) all employed the *OBOR*, it follows that these meta-analyses actually provided no conclusive evidence for the key observation of homosexual men having more older brothers relative to older sisters than heterosexual men. Our findings, of course, do not preclude the possibility that there may nevertheless be a meaningful stronger association between the number of older brothers and sexual orientation than between the number of older sisters and sexual orientation in men as required by the FBOE. This is one of the reasons for why in Part III we provide a set of new meta-analyses, using more adequate effect-size metrics, along with an advanced meta-analytic framework.

## PART III: MULTIVERSE (SPECIFICATION-CURVE) META-ANALYSIS OF THE LITERATURE

The *OBOR* is inappropriate for quantifying the theoretical estimand of the FBOE, implying that most of the meta-analytic evidence for the key observation of the FBOE (homosexual men have more older brothers relative to older sisters than heterosexual men) is not well founded. Thus, there is a demand to re-analyse the three meta-analyses which employed the *OBOR* (*Blanchard, 2018a*, *2018b*; *Blanchard et al., 2021*), but using a more appropriate effect-size metric. Moreover, the very existence of seven non-cumulative meta-analyses–each different with respect to the subset of available samples included (see Table 1) and the method used for amalgamating the evidence from these samples–demonstrates the issue of researcher degrees of freedom (*Simmons, Nelson & Simonsohn, 2011*) in specifying a single meta-analysis (see Supplement S4 for a discussion of researcher degrees of freedom in primary analyses on the FBOE). In Part III, we therefore set out to address this issue of researcher degrees of freedom in meta-analyses by
extending our re-analyses of *Blanchard (2018a*, *2018b)* and *Blanchard et al. (2021)* to include alternatively specified meta-analyses over the same subsets of samples.[8] Furthermore, and for the first time, we report the results of meta-analyses encompassing all extant male and female samples, thereby comparing the magnitudes of the association between older brothers and homosexual orientation between men and women. This difference in magnitude is crucial to the claim that the FBOE is specific to men. Finally, we offer a glimpse at the "garden of forking paths" (*Gelman & Loken, 2013*) of conceivable alternative meta-analyses for both male and female samples using the framework of specification-curve and multiverse meta-analysis (*Pietschnig et al., 2022*; *Simonsohn, Simmons & Nelson, 2020*; *Steegen et al., 2016*; *Voracek, Kossmeier & Tran, 2019*). To this end, we identified several necessary steps in specifying a single meta-analysis, where researcher degrees of freedom may possibly affect the outcome of any particular meta-analysis.

## Methods

### *Literature search and description of available samples*

We ascertained that the fourth, fifth, and seventh of the meta-analyses (*Blanchard, 2018a*, *2018b*; *Blanchard et al., 2021*) comprised 45 unique, non-overlapping samples of homosexual and heterosexual male participants originating from 35 studies. This count deviates from the count of 44 samples reported by *Blanchard (2018a*, *2018b)* and *Blanchard et al. (2021)* and can be traced back to *Blanchard (2018a)*, who merged the two unrelated samples in *Blanchard et al. (1996)* into a single sample. We could not see how this step could be precisely justified, and thus treated these two independent samples in *Blanchard et al. (1996)* as separate.

We were able to retrieve all but one of the publications of the primary studies included in the three focal meta-analyses (*Blanchard, 2018a*, *2018b*; *Blanchard et al., 2021*). The study by Krupp was cited as an unpublished manuscript in the seventh meta-analysis (*Blanchard et al., 2021*). The mean and/or total numbers of older brothers and older sisters and the corresponding standard deviation were extracted, if these were reported or could be determined through reported statistics. Twenty-one of these primary publications did not contain enough information to deduce the mean number of older brothers and older sisters or the corresponding standard deviations of these. Thus, we drew on the summary table in the Appendix of *Blanchard (2018a)* and Table 1 of *Blanchard et al. (2021)* to obtain the mean number of older brothers and older sisters for the following study samples: *Blanchard & Sheridan (1992)*, *Blanchard & Zucker (1994)*, *Zucker & Blanchard (1994)*, *Blanchard et al. (1995)*, *Blanchard & Bogaert (1996a*, *1996b)*, *Bogaert et al. (1997)*, *Blanchard & Bogaert (1998*; all three samples), *Ellis & Blanchard (2001)* and *Blanchard et al. (2006)*; samples named "Bogaert other", "Bogaert non-biological families", and "Blanchard"), *Blanchard et al. (2012*; all three samples) and *Krupp (2014*, as cited in *Blanchard et al., 2021*; both samples). The mean numbers of older brothers and older sisters of three additional samples were obtained from Table 1 in *Blanchard (2018b)*. These samples were originally referenced in *Bogaert (2010)* and *Zietsch et al. (2012*; two samples:

Table 4 Means and standard deviations of older brothers and sisters for the full set of male samples.

| Sample | Sexual orientation | N | Mean number of older brothers (SD) | Mean number of older sisters (SD) | 1 | 2 | 3 | 4 | 5 | 6 | 7 |
|---|---|---|---|---|---|---|---|---|---|---|---|
| Blanchard & Sheridan (1992) | Homo | 193 | 1.04 (-) | 0.82 (-) | X | X |  | X |  |  |  |
|  | Hetero | 273 | 0.49 (-) | 0.48 (-) |  |  |  |  |  |  |  |
| Blanchard & Zucker (1994) | Homo | 569 | 0.50 (-) | 0.45 (-) | X | X |  | X |  |  | X |
|  | Hetero | 281 | 0.44 (-) | 0.36 (-) |  |  |  |  |  |  |  |
| Zucker & Blanchard (1994) | Homo | 98 | 0.45 (-) | 0.49 (-) | X | X |  | X |  |  | X |
|  | Hetero | 84 | 0.39 (-) | 0.35 (-) |  |  |  |  |  |  |  |
| Blanchard et al. (1995) | Homo | 156 | 0.63 (-) | 0.43 (-) | X | X |  | X |  |  |  |
|  | Hetero | 156 | 0.42 (-) | 0.39 (-) |  |  |  |  |  |  |  |
| Blanchard & Bogaert (1996a) | Homo | 799 | 0.70 (1.12) | 0.59 (0.97) | X | X |  | X |  |  | X |
|  | Hetero | 3,807 | 0.58 (0.98) | 0.54 (0.93) |  |  |  |  |  |  |  |
| Blanchard & Bogaert (1996b) | Homo | 302 | 0.71 (-) | 0.60 (-) | X | X |  | X |  |  | X |
|  | Hetero | 302 | 0.69 (-) | 0.68 (-) |  |  |  |  |  |  |  |
| Blanchard et al. (1996) Study 1 | Homo | 83 | 1.04 (1.19) | 0.73 (1.04) | X | X |  | X |  |  |  |
|  | Hetero | 58 | 0.76 (1.41) | 0.52 (0.75) |  |  |  |  |  |  |  |
| Blanchard et al. (1996) Study 2 | Homo | 21 | 0.81 (0.68) | 0.24 (0.44) | X | X |  | X |  |  |  |
|  | Hetero | 21 | 0.33 (0.58) | 0.05 (0.22) |  |  |  |  |  |  |  |
| Bogaert et al. (1997) | Homo | 68 | 0.76 (-) | 0.81 (-) |  | X |  | X |  |  | X |
|  | Hetero | 57 | 0.56 (-) | 0.70 (-) |  |  |  |  |  |  |  |
| Blanchard & Bogaert (1998) Offenders against adults | Homo | 157 | 0.82 (-) | 0.71 (-) | X | X |  | X |  | X | X |
|  | Hetero | 173 | 0.89 (-) | 0.91 (-) |  |  |  |  |  |  |  |
| Blanchard & Bogaert (1998) Offenders against children | Homo | 42 | 1.00 (-) | 0.95 (-) |  | X |  | X |  | X | X |
|  | Hetero | 143 | 1.08 (-) | 1.09 (-) |  |  |  |  |  |  |  |
| Blanchard & Bogaert (1998) Offenders against pubescents | Homo | 69 | 1.14 (-) | 1.13 (-) |  | X |  | X |  | X | X |
|  | Hetero | 127 | 1.17 (-) | 0.94 (-) |  |  |  |  |  |  |  |
| Blanchard et al. (1998) | Homo | 385 | 0.53 (0.90) | 0.43 (0.77) | X | X |  | X |  | X | X |
|  | Hetero | 225 | 0.32 (0.65) | 0.43 (0.85) |  |  |  |  |  |  |  |
| Blanchard et al. (2000) | Homo | 65 | 1.08 (-) | 0.78 (-) |  | X |  | X |  |  |  |
|  | Hetero | 152 | 0.76 (-) | 0.69 (-) |  |  |  |  |  |  |  |
| Green (2000) | Homo | 106 | 0.90 (1.11) | 0.79 (1.11) |  |  |  | X | X |  |  |
|  | Hetero | 135 | 0.58 (0.88) | 0.61 (0.95) |  |  |  |  |  |  |  |
| Ellis & Blanchard (2001) | Homo | 175 | 0.67 (-) | 0.49 (-) |  | X |  | X |  | X | X |
|  | Hetero | 971 | 0.51 (-) | 0.50 (-) |  |  |  |  |  |  |  |
| Rahman, Wilson & Abrahams (2004) | Homo | 60 | 0.58 (0.74) | 0.85 (1.32) |  |  |  | X |  |  |  |
|  | Hetero | 60 | 0.48 (0.79) | 0.53 (0.85) |  |  |  |  |  |  |  |
| Bogaert (2005) | Homo | 79 | 0.91 (-) | 0.63 (-) |  |  |  |  |  | X |  |
|  | Hetero | 2,721 | 0.69 (-) | 0.65 (-) |  |  |  |  |  |  |  |
| King et al. (2005) | Homo | 301 | 0.66 (0.88) | 0.59 (0.88) |  |  |  | X | X |  | X |
|  | Hetero | 404 | 0.47 (0.71) | 0.43 (0.73) |  |  |  |  |  |  |  |
| Rahman (2005) | Homo | 30 | 1.09 (0.90) | 0.58 (0.99) |  |  |  | X |  |  |  |
|  | Hetero | 31 | 0.40 (0.72) | 0.53 (0.97) |  |  |  |  |  |  |  |

(Continued)

| Sample | Sexual orientation | N | Mean number of older brothers (SD) | Mean number of older sisters (SD) | 1 | 2 | 3 | 4 | 5 | 6 | 7 |
|---|---|---|---|---|---|---|---|---|---|---|---|
| *Blanchard et al. (2006)* "Blanchard" subsample | Homo | 92 | 1.07 (-) | 0.86 (-) | | | | X | | | |
| | Hetero | 672 | 0.83 (-) | 0.82 (-) | | | | | | | |
| *Blanchard et al. (2006)* "Bogaert other" subsample | Homo | 280 | 0.50 (-) | 0.41 (-) | | | | X | | X | X |
| | Hetero | 222 | 0.38 (-) | 0.41 (-) | | | | | | | |
| *Blanchard et al. (2006)* "Bogaert non-biological family" subsample | Homo | 267 | 0.82 (-) | 0.65 (-) | | | | X | | X | X |
| | Hetero | 148 | 0.51 (-) | 0.45 (-) | | | | | | | |
| *Frisch & Hviid (2006)* | Homo | 1,890 | 0.37 (0.62) | 0.30 (0.59) | | | | | X | | |
| | Hetero | 429,181 | 0.34 (0.61) | 0.27 (0.55) | | | | | | | |
| *Blanchard & Lippa (2007)* | Homo | 8,279 | 0.53 (-) | 0.51 (-) | | | | | | | |
| | Hetero | 79,519 | 0.45 (-) | 0.44 (-) | | | | | | | |
| *Vasey & VanderLaan (2007)* | Homo | 83 | 2.27 (1.84) | 2.08 (1.71) | | | | X | | | |
| | Hetero | 114 | 1.23 (1.37) | 1.25 (1.20) | | | | | | | |
| *Zucker et al. (2007)* | Homo | 43 | 0.26 (0.49) | 0.84 (1.13) | | | | | | | |
| | Hetero | 49 | 0.43 (0.65) | 1.00 (1.29) | | | | | | | |
| *Rahman et al. (2008)* Non-white subsample | Homo | 20 | 0.60 (0.82) | 0.45 (0.60) | | | | X | | | |
| | Hetero | 53 | 0.81 (1.05) | 0.84 (1.13) | | | | | | | |
| *Rahman et al. (2008)* White subsample | Homo | 127 | 0.55 (0.82) | 0.49 (0.79) | | | | X | | | |
| | Hetero | 102 | 0.52 (0.87) | 0.55 (1.11) | | | | | | | |
| *Camperio-Ciani, Iemmola & Blecher (2009)* | Homo | 65 | 0.63 (-) | 0.38 (-) | | | | | | | |
| | Hetero | 88 | 0.28 (-) | 0.39 (-) | | | | | | | |
| *Rahman, Clarke & Morera (2009)* | Homo | 100 | 0.89 (0.90) | 0.63 (0.79) | | | | X | | | |
| | Hetero | 100 | 0.63 (0.86) | 0.51 (0.82) | | | | | | | |
| *Bogaert (2010)* | Homo | 132 | 0.68 (1.07) | 0.67 (1.12) | | | | | X | | |
| | Hetero | 5,472 | 0.58 (0.81) | 0.57 (0.81) | | | | | | | |
| *Schwartz et al. (2010)* | Homo | 677 | 0.80 (1.25) | 0.66 (1.09) | | | | X | X | | X |
| | Hetero | 873 | 0.56 (0.85) | 0.51 (0.82) | | | | | | | |
| *Gómez-Gil et al. (2011)* | Homo | 287 | 1.01 (1.27) | 0.85 (1.13) | | | | X | X | | |
| | Hetero | 38 | 0.42 (0.72) | 0.63 (0.85) | | | | | | | |
| *Kangassalo, Pölkki & Rantala (2011)* | Homo | 63 | 0.70 (-) | 1.03 (-) | | | | X | | | |
| | Hetero | 273 | 0.41 (-) | 0.45 (-) | | | | | | | |
| *VanderLaan & Vasey (2011)* Replication Sample | Homo | 133 | 1.92 (1.68) | 1.70 (1.59) | | | | | X | X | |
| | Hetero | 208 | 0.86 (1.15) | 1.02 (1.15) | | | | | | | |
| *Blanchard et al. (2012)* Hebephile | Homo | 79 | 0.94 (-) | 0.78 (-) | | | | | | | X |
| | Hetero | 704 | 0.86 (-) | 0.86 (-) | | | | | | | |
| *Blanchard et al. (2012)* Pedophile | Homo | 101 | 0.97 (-) | 0.80 (-) | | | | | | | X |
| | Hetero | 141 | 0.66 (-) | 0.78 (-) | | | | | | | |
| *Blanchard et al. (2012)* Teleiophile | Homo | 91 | 1.09 (-) | 0.85 (-) | | | | | | | X |
| | Hetero | 998 | 0.82 (-) | 0.77 (-) | | | | | | | |
| *Schagen et al. (2012)* | Homo | 94 | 0.51 (0.70) | 0.17 (0.38) | | | | X | X | | |
| | Hetero | 875 | 0.34 (0.62) | 0.32 (0.58) | | | | | | | |

| Table 4 (continued) | | | | | | | | | | | |
| --- | --- | --- | --- | --- | --- | --- | --- | --- | --- | --- | --- |
| **Sample** | **Sexual orientation** | *N* | **Mean number of older brothers** *(SD)* | **Mean number of older sisters** *(SD)* | 1 | 2 | 3 | 4 | 5 | 6 | 7 |
| *Zietsch et al. (2012)* "Female co-twin" subsample | Homo | 24 | 0.96 (-) | 0.92 (-) | | | | | X | | |
| | Hetero | 622 | 0.80 (-) | 0.86 (-) | | | | | | | |
| *Zietsch et al. (2012)* "Male co-twin" subsample | Homo | 36 | 0.83 (-) | 0.61 (-) | | | | | X | | |
| | Hetero | 743 | 0.82 (-) | 0.70 (-) | | | | | | | |
| *Krupp (2014)* Hebephile | Homo | 31 | 0.42 (-) | 0.35 (-) | | | | | | | X |
| | Hetero | 82 | 0.26 (-) | 0.44 (-) | | | | | | | |
| *Krupp (2014)* Pedophile | Homo | 27 | 0.59 (-) | 0.33 (-) | | | | | | | X |
| | Hetero | 28 | 0.21 (-) | 0.32 (-) | | | | | | | |
| *VanderLaan et al. (2014)* | Homo | 346 | 0.42 (0.67) | 0.31 (0.65) | | | | X | | X | |
| | Hetero | 210 | 0.35 (0.66) | 0.31 (0.61) | | | | | | | |
| *Xu & Zheng (2014)* | Homo | 215 | 0.12 (0.40) | 0.25 (0.59) | | | | | | | |
| | Hetero | 160 | 0.15 (0.42) | 0.31 (0.81) | | | | | | | |
| *Yule, Brotto & Gorzalka (2014)* | Homo | 64 | 0.55 (-) | 0.78 (-) | | | | | | | |
| | Hetero | 190 | 0.40 (-) | 0.35 (-) | | | | | | | |
| *Bozkurt, Bozkurt & Sonmez (2015)* | Homo | 60 | 1.32 (1.16) | 1.13 (1.32) | | | X | X | | | |
| | Hetero | 61 | 0.44 (0.67) | 1.02 (1.28) | | | | | | | |
| *Currin, Gibson & Hubach (2015)* | Homo | 118 | 0.52 (0.80) | 0.48 (0.82) | | | | X | | | X |
| | Hetero | 500 | 0.57 (0.88) | 0.49 (0.79) | | | | | | | |
| *Kishida & Rahman (2015)* | Homo | 905 | 0.63 (0.91) | 0.59 (0.89) | | | | X | | | X |
| | Hetero | 999 | 0.56 (0.88) | 0.53 (0.89) | | | | | | | |
| *Austin (2017)* | Homo | 42 | 1.02 (1.24) | 0.76 (1.03) | | | | | | | |
| | Hetero | 40 | 0.63 (0.74) | 0.48 (0.68) | | | | | | | |
| *Semenyna, VanderLaan & Vasey (2017)* | Homo | 230 | 1.86 (1.75) | 1.85 (1.71) | | | | | | | |
| | Hetero | 228 | 1.39 (1.39) | 1.19 (1.25) | | | | | | | |
| *Vanderlaan et al. (2017)* | Homo | 118 | 0.92 (1.21) | 0.86 (-) | | | | X | | | |
| | Hetero | 143 | 0.74 (0.99) | 0.55 (-) | | | | | | | |
| *Xu & Zheng (2017)* | Homo | 481 | 0.25 (0.58) | 0.47 (0.91) | | | | | | | |
| | Hetero | 392 | 0.28 (0.57) | 0.42 (0.77) | | | | | | | |
| *Swift-Gallant et al. (2018)* | Homo | 243 | 0.58 (0.95) | 0.50 (0.76) | | | | | | | |
| | Hetero | 91 | 0.55 (0.83) | 0.43 (0.82) | | | | | | | |
| *Nila et al. (2019)* | Homo | 116 | 957.00 (1.23) | 1.09 (1.24) | | | | | | | X |
| | Hetero | 62 | 677.00 (1.50) | 935.00 (1.72) | | | | | | | |
| *Tran, Kossmeier & Voracek (2019)* | Homo | 53 | 0.43 (0.67) | 0.43 (0.72) | | | | | | | |
| | Hetero | 1,730 | 0.40 (0.69) | 0.38 (685.00) | | | | | | | |
| *Xu, Norton & Rahman (2019)* | Homo | 28 | 0.50 (0.64) | 0.22 (0.42) | | | | | | | |
| | Hetero | 2,138 | 0.38 (0.62) | 0.36 (0.61) | | | | | | | |
| *Apostolou (2020)* | Homo | 183 | 0.50 (-) | 0.39 (-) | | | | | | | X |
| | Hetero | 593 | 0.35 (-) | 0.31 (-) | | | | | | | |
| *Gómez Jiménez, Semenyna & Vasey (2020)* | Homo | 244 | 1.16 (1.32) | 1.14 (1.36) | | | | | | | X |
| | Hetero | 194 | 0.75 (0.98) | 0.70 (1.00) | | | | | | | |

*(Continued)*

| Table 4 (continued) | | | | | | | |
| --- | --- | --- | --- | --- | --- |
| Sample | Sexual orientation | N | Mean number of older brothers (SD) | Mean number of older sisters (SD) | 1 2 3 4 5 6 7 |
| *Khorashad et al. (2020)* | Homo | 92 | 2.32 (1.75) | 0.97 (1.15) |       X   X |
| | Hetero | 72 | 1.10 (1.12) | 0.99 (1.23) | |
| *Skorska & Bogaert (2020)* Subsample 1 | Homo | 174 | 0.33 (-) | 0.17 (-) |          X |
| | Hetero | 6,562 | 0.26 (-) | 0.23 (-) | |
| *Skorska & Bogaert (2020)* Subsample 2 | Homo | 167 | 0.30 (-) | 0.17 (-) | |
| | Hetero | 6,228 | 0.26 (-) | 0.22 (-) | |
| *Skorska et al. (2020)* | Homo | 205 | 0.28 (0.50) | 0.30 (0.59) | |
| | Hetero | 286 | 0.26 (0.51) | 0.36 (0.68) | |

**Note:**

Column "Group" denotes sexual orientation (Homo = Homosexual, Hetero = Heterosexual) of sample. "-" denotes non-reported missing values. Cell entries of x *vs* blank space indicate whether a given sample was included *vs* not included in each of the seven previous meta-analyses. The columns labelled "1" through "7" correspond to the meta-analyses by *Jones & Blanchard (1998)* (1), *Blanchard (2004)* (2), *Blanchard & VanderLaan (2015)* (3), *Blanchard (2018a)* (4), *Blanchard (2018b)* (5), *Blanchard et al. (2020)* (6), *Blanchard et al. (2021)* (7).

"female co-twins" and "male co-twins"). The mean number of older brothers and older sisters and their standard deviations for the homosexual and heterosexual male samples in *Frisch & Hviid (2006)* were obtained from *Blanchard & VanderLaan (2015)*. We retrieved 19 additional samples through database searches, cited reference searches, and citation alerts (all of these based on Google Scholar). One of these samples was retrieved from *Tran, Kossmeier & Voracek (2019)* and had previously been analyzed in a different context (the data are provided online, https://osf.io/3wnhu/).

The sole inclusion criterion used to obtain additional samples (Table 4) was that the number of older brothers (average or total) and older sisters among homosexual and heterosexual male participants had to be available, as the number of younger brothers is irrelevant to the estimation of the specific association between older brothers and sexual orientation. Furthermore, we also extracted the mean or total numbers of younger brothers and sisters, if these were reported, to compare the magnitude of the *OBOR* to the magnitude of the *OR* for older brothers *vs* older sisters (Eq. (6) and Model 10 in Table 3).

Seventeen samples also reported sibling data on homosexual and heterosexual women; these data were also extracted (see Table 5). In order to obtain the mean number of each sibling type for the female samples in *Frisch & Hviid (2006)*, we used the same approach as *Blanchard & VanderLaan (2015)* for obtaining the mean number of each sibling type for the male participants: Table 4 of *Frisch & Hviid (2006)* contained the number of participants who had 0, 1, 2, or 3 or more siblings in each category (older brothers, older sisters, younger brothers, and younger sisters). We multiplied the numbers of participants in each category by the respective numbers of older brothers, older sisters, younger brothers, and younger sisters, and then divided the resulting sums by the total numbers of participants. Theoretically, this may lead to a negative bias in the estimate of the mean number of each sibling type. However, we contend that this bias is practically negligible, since having three or more siblings of the same sex and sibship position may be regarded as

**Table 5 Means and standard deviations of older brothers and sisters for the full set of female samples.**

| Sample | Sexual orientation | N | Mean number of older brothers (SD) | Mean number of older sisters (SD) |
|---|---|---|---|---|
| *Blanchard et al. (1998)* | Homo | 162 | 0.41 (0.75) | 0.30 (0.69) |
| | Hetero | 192 | 0.43 (0.83) | 0.41 (0.81) |
| *Green (2000)* | Homo | 73 | 0.58 (1.05) | 0.74 (1.09) |
| | Hetero | 7 | 0.71 (0.95) | 0.14 (0.38) |
| *Rahman, Wilson & Abrahams (2004)* | Homo | 60 | 0.63 (0.97) | 0.68 (1.09) |
| | Hetero | 60 | 0.65 (1.05) | 0.45 (1.04) |
| *Rahman (2005)* | Homo | 30 | 0.65 (0.89) | 0.58 (0.90) |
| | Hetero | 29 | 0.63 (0.76) | 1.13 (1.59) |
| *Frisch & Hviid (2006)* | Homo | 1,537 | 0.40 (-) | 0.28 (-) |
| | Hetero | 453,121 | 0.38 (-) | 0.31 (-) |
| *Blanchard & Lippa (2007)* | Homo | 7,013 | 0.43 (-) | 0.37 (-) |
| | Hetero | 64,968 | 0.48 (-) | 0.43 (-) |
| *Bogaert (2010)* | Homo | 75 | 0.55 (0.83) | 0.71 (0.99) |
| | Hetero | 5,335 | 0.59 (0.79) | 0.56 (0.80) |
| *Gómez-Gil et al. (2011)* | Homo | 154 | 0.82 (1.02) | 0.79 (1.14) |
| | Hetero | 6 | 1.50 (1.98) | 1.50 (2.07) |
| *Schagen et al. (2012)* | Homo | 95 | 0.25 (-) | 0.23 (-) |
| | Hetero | 914 | 0.36 (-) | 0.33 (-) |
| *Xu & Zheng (2014)* | Homo | 42 | 0.10 (0.07) | 0.07 (-) |
| | Hetero | 255 | 0.15 (0.45) | 0.19 (0.47) |
| *Xu & Zheng (2017)* | Homo | 431 | 0.16 (0.45) | 0.21 (0.70) |
| | Hetero | 554 | 0.22 (0.57) | 0.23 (0.51) |
| *Tran, Kossmeier & Voracek (2019)* | Homo | 28 | 0.32 (0.62) | 0.46 (0.84) |
| | Hetero | 1,947 | 0.36 (0.66) | 0.38 (0.67) |
| *Xu, Norton & Rahman (2019)* | Homo | 16 | 0.63 (0.81) | 0.31 (0.60) |
| | Hetero | 2,391 | 0.38 (0.63) | 0.37 (0.60) |
| *Apostolou (2020)* | Homo | 76 | 0.41 (-) | 0.24 (-) |
| | Hetero | 366 | 0.36 (-) | 0.34 (-) |
| *Khorashad et al. (2020)* | Homo | 107 | 1.04 (1.26) | 1.29 (1.36) |
| | Hetero | 78 | 1.15 (1.27) | 1.05 (1.27) |
| *Skorska & Bogaert (2020)* Subsample 1 | Homo | 143 | 0.20 (-) | 0.20 (-) |
| | Hetero | 7,727 | 0.22 (-) | 0.21 (-) |
| *Skorska & Bogaert (2020)* Subsample 2 | Homo | 136 | 0.21 (-) | 0.20 (-) |
| | Hetero | 7,297 | 0.23 (-) | 0.21 (-) |

**Note:**
Homo, Homosexual; Hetero, Heterosexual; "-", denotes non-reported missing values. None of these female samples has been included in any of the seven prior related meta-analyses.

an extremely rare event (for instance, only nine out of the 1,573 homosexual women in *Frisch & Hviid (2006)*, reported having three or more older brothers).

While the standard deviation was not needed for the effect-size metric we employed for meta-analysis (*i.e.*, the ln*OR* in Eq. (6); see next subsection), we nevertheless decided to report it in Tables 5 and 6 below, to highlight the pattern of non-reporting of basic

**Table 6 Specification matrix indicating the factor-level of each of the 64 male samples for each of the 10 which factors.**

| Sample | In previous MA | Lab | Archives | Feminine | GDY | Clinical | Classification | Probability | Stopping | Teleio- *vs* pedo-/hebephiles |
|---|---|---|---|---|---|---|---|---|---|---|
| *Blanchard & Sheridan (1992)* | X | X | | X | X | X | Indirect | | | |
| *Blanchard & Zucker (1994)* | X | X | | | | | Various | | | |
| *Zucker & Blanchard (1994)* | X | X | X | | | | Various | | | |
| *Blanchard et al. (1995)* | X | X | | X | X | X | Indirect | | | |
| *Blanchard & Bogaert (1996a)* | X | X | X | | | | Indirect | | | |
| *Blanchard & Bogaert (1996b)* | X | X | | | | | Single item | | | |
| *Blanchard et al. (1996)* Study 1 | X | X | X | X | X | X | Single item | | | |
| *Blanchard et al. (1996)* Study 2 | X | X | X | X | X | X | None | | | |
| *Bogaert et al. (1997)* | X | X | | | | X | Various | | | X |
| *Blanchard & Bogaert (1998)* Offenders against adults | X | X | X | | | X | Indirect | | | |
| *Blanchard & Bogaert (1998)* Offenders against children | X | X | X | | | X | Indirect | | | X |
| *Blanchard & Bogaert (1998)* Offenders against pubescents | X | X | X | | | X | Indirect | | | X |
| *Blanchard et al. (1998)* | X | X | | | | | Various | | | |
| *Blanchard et al. (2000)* | X | X | X | | | X | Various | | | X |
| *Green (2000)* | X | | X | X | X | X | Various | | | |
| *Ellis & Blanchard (2001)* | X | X | | | | | Single item | | | |
| *Rahman, Wilson & Abrahams, 2004* | X | | | | | | Various | | | |
| *Bogaert (2005)* | X | X | X | | | | Various | X | | |
| *King et al. (2005)* | X | | X | | | X | Various | | | |
| *Rahman (2005)* | X | | | | | | Various | | | |
| *Blanchard et al. (2006)* "Blanchard" subsample | X | X | | | | X | Various | | | |
| *Blanchard et al. (2006)* "Bogaert other" subsample | X | X | | | | | Various | | | |
| *Blanchard et al. (2006)* "Bogaert non-biological family" subsample | X | X | | | | | Various | | | |
| *Frisch & Hviid (2006)* | X | | | X | | | Single item | X | | |
| *Blanchard & Lippa (2007)* | | X | X | | | | Various | | | |
| *Vasey & VanderLaan (2007)* | X | X | | X | | | Single item | | | |
| *Zucker et al. (2007)* | | X | | | X | X | Single item | | | |
| *Rahman et al. (2008)* Non-white subsample | X | | X | | | | Various | | | |
| *Rahman et al. (2008)* White subsample | X | | X | | | | Various | | | |
| *Camperio-Ciani, Iemmola & Blecher (2009)* | | | | | | | Various | | | |
| *Rahman, Clarke & Morera (2009)* | X | | | | | | Various | | | |
| *Bogaert (2010)* | X | X | X | | | | Various | X | | |
| *Schwartz et al. (2010)* | X | | X | | | | Various | | | |
| *Gómez-Gil et al. (2011)* | X | | X | X | X | X | Single item | | | |
| *Kangassalo, Pölkki & Rantala (2011)* | X | | | | | | Various | | | |
| *VanderLaan & Vasey (2011)* Replication Sample | X | X | X | X | | | Various | | | |
| *Blanchard et al. (2012)* Hebephile | X | X | X | | | X | Indirect | | | X |
| *Blanchard et al. (2012)* Pedophile | X | X | X | | | X | Indirect | | | X |
| *Blanchard et al. (2012)* Teleiophile | X | X | X | | | X | Indirect | | | X |

| Sample | In previous MA | Lab | Archives | Feminine | GDY | Clinical | Classification | Probability | Stopping | Teleio- *vs* pedo-/hebephiles |
|---|---|---|---|---|---|---|---|---|---|---|
| *Schagen et al. (2012)* | X | X | X | X | X | X | None | | | |
| *Zietsch et al. (2012)* "Female co-twin" subsample | X | | | | | | Single item | X | | |
| *Zietsch et al. (2012)* "Male co-twin" subsample | X | | | | | | Single item | X | | |
| *Krupp (2014)* Hebephile | X | | | | | X | – | | | X |
| *Krupp (2014)* Pedophile | X | | | | | X | – | | | X |
| *VanderLaan et al. (2014)* | X | X | | X | X | X | Various | | | |
| *Xu & Zheng (2014)* | | | | | | | Single item | | | |
| *Yule, Brotto & Gorzalka (2014)* | | | X | | | | Single item | | | |
| *Bozkurt, Bozkurt & Sonmez, 2015* | X | | X | X | X | | Single item | | | |
| *Currin, Gibson & Hubach, 2015* | X | | | | | | Various | | | |
| *Kishida & Rahman (2015)* | X | | X | | | | Various | | | |
| *Austin (2017)* | | | | | | | Various | | | |
| *Semenyna, VanderLaan & Vasey (2017)* | | X | | | | | Various | | | |
| *Vanderlaan et al. (2017)* | X | X | | X | X | X | Indirect | | | |
| *Xu & Zheng (2017)* | | | | | | | Various | | | |
| *Swift-Gallant et al. (2018)* | | X | X | | | | Various | | | |
| *Nila et al. (2019)* | X | | | | | | Various | | | |
| *Tran, Kossmeier & Voracek, 2019* | | | X | | | | Single item | | | |
| *Xu, Norton & Rahman (2019)* | | | | | | | Various | X | | |
| *Apostolou (2020)* | X | | X | | | | Various | | | |
| *Gómez Jiménez, Semenyna & Vasey (2020)* | X | X | | | | | Various | | | |
| *Khorashad et al. (2020)* | X | X | | X | X | X | Indirect | | | |
| *Skorska & Bogaert (2020)* Subsample 1 | X | X | X | | | | Single item | X | | |
| *Skorska & Bogaert (2020)* Subsample 2 | | X | X | | | | Single item | X | | |
| *Skorska et al. (2020)* | | X | X | | | | Various | | | |

**Note:**
Studies are ordered by year of publication. "-", not reported. For the nine binary factors the table entries ("X" *vs* blank cell) correspond to: Sample was included in any of the seven previous meta-analyses *vs* not (for factor in Previous MA), sample was recruited in a clinical setting *vs* not (for factor clinical), sample was denoted "feminine" by *Blanchard (2018a)* *vs* not (for factor Feminine), sample contained participants with GDY diagnosis *vs* not (for factor GDY), sample was obtained using a random sampling procedure *vs* not (for factor Probability), sample originated from a study co-authored by Blanchard, Bogaert, Zucker or VanderLaan *vs* not (for factor Lab), sample was declard as consisting of "pedo- or hebephiles" *vs* "teleiophiles" by Blanchard, *Krupp (2014)* (for factor teleio- *vs* pedo-/hebephiles), a stopping rule was declared present *vs* not (for factor stopping), sample originated from a study publised in Archives of Sexual Behavior *vs* not (for factor archives). For the factor *Classification* four levels indicating how sexual orientation was assessed: Level "Various" indicates a mixture of methods or questionnaire summary scores; level "Indirect" indicates that behavioural measures or expert ratings were used; level "Single Item" indicates that only a single piece of information such as self-identified sexual orientation was used; level "none" indicates that sexual orientation was not assessed at all.

summary statistics in this research field. Of note, it was impossible to deduce the standard deviations for a total of 28 of the 64 samples in Table 4, due to insufficient reporting, although the respective analyses in these primary studies were carried out on the participant level and thus most likely required standard errors to obtain a test statistic.

Most of the available samples were subjected to a binary classification scheme with respect to participants' sexual orientation. In addition, some samples contained sibship data of participants, whose sexual orientation was classified as bisexual (*Apostolou, 2020*; *Camperio-Ciani, Iemmola & Blecher, 2009*; *Green, 2000*; *Skorska & Bogaert, 2020*; *Tran, Kossmeier & Voracek, 2019*) or asexual (*Green, 2000*; *Yule, Brotto & Gorzalka, 2014*), while

in other samples bisexual participants were simply merged with the homosexual group (*Blanchard et al., 2006*; *Blanchard & Bogaert, 1996a*; *Blanchard & Lippa, 2007*; *Bogaert, 2005*, *2010*; *Bogaert et al., 1997*; *Xu & Zheng, 2014*).

Since the FBOE specifically states that the number of older brothers should increase the odds of homosexual orientation (*e.g.*, *Blanchard, 2018b*), the bisexual and asexual groups were excluded from Tables 5 and 6, whenever possible.

*Homosexual* and *heterosexual* were not the only labels assigned to the two categories in the binary classification schemes used throughout the available samples. Other binary labels were *heterosexual vs. nonheterosexual* (*Kangassalo, Pölkki & Rantala, 2011*; *Xu & Zheng, 2017*; *Yule, Brotto & Gorzalka, 2014*; *Zietsch et al., 2012*), *homosexual vs. nonhomosexual* (*Blanchard et al., 1996*; *Blanchard & Sheridan, 1992*; *Gómez-Gil et al., 2011*; *Green, 2000*), and *androphilic vs. gynephilic* (*Semenyna, VanderLaan & Vasey, 2017*; *Skorska & Bogaert, 2020*; *Vanderlaan et al., 2017*; *VanderLaan & Vasey, 2011*; *Vasey & VanderLaan, 2007*). In addition, two studies did not assess sexual orientation at all, but rather compared the sibship compositions of male-assigned individuals diagnosed with gender dysphoria (GDY) to heterosexual controls (*Blanchard et al., 1996*; *Schagen et al., 2012*). *Blanchard (2018a)* did not provide a justification as to why these samples should be included in his meta-analysis,[9] but given that *Blanchard (2018a)* classified GDY participants as homosexual, it appears that in this case GDY was regarded as a proxy for homosexual orientation. Regardless of whether based on subject-matter insights or less so, such varying classification schemes constitute researcher degrees of freedom in the process of devising an analysis plan.

In line with the fourth, fifth, and seventh meta-analysis (*Blanchard, 2018a*, *2018b*; *Blanchard et al., 2021*), the groups from these primary samples were categorized as either homosexual or heterosexual in Tables 5 and 6. Groups labelled as *heterosexual*, *nonhomosexual*, or *gynephilic* were assigned to the heterosexual category, whereas groups labelled as *homosexual*, *nonheterosexual* and *androphilic* were assigned to the homosexual category. For the two samples without an explicit assessment of sexual orientation (*Blanchard et al., 1996*; *Schagen et al., 2012*), GDY participants were assigned to the homosexual group, whereas the controls to the heterosexual group (as in *Blanchard, 2018a*). Tables 5 and 6 display only the mean numbers and standard deviations of older brothers and older sisters. The corresponding statistics for younger brothers and younger sisters can be found in the data provided online (https://osf.io/3wnhu/).

## Data analysis

In choosing an effect size for meta-analysis, we followed the approach of *Blanchard et al. (2020)* in modelling reported siblings as independent observations (*Blanchard, 2018a*, *2018b*). Note that no further information on the reported siblings was available than the number of times each sibling type occurred in a sample.

We quantified the difference in the number of older brothers between the homosexual and heterosexual group through the natural logarithm of the odds ratio (ln*OR*) of older brothers *vs.* older sisters, as described in Eq. (6) above and Model 10 in Table 3. As shown in Parts I and II, this ln*OR* adjusts for the number (or the proportion) of older siblings and

[9] The use of proxy variables for classifying sexual orientation was in fact listed as one of the five exclusion criteria in *Blanchard (2018a)*.
remains unaffected by group differences in the total number of siblings. In order to compute this ln*OR*, the total numbers of older brothers and sisters of the homosexual and heterosexual participants are needed, instead of the mean numbers listed in Tables 5 and 6. These totals can be obtained by multiplying the Tables 5 and 6 mean values with the corresponding sample size (column *N*) and rounding to the nearest integer.[10] Alternatively, one could use the difference in the proportion of older brothers among older siblings between homosexual and heterosexual men and women as an effect size. Corresponding meta-analyses using the difference in proportions as an effect size can be found in Supplement S6 (Subsection 1).

[10] Although we recommend using the raw count data provided online, as rounding errors could affect the results.

We fitted random-effects models to five specific study sets comprising the 31 male samples in *Blanchard (2018a)*, the six male probability samples in *Blanchard (2018b)*, the 24 male samples in *Blanchard et al. (2021)*, the entire set of 64 male samples, and the 17 female samples (results of fixed-effect analyses are provided in Supplement 6, Subsection 2). For all random-effects models, the between-sample variance $\tau^2$ was estimated using the restricted maximum-likelihood (REML) estimator as implemented in the *R* package *metafor* (*Viechtbauer, 2010*).

To further assess the extent to which the effect might differ between men and women, we also computed a combined meta-analysis over the entire set of 81 (male and female) samples, with participant sex (male *vs.* female) serving as a predictor variable. Because each female sample was embedded in a study alongside a male sample (to our knowledge, there are no all-female studies in this literature), there may be some degree of dependency between effect-size estimates from the same study. This study-level dependency can be modelled using a three-level-meta-analysis (*e.g.*, *Van den Noortgate et al., 2013*), which requires the estimation of a second, study-level, variance component ($\tau_2^2$) in addition to the first, sample-level, variance component ($\tau_1^2$).

## Specification-curve and multiverse meta-analysis

Invariably, the results of any meta-analysis are subject to a sequence of researcher-dependent decisions (or degrees of freedom) pertaining to the inclusion of studies, the assumed meta-analytic model and the estimation of its parameters (*Voracek, Kossmeier & Tran, 2019*). For instance, *Blanchard (2018a)* defined the proximal assessment of sexual orientation *via* same-sex marriage as sufficiently invalid to justify the exclusion of one of the largest samples (*Frisch & Hviid, 2006*) available. In contrast, samples comprised of individuals whose sexual orientation was classified using phallometric testing (*Blanchard et al., 2006*, *2000*) were considered worthy of inclusion; as were studies which categorized children as young as 2 years of age as "prehomosexual" (*Blanchard et al., 1995*; *VanderLaan et al., 2014*), based on the "femininity" rating of their behaviours; or studies which drew on expert ratings of criminal or clinical records (*Blanchard & Bogaert, 1998*; *Blanchard & Sheridan, 1992*; *Bogaert et al., 1997*) to categorize individuals. While the inclusion criteria of *Blanchard (2018a)* are based on subject-matter considerations, other researchers could disagree on substantive grounds and argue that same-sex marriage constitutes a less error-prone indicator of homosexual orientation than changes in penile blood volume upon exposure to certain stimuli (as is the case in

phallometric testing) and that the latter, not the former study, should be noneligible for meta-analysis. Consequently, other researchers could opt to analyse a different set of samples than *Blanchard (2018a)* and therefore obtain different results. The perceived validity of differing approaches for classifying sexual orientation is just one out of various crucial steps in the sequence of researchers' decisions, which lead to a particular meta-analysis.

With regards to the decision of which meta-analytic model should be fitted to the data, *Blanchard (2018a*, *2018b)* and *Blanchard et al. (2021)* solely reported the results of random-effects meta-analyses. Even though some researchers may argue that in the presence of effect-size heterogeneity random-effects models need to be fitted to the data, other researchers may disagree and choose a fixed-effect model instead (see *Hedges & Vevea, 1998*; *Rice, Higgins & Lumley, 2018*).

There are many more such decision steps or researcher degrees of freedom in specifying a meta-analysis, thus demanding researchers to choose one out of several possible alternatives and, for each combination of such alternatives, specify a different meta-analytic model. Thus, for each reported meta-analysis, there is a sizeable collection of unreported, alternatively specified, meta-analyses, which may lead to considerably different conclusions.

Instances of disagreement among researchers about how a given set of data should be analyzed are not confined to the realm of meta-analysis. The numerous degrees of freedom researchers have at their hands in primary data analysis are well documented (*Silberzahn et al., 2018*; *Simmons, Nelson & Simonsohn, 2011*; *Simonsohn, Simmons & Nelson, 2020*; *Steegen et al., 2016*). Pointing out the issue of researcher degrees of freedom in a given analysis may occasionally be taken up as an accusation of questionable research practices (*e.g.*, "*p*-hacking," "fishing for statistical significance"; *Gelman & Loken, 2013*). Yet, even in the absence of questionable research practices, researchers may still disagree considerably with respect to the goals of a given study and how the data should be analysed, and thus how the research question should be optimally answered (*e.g.*, *Auspurg & Brüderl, 2021*; *Silberzahn et al., 2018*).

Given the obvious structural similarities between primary data analysis and meta-analysis (effect sizes correspond to individual participant data), specification-curve and multiverse meta-analysis (*Voracek, Kossmeier & Tran, 2019*) builds on two almost identical approaches for tackling the issue of researcher degrees of freedom, taken from the realm of primary data analysis, namely specification-curve analysis (*Simonsohn, Simmons & Nelson, 2020*) and multiverse analysis (*Steegen et al., 2016*). These approaches identify all reasonably specified analysis plans and report all of the corresponding results. The set of results obtained then serves as the foundation for an extra data-analytic step of inferring the state of the substantive research question. For the identification of reasonable analysis plans, *Simonsohn, Simmons & Nelson (2020)* and *Voracek, Kossmeier & Tran (2019)* suggested to survey potential decision steps in the process of devising an analysis and to determine all possible combinations of the alternatives offered at each decision step. All nonredundant data-analytic specifications retrieved in this manner are then considered as reasonable specifications. For an application of specification-curve analysis in the closely

related field of birth-order effects on personality traits, see *Rohrer, Egloff & Schmukle (2017)*.

We follow the terminology introduced by *Voracek, Kossmeier & Tran (2019)* and refer to the decision steps in specifying a meta-analysis as Which and How factors. The Which factors (or, the data multiverse) refer to decisions leading up to the set of samples (or studies) included in a meta-analysis (*i.e.*, which primary studies are included in a meta-analysis) and are mostly determined by features of the samples themselves. The How factors (or, the model multiverse) subsume the decisions leading up to the technical implementation of a meta-analysis over a given set of samples, such as the selected meta-analytic model and the effect-size metric (*i.e.*, how the primary studies are meta-analysed). Table 6 lists the 10 Which factors we derived by scrutinizing the inclusion criteria of prior related meta-analyses (*Blanchard, 2018a*, *2018b*; *Blanchard & VanderLaan, 2015*; *Blanchard et al., 2021*), from one critical commentary by *Zietsch (2018)*, and from our own concerns.

1. The factor *In Previous Meta-Analysis* indicated whether a sample had been included in any of the seven previous meta-analyses (*Blanchard, 2004*, *2018a*, *2018b*, *2020*; *Blanchard & VanderLaan, 2015*; *Jones & Blanchard, 1998*). Given that all of these were conducted by the same group of researchers (spearheaded by Blanchard), some samples may well have been judged more favourable for meta-analytic inclusion (possibly based on prior convictions that the FBOE is real).

2. The factor *Lab* indicated whether Blanchard, Bogaert, Zucker, or VanderLaan were listed as co-authors of the publications the samples appeared in. These researchers frequently collaborated on FBOE publications.

3. The factor *Archives* indicated whether a study was published in the journal *Archives of Sexual Behavior*. Thirty-two out of the 64 male samples appeared in articles published in this journal. As of mid-2022, Zucker served as Editor and Blanchard, Bogaert and VanderLaan as Editorial Board members of this journal (https://www.springer.com/journal/10508/editors).

4. The factor *Feminine* indicated whether a given sample had been declared a "feminine" sample by *Blanchard (2018a)*, who theorized that the association between older brothers and homosexual orientation is stronger in "feminine" samples, as opposed to "non-feminine/cisgender" samples.

5. The factor *GDY* indicated whether samples comprised individuals diagnosed with GDY. With exception of *Vasey & VanderLaan (2007)* and the replication sample in *VanderLaan & Vasey (2011)*, all "feminine" samples comprised individuals diagnosed with GDY. It might be argued that GDY samples should be excluded from meta-analysis.

6. The factor *Clinical* indicated whether participants were recruited in a clinical setting. This issue was pointed out by *Zietsch (2018)*, who questioned the representativeness of such samples.

7. The factor *Classification* distinguished between four different methods for classifying sexual orientation, as used in the FBOE literature. We assigned each sample's classification scheme to one of the three broad categories (*i.e.*, factor levels), which reflect different levels of reliability of classification. The factor level *Single-Item* referred to studies which determined sexual orientation *via* a single piece of information (such as asking participants which category they would ascribe themselves to, or whether participants were in a same-sex relationship; *Frisch & Hviid, 2006*), as opposed to a summary score of multiple items or expert ratings. The factor level *Various* subsumed samples for which sexual orientation had been determined by computing summary scores over multiple items (such as the Kinsey scale), or which combined participants whose sexual orientation had been assessed *via* a mixture of methods, such as the "Blanchard" subsample in *Blanchard et al. (2006)*. The factor level *Indirect* was assigned to samples wherein participants were categorized by either employing behavioural measures (*e.g.*, phallometric testing, feminine behaviours in children as predictors of homosexual orientation), criminal offence records, or expert ratings to determine sexual orientation. Also, two studies (*Blanchard & Bogaert, 1996a*, Study 2; *Schagen et al., 2012*) did not classify sexual orientation *per se* (but used GDY diagnosis as a proxy). These studies are indicated by the factor level *none*.

8. The factor *Probability* indicated whether a sample was declared a "probability sample", *i.e.*, samples comprised of participants who were randomly selected with respect to sexual orientation *Zietsch (2018)*, or samples that had been obtained through some kind of random sampling procedure (*Skorska & Bogaert, 2020*; *Xu, Norton & Rahman, 2019*. All other samples were regarded as non-probabilistic convenience samples.

9. The factor *Stopping Rule* indicated whether the authors of a primary study or *Blanchard (2018a)* assumed the presence of a stopping rule, such as one-child policies (*e.g.*, *Xu & Zheng, 2014*), or parents ceasing to reproduce after having at least one male and one female offspring (*Blanchard & Lippa, 2007*). *Blanchard (2018a)* noted that the proposed excess of older brothers may go undetected in the presence of a stopping rule.

10. Finally, the factor *Pedo-/Hebephiles* indicated whether a sample comprised individuals who were classified as pedophile or hebephile by *Blanchard et al. (2021)*, who investigated this factor as a moderator.

We considered only a single factor in terms of how to analyse these subsets of samples from the *corpus* of primary studies (*i.e.*, the How factor), namely, fitting a random-effects model, as in *Blanchard (2018a, 2018b)* and *Blanchard et al. (2021)*, or a fixed-effect model.

Counting all the factor-level combinations (considering also their union, where applicable) of the 10 Which factors and the single How factor resulted in a total of 3 (In Previous Meta-Analysis) × 3 (Lab) × 3 (Archives) × 3 (Feminine) × 3 (GDY) × 3 (Clinical) × 5 (Classification) × 3 (Probability) × 3 (Stopping Rule) × 3 (Pedo-/Hebephiles) × 2 (fixed effect *vs.* random effects) = 196,830 possible meta-analytic specifications for the 64 male samples. This set of specifications comprised 1,628 unique, non-redundant combinations of at least two from the 64 male samples in Table 4 (*i.e.*, the set of reasonable specifications; see analysis code provided online: https://osf.io/3wnhu/). Owing to the lack of previous

meta-analyses of an older brother effect in female samples, the factor *In Previous Meta-Analysis* was not applicable when determining the subset of possible meta-analyses for female samples, as was the case for the factors *Feminine* and *Pedo-/Hebephile*. For the remaining applicable factors, there were 3 (Lab) × 3 (Archives) × 3 (GDY) × 3 (Clinical) × 5 (Classification) × 3 (Probability) × 3 (Stopping Rule) × 2 (fixed effect *vs.* random effects) = 7,290 possible meta-analytic specifications. Of these, 212 turned out to be unique and non-redundant specifications comprising at least two from the 17 available female samples.

All analyses were carried out in *R* (version 4.0.3; *R Core Team, 2021*). Meta-analyses were computed using the *metafor* package (version 3.0.2; *Viechtbauer, 2010*), graphical displays of results were generated using *metaviz* (Version 0.3.1; *Kossmeier, Tran & Voracek, 2021*), and *ggplot2* (*Wickham et al., 2019*; see online material for code; https://osf.io/3wnhu/).

We follow recommendations (*Morey et al., 2016*) of stating how the reported interval estimates were obtained. Confidence intervals for summary ln*OR*s were based on the normal approximation to the sampling distribution of the point estimator of the summary effect (*i.e.*, the ln*OR*s; these intervals being the default option in *metafor*; *Viechtbauer, 2010*). Confidence intervals for the between-sample standard deviation, $\tau$, were based on the profile likelihood of $\tau^2$ (*Hardy & Thompson, 1996*), which is the default when fitting meta-analyses using the *rma.mv()* function in the *metafor* package (*Viechtbauer, 2010*). Prediction intervals were obtained using the quantiles of the standard normal distribution. This is also the default option in *metafor*.

Further, in reporting the results of the meta-analytic specification-curves for male and female samples, we also made use of the *S* value (*Rafi & Greenland, 2020*), a mapping of the *p* value onto an additive scale without an upper bound, given by $S = \log_2(1/p)$, where $\log_2$ denotes the base two logarithm. Like the *p* value, the *S* value indicates the degree of compatibility between the observed data and the statistical null model, which not only comprises the null hypothesis, but also a host of statistical background assumptions, pertaining to the assumed data generating process, the absence of selection bias, and the sufficiently correct specification of the model, among other possible assumptions (*Greenland et al., 2022*; *Rafi & Greenland, 2020*). The *S* value is interpretable as bits of information against the null model. For instance, a (just significant) *p* value of 0.05 corresponds to an *S* value of 4.32, *i.e.*, conveys 4.32 bits of information against the null model. The unit of bits can further be interpreted as a surprisal value for observing a given *S* value. An *S* value of 4.32 would be as surprising as observing four heads out of four tosses of a fair coin–which is not that much surprising and certainly would not already justify the conclusion that the coin is unfair.

## RESULTS

### Meta-analyses of the five specific study sets

Table 7 provides a summary of the meta-analyses over the five specific subsets (Men included in *Blanchard, 2018a*; Men included in *Blanchard, 2018b*; Men included in *Blanchard et al., 2021*; Men full set; Women full set) of male and female samples using

ln*OR* as an effect size (the fixed-effect analysis results are provided in Supplement 6, Subsection 2). There was little disagreement between the estimated summary ln*OR*s across all four specific male study sets. Depending on which specific study set is interpreted, the odds for observing an older brother among the set of all older siblings reported by homosexual participants (male or female) were between 7% (for the Women full set) and 17% (for the 31 samples included in *Blanchard, 2018a*) greater than those same odds for the heterosexual participants. However, the 95% *CI*s suggest that these estimates were compatible with a 6% decrease as well as with a 35% increase (*i.e.*, the respective lower and upper bounds of the 95% *CI* of the summary estimate for the six probability samples included in *Blanchard, 2018a*) for these odds. While the precision of these summary ln*OR*s varied considerably across the specific study sets (the shortest 95% *CI* spanned 0.11 ln*OR*s, the widest 95% *CI* spanned 0.36 ln*OR*s), all of the point estimates were directionally consistent with a greater number of older brothers among the older siblings of homosexual men as predicted by the FBOE. This greater number of older brothers was not specific to the male samples, however, as the summary ln*OR*s for the female samples were also consistent with a greater number of older brothers in homosexual women, with an estimated 7% greater odds of observing an older brother among the older siblings of homosexual *vs* heterosexual women.

As stated above, we also combined the male and female samples into a single data set and fitted random-effects meta-anlyses using sex (male *vs* female) as a predictor variable to compare ln*OR*s between these two groups of samples. The regression coefficient for sex can be interpreted as an estimate of the difference in ln*OR* between male and female samples.

This difference amounted to 0.07, 95% *CI* [0.08–0.22], in the random-effects model. The three-level meta-analysis returned results almost identical to this model, with a between-group difference of 0.06, 95% *CI* [−0.08 to 0.20], $\hat{\tau}_1 = 0.12$, 95% *CI* [0.00–0.22], $\hat{\tau}_2 = 0.11$, 95% *CI* [0.00–0.22]. These results suggest that conditional on the assumed statistical model(s), the available data do not convey enough information to confidently declare that the male and female samples differ with respect to the greater number of older brothers among the older siblings of homosexual *vs*. heterosexual individuals. To put things in perspective: this inconclusive finding is based on the data of 30,000 homosexual individuals (17,134 older brothers and 15,286 older sisters reported). Moreover, it calls into question previous confident claims about the excess of older brothers in homosexual individuals being specific to men (*e.g.*, *Bogaert et al., 1997*; *Bogaert & Skorska, 2011*).

With regards to effect-size heterogeneity, the $\hat{\tau}$s for the random-effects models were all of the same magnitude as the corresponding summary ln*OR*s themselves. Population effects thus appeared highly variable across samples. We computed 95% prediction intervals (*PI*s) to illustrate effect-size heterogeneity. A 95% *PI* can be interpreted as providing an estimate of where 95% of the population effects of future studies might fall (*IntHout et al., 2016*). The 95% *PI* suggest a wide range of effects, which are consistent with both a greater as well as a lower number of older brothers relative to older sisters among homosexual *vs*. heterosexual men, depending on the population.

To facilitate the comparison of the magnitude of the *OR* for older brothers *vs* older sisters to that of the *OBOR*, Table 8 lists the results of the meta-analyses using the ln*OBOR*

**Table 7 Summary of results of random-effects meta-analyses using lnOR as an effect size.**

| Set | k | Group | N | #OB | #OS | REM [95% CI] (Unit: lnOR) | $\hat{\tau}$ [95% CI] | PI |
|---|---|---|---|---|---|---|---|---|
| Men included in *Blanchard (2018a)* | 31 | Homo | 7,141 | 5,447 | 4,523 | 0.16 [0.09– 0.24] | 0.12 [0.00–0.25] | [−0.09 to 0.41] |
| | | Hetero | 12,504 | 7,245 | 6,885 | | | |
| Men included in *Blanchard (2018b)* | 6 | Homo | 2,335 | 971 | 778 | 0.12 [−0.06 to 0.30] | 0.14 [0.00–0.46] | [−0.22 to 0.45] |
| | | Hetero | 445,301 | 155,579 | 123,289 | | | |
| Men included in *Blanchard et al. (2021)* | 24 | Homo | 6,084 | 4,241 | 3,712 | 0.10 [0.04–0.16] | 0.00 [0.00–0.12] | - |
| | | Hetero | 12,118 | 7,303 | 6,933 | | | |
| Men full set | 64 | Homo | 20,863 | 12,863 | 11,610 | 0.12 [0.07– 0.18] | 0.13 [0.05–0.21] | [−0.13 to 0.38] |
| | | Hetero | 552,365 | 204,527 | 171,438 | | | |
| Women full set | 17 | Homo | 10,178 | 4,271 | 3,676 | 0.07 [−0.12 to 0.26] | 0.28 [0.02–0.52] | [−0.51 to 0.64] |
| | | Hetero | 545,247 | 211,425 | 175,487 | | | |

**Note:**
k, number of samples; #OB, number of older brothers; #OS, number of older sisters; REM, estimated random-effects mean; τ with hat, estimated standard deviation of population effects; *PI*, 95% prediction interval. "-", Redundant; as due to the REML estimate of the standard deviation of population effects being 0 exactly, the prediction intervals and confidence intervals are numerically identical.

as an effect size. It is evident that if the *OBOR* is falsely interpreted as quantifying the effect of interest, one would exaggerate the magnitude of the effect in male samples considerably.

*Blanchard (2018a*; fourth meta-analysis) found that some of the heterogeneity of the *OBOR* could be explained by two distinct subgroups of samples, namely "feminine" samples and "non-feminine/cisgender" samples. The re-analysis of *Blanchard (2018a)* using the lnOR returned random-effects summary estimates of lnOR = 0.09, 95% *CI* [0.03–0.17], and lnOR = 0.29, 95% *CI* [0.17–0.41]. Across all 47 samples which were not denoted "feminine" by *Blanchard (2018a)*, we obtained random-effects estimates of lnOR = 0.08, 95% *CI* [0.02–0.13]. This subgroup analysis suggests a sizeable difference in the effect between "feminine" and "non-feminine/cisgender" individuals.

Figure 3 displays a raindrop forest plot (*Kossmeier, Tran & Voracek, 2021*; *Schild & Voracek, 2015*; see *Kossmeier, Tran & Voracek, 2020*, for the advantages of this graphical device) accompanying the random-effects meta-analysis of all male-sample lnOR effect sizes (the respective plot of the fixed-effect meta-analysis is provided in Supplement S6, Subsection 2). Therein, the observed effect-size estimates are arranged (from top to bottom) in ascending order of their meta-analytic weight. Two patterns are visible in Fig. 3. First, only seven out of 64 (11%) confidence intervals' lower bounds were greater than zero. That is, only seven samples returned a statistically significant effect in the right direction when using the (adequate) lnOR effect size metric. This contrasts the "consistent" and "reliable" (*e.g.*, *Blanchard, 2004*; *Blanchard & VanderLaan, 2015*) greater number of older brothers relative to older sisters among homosexual *vs.* heterosexual men previously reported in all these primary studies.

Second, with increasing study weight, the effect estimates seemed to approach zero. This is indicative of small-study effects (*i.e.*, smaller, less precise, studies report larger effects than larger, more precise, studies). The funnel plot in Fig. 4 further illustrates this observation. Assuming the absence of publication bias, one would expect the observed

**Table 8 Summary of results of random-effects meta-analyses using lnOBOR as an effect size.**

| Set | k | Group | N | #OB | #Other | REM [95% CI] (Unit: lnOBOR) | $\hat{\tau}$ | PI |
|-----|---|-------|---|-----|--------|------------------------------|--------------|-----|
| Men included in *Blanchard (2018a)* | 31 | Homo | 7,141 | 5,447 | 11,926 | 0.39 [0.28– 0.51] | 0.27 [0.18–0.39] | [−0.14 to 0.93] |
| | | Hetero | 12,504 | 7,245 | 21,662 | | | |
| Men included in *Blanchard (2018b)* | 6 | Homo | 2,335 | 971 | 2,582 | 0.20 [0.12– 0.27] | 0.00 [0.00–0.29] | - |
| | | Hetero | 445,301 | 155,579 | 525,105 | | | |
| Men included in *Blanchard et al. (2021)* | 24 | Homo | 6,084 | 4,241 | 9,985 | 0.25 [0.20– 0.30] | 0.00 [0.00–0.13] | |
| | | Hetero | 12,118 | 7,303 | 21,495 | | | |
| Men full set | 59* | Homo | 12,364 | 8,342 | 18,837 | 0.33 [0.26– 0.40] | 0.20 [0.14–0.28] | [−0.06 to 0.72] |
| | | Hetero | 470,157 | 167,922 | 562,943 | | | |
| Women full set | 14[†] | Homo | 3,074 | 1,210 | 3,180 | 0.07 [−0.07 to 0.22] | 0.14 [0.00–0.36] | [−0.24 to 0.39] |
| | | Hetero | 472,553 | 176,499 | 547,359 | | | |

**Note:**
k, number of samples; #OB, number of older brothers; #Other, number of older sisters; REM, estimated random-effects mean; τ with hat, estimated standard deviation of population effects; *PI*, 95% prediction interval. "-", redundant; as due to the REML estimate of the standard deviation of population effects being 0 exactly, the prediction intervals and confidence intervals are numerically identical. *k = 59 (*vs* 64) due to nonreporting of number of younger brothers and/or younger sisters in *Blanchard & Lippa (2007)*, *Camperio-Ciani, Iemmola & Blecher (2009)*, *Kangassalo, Pölkki & Rantala (2011)*, *Yule, Brotto & Gorzalka (2014)*, and *Xu, Norton & Rahman (2019)*.
[†]k = 14 (*vs* 17) due to nonreporting of number of younger brothers and/or sisters in *Blanchard & Lippa (2007)*, *Bogaert (2010)*, and *Xu, Norton & Rahman (2019)*.

effect sizes to be distributed symmetrically around the summary effect estimate (*Vevea, Coburn & Sutton, 2019*). This was visually not the case in Fig. 4 and the observed distribution of effect sizes also appeared to be incompatible with a model assuming symmetry of effect sizes, as was indicated by *Egger et al. (1997)*'s regression: Regressing the observed effect sizes on their standard errors, the regression coefficient for the standard error was 0.80, 95% *CI* [0.28–1.31], in the random-effects model. This indicates that (when rescaling these coefficients by dividing them by 10), on average, two effect sizes differing by 0.1 standard errors were expected to differ by 0.08 ln*OR*s. This difference was sizeable in comparison to all summary estimates listed in Table 7.

### Specification-curve and multiverse meta-analyses

To demonstrate which other sample-specific factors could potentially influence the estimates of the meta-analytic summary estimates and thus the qualitative conclusion about the presence of a greater number of older brothers relative to older sisters among homosexual *vs* heterosexual men, we now consider the results of the specification curve meta-analyses described above.

The results of the 1,628 and 212 non-redundant meta-analytic specifications for the set of male and female samples are summarized in Figs. 5 and 6. From top to bottom, the panels encode each specification's estimated summary effect and 95% confidence interval, number of samples included (density plot), and combination of factor levels (tile plot). The point estimates are arranged in ascending order and are connected by a black solid line, representing the specification curve. In addition, each specification is color-coded, using hues of six distinct colors (red, orange, yellow, green, blue, violet) on the spectrum from red to violet. A specification's color and hue are indicative of the number of samples included in the meta-analysis, with the number of samples increasing in the following order: Red, orange, yellow, green, blue, violet.

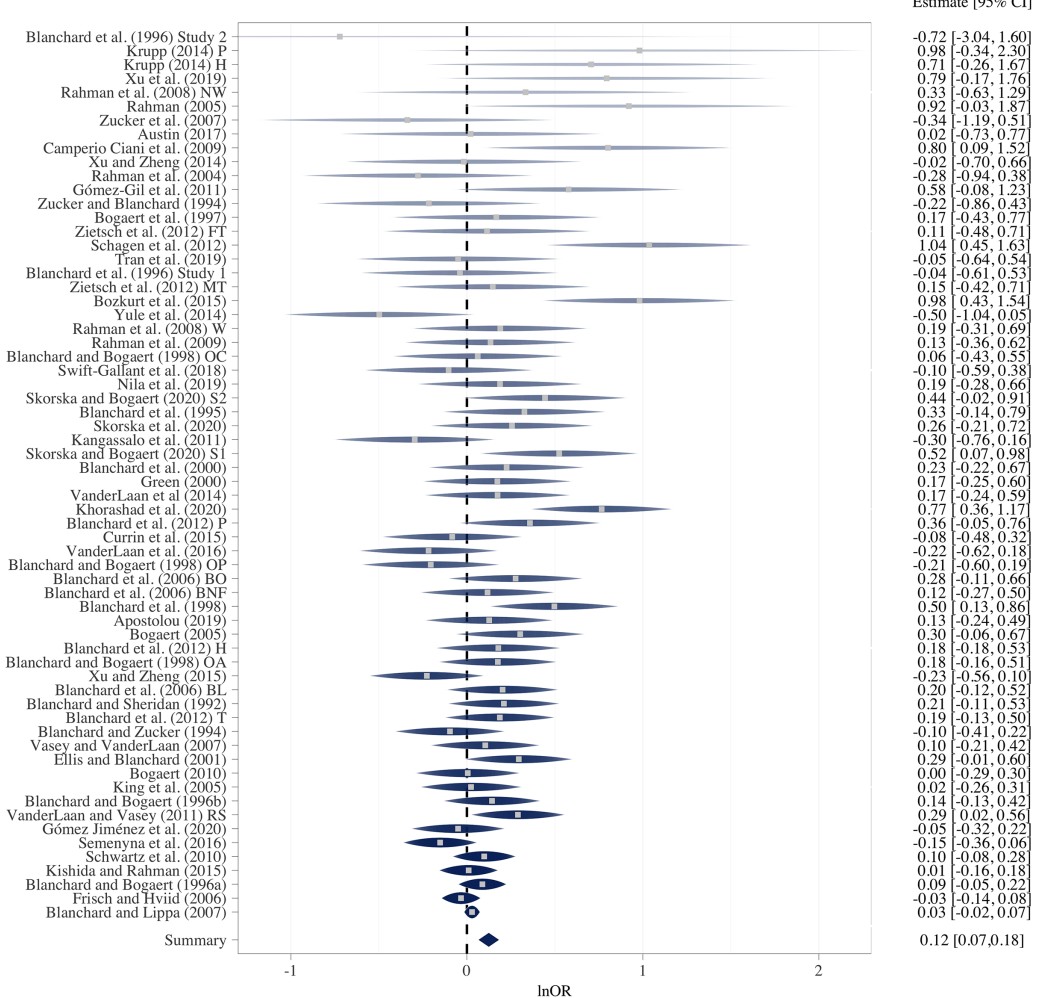

Estimate [95% CI]

| Study | Estimate [95% CI] |
|---|---|
| Blanchard et al. (1996) Study 2 | -0.72 [-3.04, 1.60] |
| Krupp (2014) P | 0.98 [-0.34, 2.30] |
| Krupp (2014) H | 0.71 [-0.26, 1.67] |
| Xu et al. (2019) | 0.79 [-0.17, 1.76] |
| Rahman et al. (2008) NW | 0.33 [-0.63, 1.29] |
| Rahman (2005) | 0.92 [-0.03, 1.87] |
| Zucker et al. (2007) | -0.34 [-1.19, 0.51] |
| Austin (2017) | 0.02 [-0.73, 0.77] |
| Camperio Ciani et al. (2009) | 0.80 [0.09, 1.52] |
| Xu and Zheng (2014) | -0.02 [-0.70, 0.66] |
| Rahman et al. (2004) | -0.28 [-0.94, 0.38] |
| Gómez-Gil et al. (2011) | 0.58 [-0.08, 1.23] |
| Zucker and Blanchard (1994) | -0.22 [-0.86, 0.43] |
| Bogaert et al. (1997) | 0.17 [-0.43, 0.77] |
| Zietsch et al. (2012) FT | 0.11 [-0.48, 0.71] |
| Schagen et al. (2012) | 1.04 [0.45, 1.63] |
| Tran et al. (2019) | -0.05 [-0.64, 0.54] |
| Blanchard et al. (1996) Study 1 | -0.04 [-0.61, 0.53] |
| Zietsch et al. (2012) MT | 0.15 [-0.42, 0.71] |
| Bozkurt et al. (2015) | 0.98 [0.43, 1.54] |
| Yule et al. (2014) | -0.50 [-1.04, 0.05] |
| Rahman et al. (2008) W | 0.19 [-0.31, 0.69] |
| Rahman et al. (2009) | 0.13 [-0.36, 0.62] |
| Blanchard and Bogaert (1998) OC | 0.06 [-0.43, 0.55] |
| Swift-Gallant et al. (2018) | -0.10 [-0.59, 0.38] |
| Nila et al. (2019) | 0.19 [-0.28, 0.66] |
| Skorska and Bogaert (2020) S2 | 0.44 [-0.02, 0.91] |
| Blanchard et al. (1995) | 0.33 [-0.14, 0.79] |
| Skorska et al. (2020) | 0.26 [-0.21, 0.72] |
| Kangassalo et al. (2011) | -0.30 [-0.76, 0.16] |
| Skorska and Bogaert (2020) S1 | 0.52 [0.07, 0.98] |
| Blanchard et al. (2000) | 0.23 [-0.22, 0.67] |
| Green (2000) | 0.17 [-0.25, 0.60] |
| VanderLaan et al (2014) | 0.17 [-0.24, 0.59] |
| Khorashad et al. (2020) | 0.77 [0.36, 1.17] |
| Blanchard et al. (2012) P | 0.36 [-0.05, 0.76] |
| Currin et al. (2015) | -0.08 [-0.48, 0.32] |
| VanderLaan et al. (2016) | -0.22 [-0.62, 0.18] |
| Blanchard and Bogaert (1998) OP | -0.21 [-0.60, 0.19] |
| Blanchard et al. (2006) BO | 0.28 [-0.11, 0.66] |
| Blanchard et al. (2006) BNF | 0.12 [-0.27, 0.50] |
| Blanchard et al. (1998) | 0.50 [0.13, 0.86] |
| Apostolou (2019) | 0.13 [-0.24, 0.49] |
| Bogaert (2005) | 0.30 [-0.06, 0.67] |
| Blanchard et al. (2012) H | 0.18 [-0.18, 0.53] |
| Blanchard and Bogaert (1998) OA | 0.18 [-0.16, 0.51] |
| Xu and Zheng (2015) | -0.23 [-0.56, 0.10] |
| Blanchard et al. (2006) BL | 0.20 [-0.12, 0.52] |
| Blanchard and Sheridan (1992) | 0.21 [-0.11, 0.53] |
| Blanchard et al. (2012) T | 0.19 [-0.13, 0.50] |
| Blanchard and Zucker (1994) | -0.10 [-0.41, 0.22] |
| Vasey and VanderLaan (2007) | 0.10 [-0.21, 0.42] |
| Ellis and Blanchard (2001) | 0.29 [-0.01, 0.60] |
| Bogaert (2010) | 0.00 [-0.29, 0.30] |
| King et al. (2005) | 0.02 [-0.26, 0.31] |
| Blanchard and Bogaert (1996b) | 0.14 [-0.13, 0.42] |
| VanderLaan and Vasey (2011) RS | 0.29 [0.02, 0.56] |
| Gómez Jiménez et al. (2020) | -0.05 [-0.32, 0.22] |
| Semenyna et al. (2016) | -0.15 [-0.36, 0.06] |
| Schwartz et al. (2010) | 0.10 [-0.08, 0.28] |
| Kishida and Rahman (2015) | 0.01 [-0.16, 0.18] |
| Blanchard and Bogaert (1996a) | 0.09 [-0.05, 0.22] |
| Frisch and Hviid (2006) | -0.03 [-0.14, 0.08] |
| Blanchard and Lippa (2007) | 0.03 [-0.02, 0.07] |
| Summary | 0.12 [0.07, 0.18] |

lnOR

**Figure 3  Rainforest plot of the random-effects meta-analysis of all male sample ln*OR* effect sizes.** In this plot, the sample weights are both encoded in the degree of saturation and the thickness of the shaded regions, with heavier effect sizes being more saturated and thicker. The widths of the shaded regions correspond to 95% confidence intervals (*CI*); the point estimates are represented by the ticks inside these intervals. The teardrop shape encodes the likelihood of true values over the range of the 95% *CI*, given the observed estimate and assuming the ln*OR*s to be normally distributed with known variance.

In terms of magnitude, the estimated mean ln*OR*s of the various subsets of male samples ranged from −0.29 (*OR* = 0.74) to 0.93 (*OR* = 2.53), with an interquartile range of 0.05 to 0.14 (*OR* = 1.05 to 1.15). In total, 96% of the estimated means were greater than 0, and 47.4% of these had 95% *CI*s that did not include 0 (*i.e.*, estimated summary effects greater than 0, with $p < 0.05$, two-tailed).

The overall pattern of the specification curve seems to indicate that the specifications with many samples and/or narrow confidence intervals were closer to zero, as opposed to specifications with only a few samples and/or wider confidence intervals.

The Which factors *GDY* and *Feminine* systematically influenced the magnitude of the summary effect. Specifications that included samples containing gender-dysphoric

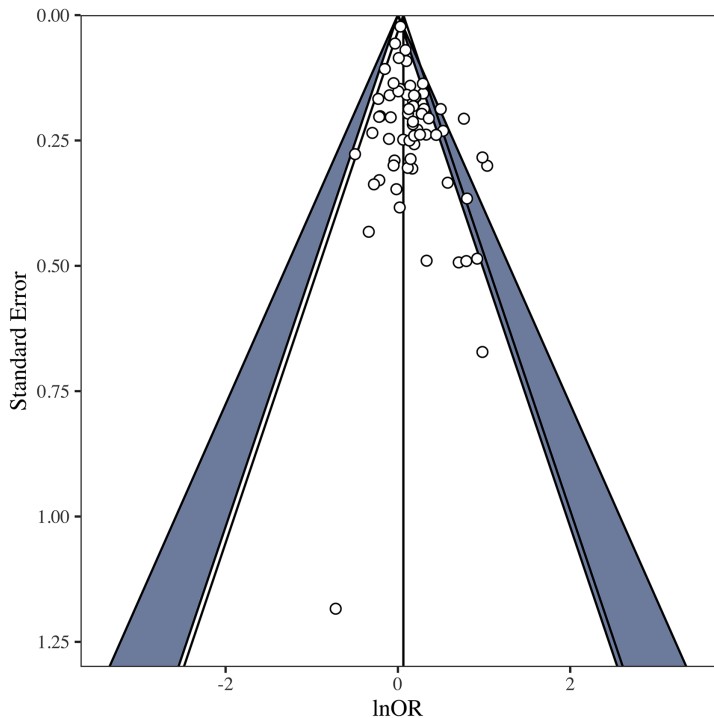

**Figure 4 Contour-enhanced funnel plot of all male sample ln*OR* effect sizes in the fixed-effect model.**
The blue regions are centred at zero and highlight effect sizes which were significantly ($0.01 < p < 0.05$)
different from zero (taking their precision into account).

individuals, or "feminine" (*Blanchard, 2018b*) samples only, produced some of the largest
effects. When including GDY as a predictor, the random-effects summary estimates for the
52 non-GDY and the 12 GDY samples were ln*OR* = 0.09, 95% *CI* [0.03–0.15], and
ln*OR* = 0.31, 95% *CI* [0.16–0.46].

Figure 5 also indicates that as the meta-analytic summary estimates increased, so did the
frequency of specifications encompassing only those samples, which had already been
included in at least one of the previous six meta-analyses (*Blanchard, 2004*, *2018a*, *2018b*;
*Blanchard et al., 2020*, *2021*; *Blanchard & VanderLaan, 2015*; *Jones & Blanchard, 1998*).
Including the Which factor *In Previous MA* as a moderator in the meta-analysis consisting
of all 64 samples, the random-effects summary estimates were ln*OR* = 0.00, 95% *CI* [−0.13
to 0.13], and ln*OR* = 0.15, 95% *CI* [0.09–0.21] (a difference of 0.15, 95% *CI* [−0.01 to 0.29],
in favour of studies included in previous meta-analyses).

The meta-analytic specifications across the female samples returned similar results.
The summary effects ranged from −0.22 to 0.54, with an interquartile range of 0.05 to 0.16.
A total of 84.9% of estimated means were greater than 0 of which 32.2% did not include
zero. Again, tighter confidence intervals were observed for summary effects close to zero.
No systematic pattern of Which/How factor level combinations was apparent.

*Voracek, Kossmeier & Tran (2019)* recommended assessing the overall evidence from a
specification-curve and multiverse meta-analysis against the respective null model with
parametric bootstrapping (inferential specification-curve plots) and an additional
histogram of *p* values. Here, we extend the idea of the histogram of *p* values to the *S* value.

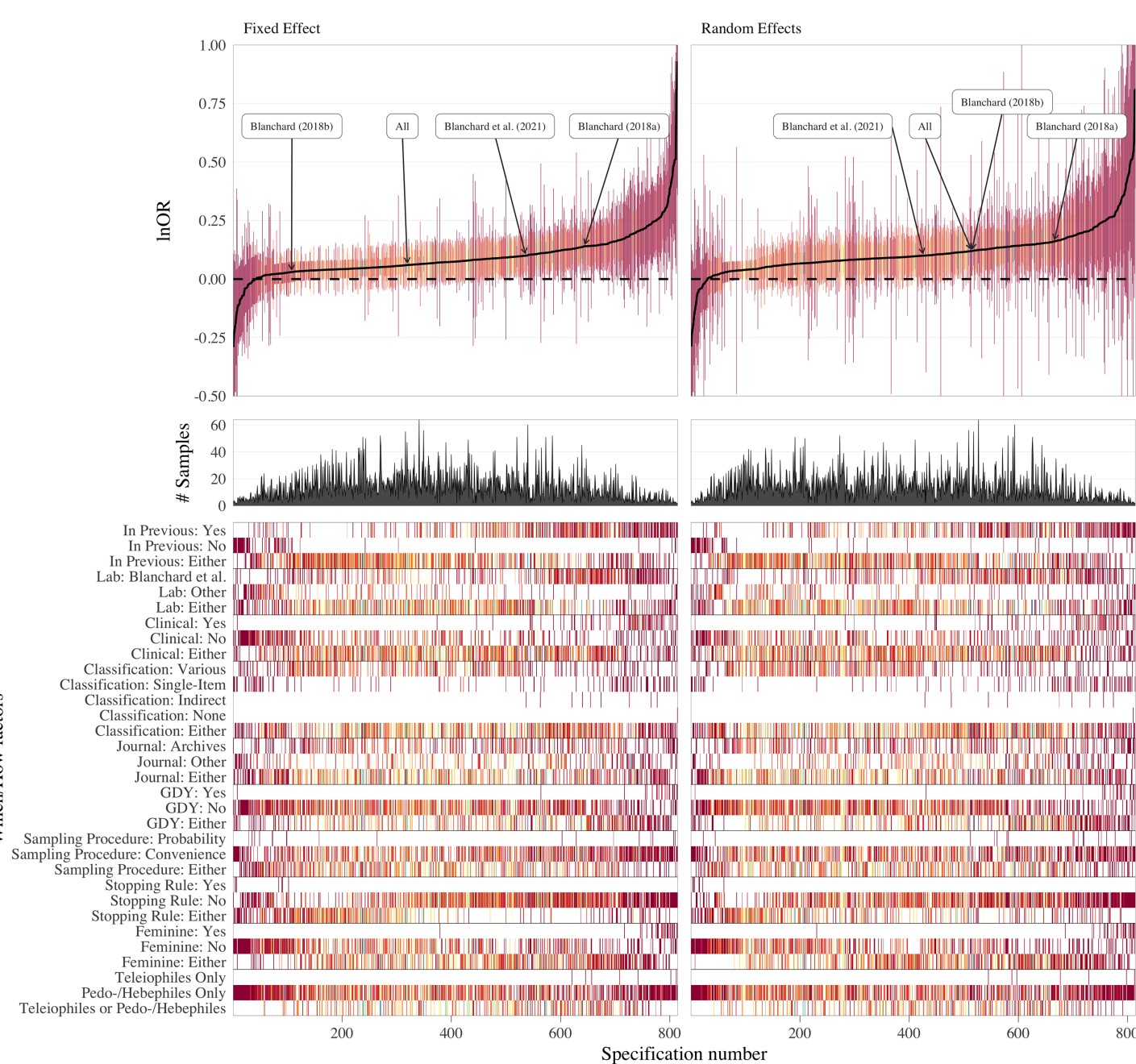

**Figure 5 Specification-curve analysis of the male samples.** From top to bottom, the panels encode each specification's estimated summary effect and 95% confidence interval, the number of samples included in the specifications (density plot), and the respective combination of factor levels (tile plot). The point estimates are arranged in ascending order and are connected by a black solid line, *i.e.*, the specification curve. On this curve, the locations of previously published meta-analyses is highlighted. Each specification is color-coded, using hues of six distinct colors (red, orange, yellow, green, blue, violet) on the spectrum from red to violet. A specification's color and hue are indicative of the number of samples included in the meta-analysis, with the number of samples increasing in the following order: red, orange, yellow, green, blue, violet.

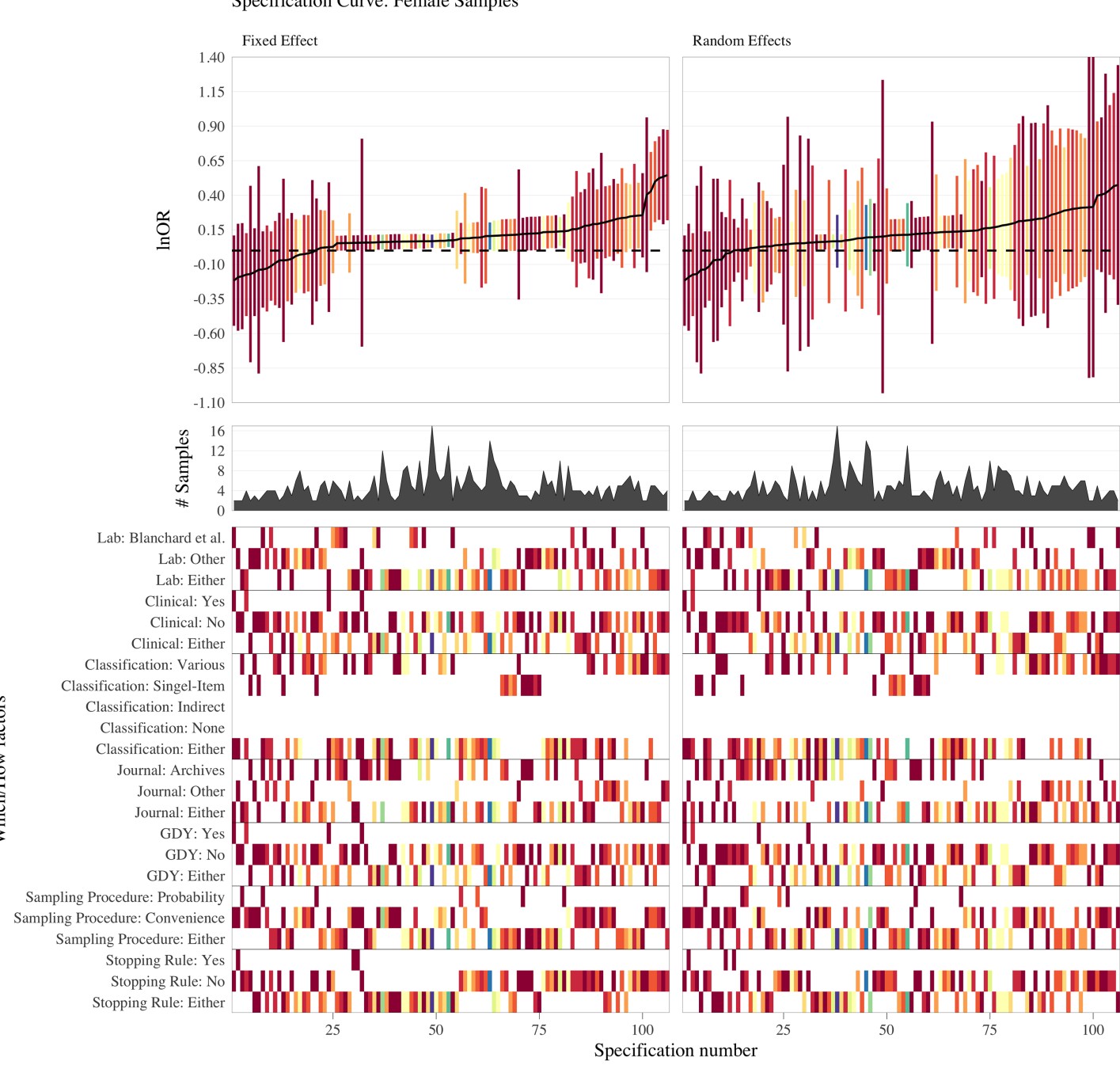

**Figure 6 Specification-curve analysis of the female samples.** From top to bottom, the panels encode each specification's estimated summary effect and 95% confidence interval, the number of samples included in the specifications (density plot), and the respective combination of factor levels (tile plot). The point estimates are arranged in ascending order and are connected by a black solid line, *i.e.*, the specification curve. Each specification is color-coded, using hues of six distinct colors (red, orange, yellow, green, blue, violet) on the spectrum from red to violet. A specification's color and hue are indicative of the number of samples included in the meta-analysis, with the number of samples increasing in the following order: red, orange, yellow, green, blue, violet.

# PeerJ

[11] Note that we have reason to believe that these assumptions were not met, as we found evidence for a small-sample bias in the *corpus* of available primary studies.

A derivation of the distribution of $S$ values under the null model (exponential with rate parameter $\lambda = \ln(2)$), with which the observed distribution can be compared, is provided in Supplement S7. The inferential specification-curve plots are presented in the Supplement S8.

Figure 7 (left) displays histograms and kernel density estimates of the 1,628 observed $S$ values, as well as the probability density function of the exponential distribution with rate parameter $\lambda = \ln(2)$. Clearly, the distribution of the observed $S$ values deviates from both the expected and simulated distribution. Thus, conditional on all statistical background assumptions being met[11] (*Greenland et al., 2022*), a true mean ln*OR* of 0 (*i.e.*, the null hypothesis) appears incompatible with the results of the specification curve meta-analysis. We can get an idea of the degree of incompatibility (or distance) between the data and test model by thinking of the distribution of these $S$ value in terms of surprise. The interquartile range of observed $S$ values was (2.18 to 6.22), the median $S$ value was 4.02. That is, under the null model, the central 50% of results would be approximately as surprising as observing all heads in 2, 3, 4, 5 or 6 tosses of a fair coin.

The interquartile range of the distribution of $S$ values for the female samples (Fig. 7, right) was 0.73 to 4.47, with a median of 1.67.

A specification curve analysis for the difference between the summary estimates in men and women can be found in Supplement 9.

## DISCUSSION

In Part III, we obtained several estimates of the odds ratio for observing an older brother within the set of older siblings, as reported by homosexual *vs* heterosexual men under various conditions and compared these estimates to the same effect in women as a point of reference.

Suppose that the four meta-analyses of the specific male sample sets summarised in Table 7 had each been put forth by a different researcher. Researcher 1 might argue that only the estimate of the six probability samples should be considered as an adequate summary of the available evidence of the excess of older brothers in homosexual men and conclude that there is not enough information to confidently declare either positive or negative estimates to be incompatible with the available data. Researcher 2 might argue that only summary estimate of the 31 samples in *Blanchard (2018a)* should be regarded as a reasonable estimate. Judging by the summary effect's confidence interval, true effects of 9% and 27% greater odds are expected for observing an older brother among the older siblings of homosexual men; hence, the effect is most likely positive and thus consistent with the FBOE. While the remaining two researchers disagree with the second researcher's exclusion criteria or model considerations, they largely agree that there is a meaningful positive effect, perhaps not lower than 4%. All but Researcher 1 would infer that their results could have only occurred if older brothers indeed increased the odds of homosexual orientation *via* the MIH.

Researcher 5 might join this imagined conversation, noting that instead of focusing on the estimated summary effects, one should also consider the prediction intervals. Accordingly, Researcher 5 concludes that, taken at face value, all these meta-analyses

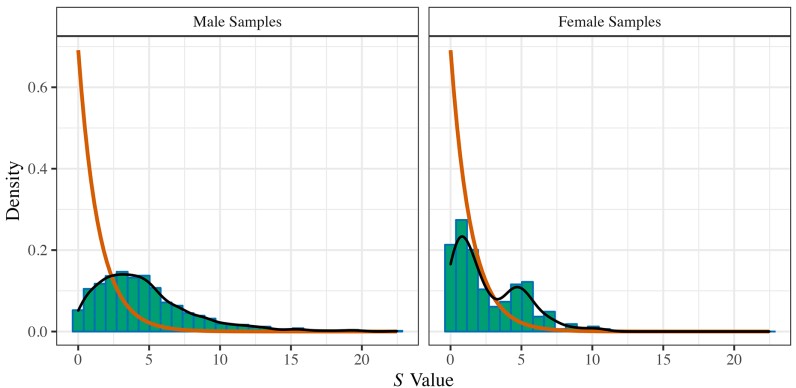

**Figure 7 Histograms and kernel-density estimates of the S values of the multiverses of meta-analytic evidence on the FBOE in all available data of male and female samples.** Histograms and kernel-density estimates of the S values of the multiverses of meta-analytic evidence on the FBOE in all available data of male (left) and female (right) samples. The plots display observed S values (histogram and black line) alongside the probability density function of an exponential distribution with rate parameter equal to ln (2) (orange line) for comparison.               

indicate that both the magnitude and direction of the effect are erratic and that the estimated summary effects are most likely inflated due to small-study effects. Furthermore, the very same methods which made the three favourable researchers believe in a meaningful association between the number of older brothers and homosexual orientation in men also suggest a similarly-sized association among homosexual women. Indeed, precisely this pattern (namely, the FBOE as not specific for men) is endorsed by the results of the very recent, extraordinarily large, population-based register study of *Ablaza, Kabátek & Perales (2022)*. Thus, our results have been independently replicated and are fully compatible with the currently best available evidence. Moreover, the observation of a similarly-sized association among homosexual women is incompatible with the current formulation of the MIH and therefore the estimated effects concerning the key observations of the FBOE contradict the biological explanation of the FBOE.

The results of Part III do not lend themselves to the confident conclusion that older brothers increase the odds of homosexual orientation and thus do not corroborate causal biological theories explaining this increase. The magnitude of the effect is highly uncertain and inconsistent across the available data. Even more important, the comparison of available male and female samples further suggests that homosexual orientation does not account for a large-enough chunk of the variation in older brothers beyond variation which may be attributable to an interplay of a host of unknown factors, which in turn might be unrelated to older brothers increasing the odds of homosexual orientation *via* the process of immunization (*Meehl, 1990*; *Yarkoni, 2022*). The observed statistical association between older brothers and sexual orientation (taken at face value) could be equally well explained by the *crud factor* (*Meehl, 1990*) rather than the MIH (for which there is almost no evidence). The crud factor refers to the truism that "in psychology and sociology everything correlates with everything" (*Meehl, 1990*, p. 204). These correlations/ associations are not type I errors. They reflect real, but complicated, causal interplays between a host of known as well as unknown variables. The MIH is just one in a large

collection of explanations one could think of as underlying the FBOE. Given that there is hardly any evidence for the MIH to begin with and that the MIH does not make any numerical prediction about the size of the association between the number of older brothers and sexual orientation, it is difficult to maintain that the MIH should be a more plausible substantive explanation than the crud factor. Moreover, if the observed association among women is considered as an estimate of the crud factor (the MIH does not apply to women), it then appears that the observed association in men is very much compatible with the crud factor.

The magnitude of the association in non-GDY individuals was much closer to 0, with an approximate 9% increase in odds. Assuming a proportion of 0.515 older brothers among the older siblings of heterosexual men, this estimate for the non-GDY group implies an approximated group difference in the proportion of older brothers among older siblings of 0.02. In comparison, the approximately 36% greater odds among GDY samples translates to a difference in proportion of 0.08. Moreover, taking the point estimate of the summary effect of $\ln OR = 0.07$ for women, we would find a difference in proportion of approximately 0.02 between homosexual and heterosexual women. It is difficult to argue that these effects are different in magnitude between men and women.

## WHY SHOULD GENDER-DYSPHORIC INDIVIDUALS SHOW AN EVEN GREATER EXCESS OF OLDER BROTHERS?

The specification-curve meta-analyses largely corroborate the findings assembled in Table 7 in that the effect fluctuated across different contexts–for men, as well as for women. The specification curves also suggested sizeable differences in the summary effect, *e.g.*, between the 11 samples comprised of male participants diagnosed with GDY *vs.* the 55 remaining samples.

Subgroup analyses (when taken at face value) would have researchers believe that the odds of observing an older brother among the older siblings of gender-dysphoric homosexual men are compatible with averages of between 16% and 61% greater odds of observing an older brother among older siblings than those same odds for the older siblings reported by the control group. In comparison, the summary effect for the older siblings of non-GDY homosexual men would be regarded as compatible with a range of 2% and 16%.

Three of the GDY samples employed behavioural measures to classify sexual orientation. In these samples, participants were categorized according to Blanchard's retrospective interpretation of their medical charts (*Blanchard & Sheridan, 1992*) or prospectively through the assessment of "feminine" behaviours of children and adolescents (*Blanchard et al., 1995*; *VanderLaan et al., 2014*). Two GDY samples (*Blanchard & Bogaert, 1996a*, Study 2; *Blanchard et al., 2012*) did not assess sexual orientation at all, but used GDY diagnoses as a proxy for homosexual orientation.

In addition, five of the GDY samples (*Blanchard & Bogaert, 1996a*; *Blanchard & Sheridan, 1992*; *Gómez-Gil et al., 2011*; *Green, 2000*) employed *Blanchard (1989)*'s typology for classifying gender dysphoric individuals as homo- *vs.* nonhomosexual/autogynephilic. The validity of this typology is disputed (*e.g.*, *Serano, 2010*). This inconsistency with

respect to the classification of sexual orientation may help to account for the relatively large, but uncertain summary effect in the GDY samples.

For instance, *Blanchard & VanderLaan, (2015)* and *VanderLaan et al. (2014)* remarked that the prospective classification of children as homosexual and heterosexual will inadvertently result in the misclassification of some individuals. However, their conclusion that any differences between groups that manage to be statistically significant, despite the extra noise brought about by the greater error in the assessment of sexual orientation, should be regarded as strong evidence that such a difference is incorrect (*Loken & Gelman, 2017*).

Given the many researcher degrees of freedom, it is generally not difficult to obtain a statistically significant difference between any groups (*Simmons, Nelson & Simonsohn, 2011*). However, for the effect estimate to become statistically significant at the 5% level, it must fall at least two standard errors away from zero. Greater measurement error implies greater standard errors and thus an estimate must be relatively far from zero to reach the significance threshold (*Gelman & Weakliem, 2009*; *Loken & Gelman, 2017*). The true difference between groups may however be much closer to zero than the statistically significant difference would indicate. As a consequence, observed differences between groups likely overestimate any underlying true difference (*Gelman & Carlin, 2014*).

Similarly, one study (*Schagen et al., 2012*) compared the number of older brothers of participants who had been formally diagnosed with GDY to a non-GDY control group–importantly, the sexual orientation of these participants was never assessed. Thus, if researchers are willing to treat the findings of *Schagen et al. (2012)* as evidence for the FBOE, they are essentially treating GDY and non-GDY as proxies for homosexual and heterosexual orientation, respectively. This deviates from numerous other study designs (*e.g.*, *Blanchard et al., 1995*; *Blanchard & Sheridan, 1992*) by the same group of researchers, wherein the sexual orientation of GDY participants was assessed and the difference in the number of older brothers between homosexual *vs* nonhomosexual GDY-participants was interpreted as evidence for the FBOE.

Thus, the same group of researchers appears to uphold contradictory convictions about how the FBOE should be investigated in GDY samples: On the one hand, a GDY diagnosis is regarded as a valid proxy for homosexual orientation, on the other hand, the meticulous distinction between nonhomosexual and homosexual GDYs must be made in order to detect a difference in the number of older brothers.

Overall, there seems to be a tremendous degree of heterogeneity regarding classification schemes, accompanied by noisy measurements and the overestimation of the excess of older brothers in one group *vs* the other. It is unsurprising that such samples would be associated with larger and more erratic effects, as opposed to samples with less measurement error and more similar classification schemes.

## GENERAL DISCUSSION AND CONCLUSION

This work set out to re-examine the statistical and empirical foundations of the FBOE by triangulating evidence from basic probability calculus (Part I), data simulation (Part II), and specification-curve and multiverse meta-analysis (Part III). In Parts I and II, we

showed that currently recommended and widely used methods and practices for quantifying the specific association between the number of older brothers and sexual orientation (*i.e.*, the theoretical estimand) rest on the false assumptions that the statistical models used to investigate this effect (a) need to adjust for the confounding effect of total family size and (b) that this can be achieved through the use of ratio variables. Furthermore, we re-emphasised the importance of adjusting for the number of older siblings, when investigating the FBOE. In Part III, we re-assessed the meta-analytic evidence for the suspected greater association between older brothers and homosexual orientation and older sisters and homosexual orientation in men using a more adequate effect-size metric. Moreover, we carried out the crucial comparison between male and female samples.

Evidently, the sibship data from over 30,000 homosexual men and women were not enough to estimate the size of the difference in the effect to a range of values which would permit confident inferences about its direction or magnitude (as judged by the 95% *CI* for the difference in effects). Across male samples, we also found evidence for a disproportionate influence of smaller samples on meta-analytic summary estimates. Such a pattern frequently signals publication bias and suggests that the meta-analytic summary estimates should be adjusted towards zero. A series of specification-curve and multiverse meta-analyses suggested an implausibly large and highly uncertain effect in samples comprising individuals diagnosed with GDY, which, upon closer examination, are expected to provide overestimates of any specific association between older brothers and sexual orientation, due to their inherently noisier classification of sexual orientation.

The uncertainty and the inconsistency (or variability) of the association between older brothers and homosexual orientation, as well as a similarly-sized effect in women, overshadows meaningful signals in the currently available data sources that might be attributable to the FBOE or the MIH. Convergent evidence for our findings was currently also published in the form of an extraordinarily large, population-based register study from the Netherlands (*Ablaza, Kabátek & Perales, 2022*).

We reiterate that the current results, but also those of *Blanchard (2018a, 2018b)* and *Blanchard et al. (2021)*, are predicated on the assumption that regarding reported siblings (as opposed to the individuals that reported them as the units of analysis) is a reasonable data-analytic approach. However, given the retrievable prior data, our analyses represent the most thorough and methodologically sound investigation of the FBOE and the MIH to date. The reason we had to work with reported siblings as the observations, rather than the participants or individuals whose data were recorded, lies in the systematic lack of reporting of basic summary statistics in the *corpus* of prior primary studies on the FBOE. Almost half of the means and standard deviations in Table 4 were not retrievable from the information provided in the respective primary publications, rendering their analyses and results irreproducible. Moreover, it appears that not a single data set on the FBOE has been made publicly available yet (the open data set presented in the current study thus is the first one).

Throughout this article, we adopted the assumption by Blanchard and colleagues that no other variables than those derivable from summary data constitute potential

confounders that need to be adjusted for. This assumption, however, is questionable. There is ample evidence that a host of other factors, such as parental age or parental loss (*e.g.*, *Frisch & Hviid, 2006*), are associated with sexual orientation. Arguably, parental age and loss are plausibly associated with an individual's number of older siblings (and, therefore, brothers) as well. Another important variable which cannot be adjusted for (because it is not obtainable from the available summary data), is the sexual orientation of the other siblings. Arguably, the FBOE predicts a pattern that within each family, the odds of homosexual orientation should increase per each older brother, while adjusting for all other relevant explanatory variables that occur at the family level. Large-scale studies, incorporating a host of potentially relevant explanatory variables (*Ablaza, Kabátek & Perales, 2022*; *Frisch & Hviid, 2006*), are urgently needed to estimate the effect of older brothers on sexual orientation.

## ACKNOWLEDGEMENTS

We thank Anton Rupprecht for his contributions in preparatory stages of this study.

### Funding

Open access funding was provided by the University of Vienna. The funders had no role in study design, data collection and analysis, decision to publish, or preparation of the manuscript.

### Grant Disclosures

The following grant information was disclosed by the authors:
University of Vienna.

### Competing Interests

The authors declare that they have no competing interests.

### Author Contributions

- Johannes K. Vilsmeier conceived and designed the experiments, performed the experiments, analyzed the data, prepared figures and/or tables, authored or reviewed drafts of the article, and approved the final draft.
- Michael Kossmeier analyzed the data, authored or reviewed drafts of the article, provided critical revision of analysis code and statistical analyses, and approved the final draft.
- Martin Voracek conceived and designed the experiments, authored or reviewed drafts of the article, supervised the project, and approved the final draft.
- Ulrich S. Tran conceived and designed the experiments, authored or reviewed drafts of the article, supervised the project, and approved the final draft.

### Data Availability

The data, codebook and analysis code are available at OSF: Vilsmeier, Johannes K, and Michael Kossmeier. 2022. "The Fraternal Birth-Order Effect as Statistical Artefact: Convergent Evidence from Probability Calculus, Simulated Data, and Multiverse Meta-Analysis." OSF. June 27. osf.io/3wnhu.

## Supplemental Information

Supplemental information for this article can be found online at http://dx.doi.org/10.7717/peerj.15623#supplemental-information.

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
