# Peer review of "The fraternal birth-order effect as a statistical artefact: convergent evidence from probability calculus, simulated data, and multiverse meta-analysis"

_PeerJ, doi:10.7717/peerj.15623_

## Round 0.1 · original submission · Major Revisions

With three reviews in hand, I can see that this manuscript is of great interest and can provide an impactful contribution to the FBOE.

Indeed, all the reviewers pointed out the relevance of this study and concurred that it is overall sound and well-conducted. However, due to its complexity, the manuscript also presents many parts that need to be improved or clarified to reach a wider audience and make its arguments more compelling.

Therefore, I recommend the authors closely look at every comment of each of the Reviewers and change their manuscripts under the guidance of the reviewers' insightful remarks.

·

Basic reporting

Basic reporting: Most of the reporting is provided in clear, unambiguous and professional English. I think there is some room to optimizing phrasing in some parts, more on that in my general comments.

Experimental design

Experimental design: The research question/central arguments seem well defined, relevant and meaningful. The research clearly fills/addresses a knowledge gap/issue.

Validity of the findings

Validity of the Findings: In general, I do not see any issue with the validity of the findings reported. I will refer to the conclusions in more detail in my general comments.

Additional comments

This article presents simulations to argue that the so-called fraternal birth order effect (older brothers increase the odds that a man reports a homosexual orientation) may be a statistical artifact. I will partially structure my review according to the points suggested by PeerJ and try to address all checks I’m supposed to provide. However, you will find the bulk of my comments under “General Comments” – sorry for that, this is just how I structure my reviews (and also how I like to receive reviews as an author). Also, I do not understand why I would paste my review into fields that are not sent to the authors (I'm writing the review explicitly with the authors in mind).

Generally speaking, I believe that this manuscript has great potential to make a meaningful contribution to the literature, clarifying a topic that has attracted some attention and lead to a considerable degree of confusion. I reviewed one of the discussed meta-analyses by Blanchard, so I got a brief glimpse of the literature, and it was really frustrating that nobody had fully addressed the underlying methodological issues. So, from my perspective, this manuscript is urgently needed to fill a gap in the literature; and I commend the authors for working their way through this literature (which seems to be a bit of a mess, in particular the more technical claims by Blanchard himself). My review focuses on ways that the manuscript could be improved to ensure that it is read and understood by the broadest possible audience. Also, I will only talk about Part 1 and Part 2. That is mainly because I’m running out of time and will be on vacation for the next two weeks. Sorry for that! I believe that my comments already impose quite a bit of work on the authors, and I’m happy to take a look at Part 3 during the revision (or maybe another reviewer will review that part in more detail).

Best regards,
Julia Rohrer (I sign all my reviews)


General Comments Major:

Starting from line 200: I can’t quite see yet how you move from (1) to (2). I understand that the numerator must be greater than the denominator, but these do not only contain P(PB|…) but also the other part (1 – P(OB|…)). Maybe there is an obvious reason why you could drop this, but it wasn’t obvious to me. Could you please explain?

Application of the law of total probability: Is…is that really the quickest and most intuitive way to get to equation 3? Maybe it is. I suspect you might lose readers here though (at least in my experience, psychologists tend to struggle with…really any sort of math). I guess if I needed to explain this, I may initially drop the conditioning on sexual orientation (and think about siblings of homosexuals and siblings of heterosexuals as separate samples). Then it’s really just the definition of a conditional probability re-arranged: P(OB|Older) = P(OB, Older)/P(Older) = P(OB)/P(Older). Not sure whether that makes sense to you. In any case, you may keep the derivation the way you did it, but I think you should either add some more verbalization to explain it, or simply provide a worked example with some numbers.

Starting from line 234: I find this illustration with numbers nice. However, as somebody coming from the substantive side, what I mainly read into what you are saying is: if we assume some birth order effect on sexual orientation (homosexual men having more older siblings), than we get a positive OBOR even if there is no excess of older brothers relative to older sisters. I think it might be helpful to spell this out? A birth order effect on sexual orientation may still be interesting (maybe). In general, I see that you are coming from the technical side, but I think for readers it is more helpful to have some guidance on what these numbers imply about reality.

Equation 6: Are you fully being serious about using the OR as a metric? I’d only suggest it to people who really, really, love Odds Ratios. I’d probably just compare the conditional probabilities P(OB|Older) between the two groups (the probability that an older sibling is a brother in homosexual men vs heterosexual men). Or, even better: estimate the effect of older brother (vs. older sister) on the probability of homosexuality, because that is probably how people think about the effect in their heads.

Line 293: I do not understand what you mean by a shift from the level of participants to the siblings reported by these participants. Wasn’t the data hierarchically organized before as well? Why would the OR but not the OBOR violate some assumption? I have a suspicion of what you are trying to get at, but it is really not clear to me. If you wanted to do an analysis on the level of the participant (which I would greatly prefer), it would be something like: Homosexuality ~ Number of older brothers + number of older siblings (that is, you compare people with the same number of older siblings but varying fractions of older brothers). (in this analysis, the effect of older brothers would not necessarily be causally identified due to endogeneous selection bias, but that is a bit of a separate question)
Note on the MROB: Did Blanchard really sort of try to reinvent statistical control? Oh boy, this literature is wild.

Line 374: I believe there’s also a substantive reason why the number of younger siblings should not be controlled for: it may be an outcome of the target’s sexuality, so controlling for it might induce collider bias (e.g., between the number of older brothers and sexuality, which may both affect parents’ decision to have more children).

Line 378: Your discussion makes it sound like what Gelman and Stern propose does not introduce collinearity, but simply controlling for older siblings does. But…the models will be exactly the same, model fit will be the same. The coefficients will just have a different meaning. But if you use the model to derive average marginal effects or something, the answers will be precisely the same. There’s three possible numbers that can be included in a model: #OS, #OB, #Older. The latter is the sum of the former two. Now if you do some linear combination of the first two and combine it with the latter that is…still the same pieces of information, you’re just splitting things up in a different way. The coefficients will have a different interpretation. But the model implied effects will be the same, as the model makes exactly the same prediction for everyone. Here’s a quick simulation:


set.seed(12345)
n <- 10000
older_sis <- rbinom(n, size = 4, prob = .50)
older_bro <- rbinom(n, size = 4, prob = .50)
older <- older_sis + older_bro
# Some arbitrary older brother effect
hom_prob <- 0.2 + older_bro * 0.10 + rnorm(n, sd = .1)
# using a probit link here
hom <- hom_prob > .5
table(hom)
# using an inadequate lpm because details dont matter here
m1 <- lm(hom ~ older_bro + older)
m2 <- lm(hom ~ older_bro + older_sis)
# lets do what gelman and stern suggest
diff <- older_bro - older_sis
m3 <- lm(hom ~ diff + older)
p1 <- predict(m1)
p2 <- predict(m1)
p3 <- predict(m1)
# it is all the same
p1[1:5]
p2[1:5]
p3[1:5]

As a side note: when there is a general birth order effect, m1 will give you no significant coefficient for older brothers. M2 will give you a significant coefficient for both older brothers and older sisters (which is, from a causal perspective, the right answer – if there is a birth order effect, that will be mediated through both older sisters and older brothers). M2 will give you no significant coefficient for the differences variable. See code below:

hom_prob2 <- 0.2 + older * 0.10 + rnorm(n, sd = .1)
hom2 <- hom_prob2 > .5
m1 <- lm(hom2 ~ older_bro + older)
m2 <- lm(hom2 ~ older_bro + older_sis)
m3 <- lm(hom2 ~ diff + older)
summary(m1)
summary(m2)
summary(m3)

Beginning of Part 2: Your explanation of what you mean by false-positive rate is a bit contorted. I guess the distinction your try to explain is “variance false positives” (that is what the statistical literature is mostly concerned with) vs. “bias false positives” (that is what, for example, the causal inference literature is often concerned with – Per Engzell used these terms in a paper I wrote with him). Maybe you just want to talk about “rate of mistaken conclusions” or something like that, that is much easier.

Methods of Part 2: I find it a bit confusing that you simulate older brother effects, but then also simulate independent effects of the number of older siblings (by setting the proportion of the number of older siblings differently for the different samples). What is an effect of the number of older siblings independently of an effect of older brothers? An effect of older sisters? Maybe you could try to setup your description of your simulations along a data generating process. So, you simulate a number of older brothers and a number of older siblings. Together, these are the number of older siblings. Now you generate the odds of homosexuality as a function of these variables. Yes, this requires you to reason backwards – deduce the birth order effect from the numbers reported in the literature – but your train of thought will probably be much easier to communicate.

Line 453: why would there be a positive correlation between the number of older brothers and the number of older sisters? Is that really an empirical observation? Like just some general fertility confounding or something? Also, what exactly would such a correlation induce in which model? You make it sound like this would necessarily result in an older sister effect that is non-zero, but if both types of siblings are in the same model, this shouldn’t happen, should it? The inclusion of older brothers blocks the backdoor path. You would just find a zero-order association.
Figures: I’d suggest that for Figure 1, you include clearer labels (in the actual figure) so that the figure can “stand on its own.” Right now, it’s impossible to make sense of the figure without reading the corresponding paragraph in parallel. I’d also really try to make the figure cognitively ergonomic. Like, where would values need to fall for unbiased estimation? There is no indication for that in the figure. I think this might require you to rethink the figures and maybe replot from scratch. However, I believe that this may be worth the effort. This is a very central part of your manuscript, and I’d want to make sure that it is accessible.

General Comments Minor (e.g., language):

Abstract, “For a quarter of a century researchers investigating the origins of sexual orientation have largely ascribed to the fraternal birth order effect (FBOE) as a fact, holding that older brothers increase the odds of homosexual orientation among men through an immunoreactivity process.”: I’m not a native speaker, but I’m not sure whether this sentence works in English. In any case, it is a bit convoluted. Maybe you could simplify this a bit for increased accessibility (e.g., start with what the fraternal birth order effect is, then talk about its status in the literature).

Abstract, “This yielded highly inconsistent and moreover similarly sized effects across 64 male and 17 female samples (N = 2,778,998), compatible with an excess as well as with a lack of older brothers in both groups, thus, suggesting that almost no variation in the number of older brothers in men is attributable to sexual orientation.”: There is a good chance that at this point, readers won’t know why the “similarly sized effects” are relevant. Once again, I think you can break this down into multiple sentence to make it more accessible.

Line 50, “Moreover, the FBOE implies an observed excess of older brothers in homosexual men, whereas the reverse does not follow logically.”: I don’t think I follow. If there is a FBOE, then homosexual men will have more older brothers; but if older men have more older brothers, the FBOE does not follow? You are right that confounding third variables might result in a spurious association between older brothers and homosexual orientation. But confounding factors could also hide a true causal effect (say, if there is a confounder that causes an excess of older sisters). So, an FBOE does not necessarily imply an observed excess of older brothers. (I don’t think it’s plausible that a true FBOE might be hidden because of some sort of confounding, but if you’re trying to make a logical statement, it should be waterproof in my opinion)

Line 69: What is a “smaller excess” of older sisters? This section was really confusing to me, maybe because it is so unclear how this statement relates to the previous statements. I think it might be worth trying to rewrite this section for greater clarity (I’d go for something like: second, the FBOE is often combined with the claim that homosexual men also have a slightly higher number of older sisters. This is borne out of empirical observation…and explained the following way…[which needs to be spelled out a bit more, in my opinion – there’s a step between miscarriages of XY male foetuses and an excess of older sisters that should be spelled out]).

Line 116: “These observations combined would strip the FBOE and the MIH of its verisimilitude”: In general, I liked this section because you spelled out your argument quite clearly. But this conclusion seemed a bit anti-climactic (and I don’t know whether epistemiologically, it makes sense to say that something is “stripped of its verisimilitude”). Maybe you could make it a bit more straightforward (think “These observation would undermine both the FBOE and the MIH as an explanation for homosexual orientation in men.”)

Line 147: I found the phrasing “implicitly defined at the level of siblings” a bit confusing. You clarify later, so it is not a big issue, but I certainly stumbled over it.

Reviewer 2 ·

Basic reporting

See below.

Experimental design

See below.

Validity of the findings

See below.

Additional comments

This paper assesses the empirical foundations of the Fraternal Birth Order Effect (FBOE) from three separate perspectives:
(a) a mathematical assessment of the relevant dependent variable
(b) a simulation study
(c) a multiverse meta-analysis

Before discussing each of these in turn, it is important to note that the authors rightly differentiate between
(i) The FBOE as a causal statement (i.e., older brothers increase the likelihood of homosexual orientation)
(ii) The FBOE as a merely correlational or observation statement

Within the FBOE as a causal statement, it seems important to further distinguish between
(i-weak) Older brothers increase the likelihood of homosexual orientation of younger-born males
(i-strong) Older brothers increase the likelihood of homosexual orientation younger-born males, but older sisters, younger brothers, and younger sisters have no effect on it

Moreover, an even more specific claim than (i-strong) is
(mih i-strong) The FBOE is a result of the maternal immune hypothesis (MIH)
because other things follow from this (e.g., there should be no association between number of older brothers and likelihood of homosexuality in women).

I believe the authors are specifically interested in (mih i-strong). At least to my reading, they make this pretty clear in their Introduction and it is implied by their discussion of the association between orientation and older brothers in females in the third part of their paper. However, sometimes they do not stay carefully attuned to this throughout the paper. I think they need to be more clear about what version of the FBOE they are specifically interested and how various analyses they do follow from or depend on the version of interest. As of now, I think this is too loose and the paper risks overclaiming (i.e., because it is attacking a narrower position than sometimes appears to be the case).

While on the topic of overclaiming, the authors’ three-perspective triangulation approach is nice and appreciated and impressive. However, fundamentally, all three perspectives are mathematical / statistical in nature and so are in a sense three variations on a single perspective rather than three separate perspectives. To put the nail in the coffin of the FBOE (or rather some particular version of the FBOE like mih i-strong) as I believe the authors seek to do requires a more holistic assessment that goes beyond the mere mathematical / statistical and engaging with and triangulating across various lines of evidence (e.g., evidence of antibodies as discussed by the authors with respect to the MIH).

Consequently, I think the title of the paper is a strong overclaim. The manuscript should indeed cause those who view the FBOE as “settled science” to question that view and remain more tentative (although maybe they should not have had such certainty in the first place). However, the manuscript does not seem to have unequivocally refuted the FBOE.

Because the title and some of the writing tends to read like the authors believe they have, I would ask them to be carefully attuned to avoiding that in any revision. In doing so, it is important to emphasize that even a proof that (mih i-strong) is false (which is lacking) need not imply (i-strong) or (i-weak) is false. In fact, I think there may be nothing in the paper speaking to either (i-weak) or (ii), which is of course okay! It is just important for the authors to make clear to readers that there are various version of the FBOE and they might each have differential support in light of the arguments put forth in this paper.

I now discuss each of the three perspectives discussed in the paper.

A. Mathematical assessment of the relevant dependent variable

Before proceeding, I have an item of confusion. On Lines 167-169, the authors write “P(OB) should be read as ‘the probability that a given sibling is an older brother to the participant who reported him or her.’” This seems underspecified. I think we would need subscript i (denoting the participant who reported) and a subscript j (denoting the given sibling). Even if we take the focal participant as given (so that the subscript i is understood), I do not see how we avoid the subscript j. Without it, P(OB) instead would seem to denote the incidence of having an older brother at all rather than that some specific sibling is an older brother.

This is important because many of the key quantities discussed in the paper such as the older brothers odds ratio (OBOR; Equation 1) follow from P(OB) or some conditional version of it and therefore cannot be interpreted without this issue clarified. Can the authors clarify?

Moving on, I agree with the decomposition of the older brothers odds ratio (OBOR) in Equation 4 and the comments immediately below that in Lines 217-222 about the logical implications that follow from it. However, what I am less clear about is the comments in Lines 222-223 that the FBOE requires “the homosexual group must show a greater proportion of older brothers among older siblings.” Does it? The quotation of Blanchard (2018a) on Lines 45-47

“Older brothers increase the odds of homosexuality of later-born males, whereas older sisters, younger brothers and youngers sisters have no effect on those odds.”

seems very clearly to be a statement about numbers and not one about proportions. Therefore, it does not seem the FBOE requires this as claimed by the authors.

Regardless of whether or not the FBOE requires this, the authors’ solution is to eschew the OBOR (as well as the OBOR and older sisters odds ratio (OSOR) in tandem) and instead to consider the odds ratio (OR; Equation 6), namely the ratio of older brothers to older sisters among homosexuals divided by that same ratio among heterosexuals.

However, it is not clear this follows from (i-strong) or (mih i-strong). I would think either the incidence of having an older brother among homosexuals to heterosexuals OR the ratio of the number of older brothers among homosexuals to heterosexuals would be adhere more closely to (i-strong) and (mih i-strong).

That is, these claims seem to be more about the incidence and numbers of older brothers and not about them as a fraction of older siblings; again, see the quotation of Blanchard (2018a) on Lines 45-47 (as an aside, mih i-strong arguably suggests the greater the number of older brothers, the higher the likelihood of homosexuality not just that incidence of older brothers increase the likelihood).

Thus, while I agree with the authors’ criticism of the OBOR, I do not think their proposed OR solution follows. It seems to involve a change in the very nature of the FBOE from being a statement about incidence or numbers to one about proportions. These are very different things.

Now, I understand the whole use of OBOR, OR, etc. in the first place are motivated by (a) the idea that homosexuals and heterosexuals might have different family sizes (and indeed seem to empirically, perhaps as an artefact of sampling/measurement) along with the idea that (b) difference in family size arguably should be adjusted for in some manner. Maybe differences in family size do need adjusting. However, it seems to me that either or both of the following are or could be the case
(i) the best way to adjust would be to obtain better samples/measures (perhaps at the level of the entire family) rather than to rely on statistical or mathematical adjustments like both Blanchard and the present authors do
(ii) perhaps differences in family sizes are not a mere artefact of sampling/measurement but require an extension of the underlying theoretical model.

More minor comments:

Lines 342-344: I think this overstates matters. The statistical model does not say they have no effect but instead says they have an effect but in this very tightly specified and inflexible manner. Please be precise.

Lines 378-379: It is wrong to say “Gelman and Stern (2006) proposed the difference between #OB and #OS as one predictor and #Older as a second predictor.” Gelman and Stern never discuss older brothers, older sisters, etc. in their paper. I know what you are trying to say here but you need to say it in a manner that is precise and correct rather than attribute something specific to Gelman and Stern that they did not actually say.

I note as an aside that a perhaps useful additional approach would also be to consider a regression on OB and OS with coefficients forced to be equal versus allowed to vary. One could perhaps also look at OB alone.

B. Simulation Study

I found this section of the paper poorly motivated and therefore not all that compelling. It also seemed like a bit much and rather unnecessary after the prior section. Further, whereas the link between the first and third parts of the paper was quite strong (i.e., an argument for OR over OBOR suggests revisiting prior meta-analyses but using OR in place of OBOR), this section just seemed to dangle in the middle. I think it could profitably be cut, especially given the length of the paper.

That said, I do believe the simulation description was clear (although choices of parameters, distributions, sample sizes, etc. were not always well justified). Also clear was the description of the models used.

Some final questions and comments: why do you employ a multivariate Poisson in this simulation? Do family sizes tend to be distributed in this manner? Often real data is overdispersed and so an overdispersed-Poisson or the Negative Binomial distribution might be a better choice. And is it reasonable to assume the same correlation structure for families of homosexuals and heterosexuals?

Further, the evaluation of the simulation focused on two-sided significance tests. I would much rather see an evaluation based on how well these approaches are able to estimate theta (bias, RMSE, CI coverage, CI width, etc.), which you say on Line 538 is what they are designed to do. Nonetheless, even remaining within the significance testing framework, the two-sided test approach seems very odd given that the FBOE predicts a direction. It would therefore make more sense to use one-sided tests.

More minor comments:

Line 425: theta is incorrectly written as cursive theta here.

Line 440-443: Can you explain why you let pi vary for the homosexual group but not the heterosexual one? And why you increased it above .5 but did not decrease it below?

Line 493: Same comment about Gelman and Stern as above.

Table 4, Model 9: The effect of interest is incorrectly written and copied from that of Model 8.

Figure 1: Why does Model 9 not appear in this figure?

C. Multiverse Meta-analysis

This section consists of two parts. The first part is a re-analysis of Blanchard (2018a, 2018b) and Blanchard et al. (2021), as well as the meta-analyses for the two sets consisting of all samples of men and women using the authors’ preferred OR metric. It also presents the OBOR metric. The second part is a multiverse meta-analysis.

The first part seemed competently executed with one exception: it seems entirely unreasonable to use a fixed effects meta-analysis model for this data. The fixed effects model is almost never appropriate in biological and social science applications like this one, and the authors themselves note the variability within and across samples. Of course some point estimates and several lower interval estimates of heterogeneity for the random effects model are zero, but that is simply because the random effects model frequently has such zero estimates even when implausible (Chung, Rabe-Hesketh, and Choi, 2013; Chung, Rabe-Hesketh, Dorie, Gelman, and Liu, 2013). Please stick with the random effects model only.

Further, I did not find the raindrop plots or albatross plots to be particularly useful. It would instead be useful to see more traditional forest plots with studies sorted in some reasonable order (e.g., effect size).

As for the second part (the multiverse meta-analysis), my concern is whether the ten “Which” factors specified on Line 822, etc. constitute true “researcher degrees of freedom” or whether the choices made by the researchers were indeed “good” or “right.” For example, around Lines 700, the authors discuss the issue of whether gender dysphoric males should be classified as homosexual and allege this is “yet another unchecked researcher degree of freedom.” Surely, however, it is a subject matter decision whether this is or is not a researcher degree of freedom: if most or all subject matter experts would generally agree that the vast majority or all of the gender dysphoric males are indeed homosexual in orientation, then it would seem not to be a researcher degree of freedom to code them as such. Indeed, on the contrary it would seem necessary to do so. Similarly with the choice of excluding bisexual and asexual groups as discussed in Line 693. Because I cannot well assess the choices made for these ten “Which” factors, I cannot place too much stock in the results of the multiverse meta-analysis.

Finally, I would again omit fixed effects models in this multiverse meta-analysis as they are not justified.

More minor comments:

Line 693-694: Why is this so obvious? Also, I think the tone here and on this whole page is bordering on unprofessional as it makes allegations which seem overly strong or at least not well founded.

Line 775: One can interpret the random effects model as having an assumed distribution as an inferential goal but one need not when one uses it. One can instead use it as a device for principled shrinkage estimation without believing anything about the population of studies. Therefore, the statement made here is too strong. I think it would be best to simply remove it.

Final comment:

The authors touch on an issue throughout that was not clear to me. In particular,
Lines 293-297
Lines 721-723
Lines 1265-1275
all pertain to the unit of analysis. I have a question: in the original samples, are there cases where more than one sibling from a given family appears? If so, then treating siblings as independent seems problematic for obvious reasons. If however only one sibling per family appears, maybe this assumption does not really matter in practice. Can the authors discuss this issue at greater length? It seems like a pretty crucial one that merits more elaboration.

·

Basic reporting

The authors follow aspirational current reporting standards. Statistics in the manuscript are reported clearly, and the online supplement on OSF provides code, codebook, and data, where possible.
The other criteria listed on PeerJ, which I will not repeat here, are met as well.

Experimental design

The approach is non-experimental, but I will use this space instead to comment that the authors identify a serious causal identification problem in the existing literature and solve it adequately. The research question (theoretical estimand) is clearly defined, there is a substantial knowledge gap (because past empirical estimands did not correctly capture the quantity of interest) and the study finally brings both in line. It’s rigorously conducted and as far as I can tell does not lag the state of the art in meta-analysis.

Validity of the findings

Where possible, the underlying data has been provided (Tran et al.). The authors revisit a messy tranche of the literature and extract the signal from the noise. They are careful about limiting their conclusions to what the data can show and provide a wonderful counterbalance to past overclaims.

Additional comments

This is extremely important work and it has evidently been a lot of work. The fraternal birth order effect has been canonized as a stylized fact, but the literature is apparently dominated by a confused approach to causal inference, wishful thinking, publication and selection bias, and a lack of openness and transparency. The evidence is much less clear than I would have thought, even though I did have my doubts. The authors bring a lot of light into the darkness here and have made a heroic effort to organize a seemingly very disordered literature. However, they struggle to summarize all their work effectively. My comments are intended to help that the implications are not lost on anyone and to make this work shine.
1. Starting with the Abstract, I would ask the authors to make the paper punchier by using clear causal language and summarizing the main conclusions (the General Discussion does this better). So, the OBOR is inappropriate to isolate the causal effect of older brothers on male homosexuality from the effect of older sisters/older siblings in general. Is the empirical data consistent with an effect of older siblings instead? Is the effect not specific to male homosexuality? Please make clear statements about this in the abstract.
The subphrase “almost no variation in the number of older brothers in men is attributable to sexual orientation” is very confusing. Obviously, being gay cannot cause anyone to have more older brothers, but the word “attributable” seems to prompt that interpretation. Shouldn’t you say “the number of older brothers does not raise the odds of being homosexual in men”? And if it is still correlated with these odds (because it indirectly reflects confounding factors), you should say so too.
Edit: After having finished this paper for a second time, I return to the abstract. I think my high-level summary would be: “when analyzed correctly, the fraternal birth order effects is small, not specific to male homosexuallity, and the existing evidence appears to be exaggerated by selection and publication bias.”
2. Throughout, I don’t particularly like the phrase “an excess of older brothers”. “excess” has a negative connotation, but more importantly the comparator is left unclear (compared to homosexual women/heterosexual men/women and heterosexual men?). I think almost everywhere this is used, you could more briefly say “have more older brothers than heterosexual men”. I know Blanchard started it, but perhaps this is another bad practice of his, which verbally encodes the lack of clarity that his estimator shares. So, you need not repeat it.
3. L50-55: Clear causal language would help make this section less confusing, but the substantive point is also wrong. If e.g. paternal age decreased the odds of homosexuality and fraternal birth order increased it, we might observe no correlational excess of older brothers among homosexual men, i.e. it’d be masked. I suggest you write something like “a causal interpretation necessitates that relevant confounding third variables have been controlled”.
4. L56-60: I’d suggest to also give an expected rate (percentage) of homosexual children for no older brothers/1 older brother, as I find most people (including me) cannot naturally interpret percentage changes in odds.
5. L70-76: This is extremely confusing reasoning, but that is mainly on Blanchard et al. You are trying to say: this wasn’t originally predicted, so the theory was retrofitted to explain it. You could and should be this frank, as it’s a ridiculous auxiliary.
6. L100-101 If these meta-analyses don’t report an effect size, I think they are reviews at best.
7. L109-110: A specific instance where any awake reader will ask themselves “an excess compared to whom?”
8. L113 Starting with “Suppose”. It’d pack more punch if you wrote simply: In the following, we show converging evidence that incorrect statistical reasoning has exaggerated the FBOE, and that, when corrected, the evidence is less certain and less specific to homosexual males than has been claimed. Together, these strands of evidence call the FBOE and the MIH into doubt.
9. L144. I wonder whether what follows could be more easily digested, if you started with the proper, intuitive way to analyze (fraternal) birth order effects e.g. Rohrer et al. 2015. Then all the deviations from the standard way (individual as unit of analysis, regression framework) are more apparent.
10. L459 Tran et al. Tran et al. (2019) repeated
11. “We refer to this frequency as the false- positive rate. This definition of a false positive rate is conceptually different from what is understood as the false positive rate“
Why not use the established term “bias“ then?
12. This probably varies from reader to reader, but I would have found only section 2, the forward simulation of data plus application of different “indices” easier to understand. Of course, it’s nice to prove the result using probability calculus (section 1), but know your audience. They were misled by the OBOR, so they might not be able to follow all this, and it could still feature in the supplement. As it stands, these sections accomplish the same goal, and though simulation and calculus aren’t redundant, you might lose some readers due to the sheer length of the combined sections. Which would be a shame, as the proper evidence synthesis in the MA is also really important.
13. The meta-analysis seems to be well-done, although I’m not an expert on MA. I enjoyed reading the somewhat narrative style, in which critiques of previous attempts at evidence synthesis are melded with the explanation why this MA is conducted in a multiverse-style. That Blanchard and colleagues excluded a dataset with homosexual marriages but included phallometric testing of two-year-olds is just crazy. I guess I don’t have to wonder which method yielded the desired conclusion.
I would appreciate seeing a standard funnel plot, as the Albatross plot is new to me and I don’t know how to read it to detect publication bias.
15. Re the citation of Silberzahn et al. 2018, these authors (https://journals.sagepub.com/doi/abs/10.1177/23780231211024421) argue that the main problem was that the theoretical estimand was left unclear. This, at least, does not seem to be the case here (if only because the authors applied forensic analysis to Blanchard’s work).
16. L1113: The results of Part III do not lend themselves to the confident
17. L1124-1180: I am sorry to say this, but this entire section strikes me as entirely fallacious. The following paragraph makes sense until the “Nevertheless”
“It is obvious that such factors have a different causal status in their relationship to the human sex ratio than does homosexual orientation of younger brothers, as the latter certainly cannot cause the sex ratio of older siblings to deviate from the population average. Nevertheless, one may use the extent of such observed variations in the sex ratio, as documented in various research sources (Gelman & Carlin, 2014; Gelman & Weakliem, 2009), and then determine where the variation in the sex ratio from the current results would rank among these and make an informed judgment about how plausible such results are. “
The reason Gelman and colleagues sometimes poke fun at Kanazawa for reporting that nurses have many more daughters etc. is that the human sex ratio is very stable (for evolutionary reasons, though see Zietsch et al., 2020) and even extreme scenarios cause only little deviation. However, the fact that causal effects _on_ HSR are known to be small does not mean that causal effects _of_ HSR have to be small.
If you want to make an evolutionary argument about why there shouldn’t be large effects that cause homosexuality, you have to open a-whole-nother can of worms and cite an entirely different literature. If you go back to my point 1 about “variation in older brothers attributable to sexual orientation” this going back and forth when the causal direction is clearly only one way is very confused and confusing. Please strike the entire section and rest assured that your argument that the evidence is fickle is still very strong.
18. Given that you have done all this work to prepare the data for meta-analysis, I would appreciate seeing a meta-analysis of the effect of older siblings and older sisters in addition to the ratio. We’ve established that the FBOE/MIH approach has misled many for years, so let’s take a step in the right direction.
19. In the general discussion, I would appreciate seeing a recommendation that less explicitly instructs researchers on how to further investigate the FBOE. Instead, we should investigate the causal antecedents of homosexuality in an open-minded and rigorous way. One aspect of this would be to consider additional confounding factors such as parental age and parental loss. I would like this to be highlighted, as these confounds clearly should be controlled and acknowledging them should drive the final nail in the coffin of the idea that we can figure this out just with summary statistics at the sibship level. Frisch and Hviid (2006) seems to go in the right direction (large population sample, clear definition of homosexuality), but is not clear about the theoretical estimand and although many variables are adjusted for, some of them may be considered mediators (e.g., length of parental marriage is adjusted for even though it is influenced by parental loss and age).

---

## Round 0.2 · Major Revisions

I just received three reviews, two from the original reviewers and another from a new review. The three reviews are pretty mixed, so I ask for another major revision.

First, I concur with the Reviewers that the Coda on the Nazi scientific roots of the FBOE should be better removed. Although potentially interesting, it seems a bit out of place and may be perceived as offensive.

In particular, I agree with Reviewers #1 and #2 that authors should discuss the two papers mentioned by the reviewers, as they seem to provide novel and additional evidence in favor of the FBOE effect.

I concur with Reviewer #1 that a fixed-effect model is inappropriate in this context.

I also wonder whether the simulation part is essential (see Reviewer #1 and Reviewer #2) as it also makes the manuscript lengthy. I leave the decision to the authors. It is up to them to decide whether that simulation is crucial for their argument. However, I would still recommend addressing the points raised by Reviewer #1 and Reviewer #3 and figuring out whether to make this section shorter and easier to read (perhaps some details could be moved to Supplemental Materials?).

On top of that, I recommend the authors go through each point raised by the Reviewers carefully and either change the manuscript or respond to their points.

Reviewer 2 ·

Basic reporting

See below.

Experimental design

See below.

Validity of the findings

See below.

Additional comments

This paper assesses the empirical foundations of the Fraternal Birth Order Effect (FBOE) from three separate perspectives:
(a) a mathematical assessment of the relevant dependent variable
(b) a simulation study
(c) a multiverse meta-analysis

This revised manuscript, which in most substantive respects remains highly similar to the original, continues to have many of the same problems as the original. The authors were not particularly responsive to my comments. Further, they miss out on key literature and the tone is frequently not scholarly and is borderline unprofessional in some places. Indeed, the revision has introduced text which borders on the libelous.

1. Literature

The revised manuscript fails to address the literature comprehensively. Specifically, I am concerned about the lack of attention given to the following two papers:

[1] Ablaza, C., Kabatek, J., & Perales, F. (2022). Are sibship characteristics predictive of same sex marriage? An examination of fraternal birth order and female fecundity effects in population-level administrative data from the Netherlands. Journal of Sex Research, 59(6), 671-683. https://doi.org/10.1080/00224499.2021.1974330

[2] Blanchard, R., Skorska, M.N. New Data on Birth Order in Homosexual Men and Women and a Reply to Vilsmeier et al. (2021a, 2021b). Arch Sex Behav (2022). https://doi.org/10.1007/s10508-022-02362-z

Paper [1] is a study of over 9 million using Dutch population data and shows an FBOE. The revision briefly refers to [1] in footnote 13 of the Discussion. The footnote claims [1] “only appeared after completion of our work” which is incorrect given that [1] is published and this revision is not. It is therefore entirely inaccurate to say [1] appeared after the *completion* of the work given the work is still in the process of revision and is not complete.

Paper [2] is a forthcoming (accepted but not yet published) response to a preprint version of this manuscript. The revision does not refer to [2] at all.

It would seem to me that any revision of the present manuscript must grapple with both of these papers, not briefly reference one in a footnote at the end of the paper and ignore the other entirely.

There is one further issue related to a comprehensive review of the literature. Specifically, I think it would be extremely helpful to be crystal clear about what Blanchard and colleagues have found from a statistical point of view. It would be nice to say Blanchard has statistically demonstrated X. Now, he might be using the wrong dependent variable (in which case it is fair to point this out). Further, X might not support either the FBOE or the maternal immune hypothesis (e.g., because it uses the wrong dependent variable). There might be various other comments and qualifications. However, it would be nice to identify what precisely has been found. As a reader who is by no means well-versed in this literature, this would help me better understand where this paper is coming from.

2. Confusion Regarding FBOE / Part I

The authors correctly define the FBOE as

“The fraternal-birth order effect (FBOE) is a research claim which states that each older brother increases the odds of homosexual orientation in men via an immunoreactivity process known as the maternal immune hypothesis.”

There are a couple of key points here.

First, the FBOE is a causal hypothesis. Note especially the word “increase” which is necessarily causal language. It does not say “is associated with” or some other such observational (non-causal) language but says “increases.” Nonetheless, the authors claim in their response letter and in their manuscript (e.g., Lines 76-77 and Lines 501-504) that they are interested in rebutting an observational (rather than causal) claim. It is of course their right to consider whatever claim they want but then they cannot say they have rebutted the FBOE which is a causal claim.

Second, and related to causality, the FBOE goes further than simply saying “each older brother increases the odds of homosexual orientation in men.” It provides a causal mechanism, namely that this increase happens due to “an immunoreactivity process known as the maternal immune hypothesis.” Therefore, the FBOE could be rebutted even if “each older brother increases the odds of homosexual orientation in men” (i.e., if that happened via a different mechanism than the maternal immune hypothesis).

Third, the FBOE is a statement about the number of older brothers—not the proportion of older brothers among all siblings or among older siblings or any other such proportion. Nonetheless, the authors continue to examine proportions (or equivalently, odds) of older brothers. While this confusion seems shared with Blanchard and other researchers on the FBOE and despite the authors’ protestations in the response letter and elsewhere that “proportions and counts contain exactly the same information” (they of course don’t when the denominators differ as they do here) and that “[a]nalyzing count data in terms of porportions [sic] is quite adequate,” one cannot rebut the FBOE without a focus on numbers rather than proportions.

The authors are therefore wrong to focus on the “observation of homosexual men having more older brothers relative to older sisters than heterosexual men (i.e., the main theoretical estimand)” (Lines 151-152) because the FBOE says something about the number of older brothers not about number of older brothers relative to number of older sisters. Therefore a relative quantity is not and should not be “the main theoretical estimand” even if such has been used by Blanchard and others.

This leads to problems throughout the manuscript. Let me focus on the section around Lines 297-314. This is the section where Reviewer 1 noted “psychologists tend to struggle with…really any sort of math” and I think this applies to the discussion in these lines as I detail below.

In this section, the authors seem to presume the FBOE is false because they set
P(OB | Older, Het) = P(OB | Older, Hom) = 0.515.
However, the FBOE does not make a claim about P(OB | Older, Het) and P(OB | Older, Hom), and in particular it does not make the claim that they differ as assumed by the authors. Instead, the FBOE states “each older brother increases the odds of homosexual orientation in men via an immunoreactivity process known as the maternal immune hypothesis” In other words, it makes a claim about the number of older brothers and the authors presumption holds only if homosexuals and heterosexuals have the same number of older siblings.

In fact, in this very example, because the authors assume “that homosexual men have more older brothers and sisters compared to heterosexual men; that is, there is no specific difference in the number older brothers that exceeds the difference in the number of older siblings” by setting P(Older | Hom) = 0.58 and P(Older | Het) = 0.48, then homosexual will in fact have more older brothers than will heterosexual men—consistent with the FBOE!
[We leave aside the observation from Lines 562-563 that this assumption is empirically contrary to fact because homosexuals actually have fewer siblings compared to heterosexuals (2.19 versus 3.31)].

Thus, in a short section designed to illustrate a refutation of the FBOE by example, the authors end up providing an example consistent with the FBOE.

3. Part II: Simulation Analysis

This part of the paper seems somewhat orthogonal to the rest and would better serve as a standalone paper. That said, it only begins to scratch the surface.

The presentation in Figure 2 however leaves a lot to be desired. It would be useful to have a separate facet by theta with the dashed line indicating the truth (which admittedly is not relevant to all models). It would also be nice to plot the rate (power for theta = -0.33 and 0.33 and false positive for theta = 0) on a separate plot rather than as a color.

Further, the simulation itself is very limited. The authors note in their response note that choices at play in the simulation (i.e., data generating process and parameter values) “were made from convenience.” This is not very compelling. It is unclear what a simulation based on convenience can tell us about real world data. I would urge the authors to put forth something realistic (i.e., based on actual data and data generating processes) so that the simulation is not constrained to being hypothetical and thus questionably useful exercise.

As a related matter and further the cause of greater realism, the simulation is very oddly designed. It begins with a person. It then probabilistically generates siblings (older and younger brothers and sisters) for that person. It finally probabilistically determines whether that person is homosexual or heterosexual.

It would be more natural to start with a family formation data generating process. Parents decide probabilistically whether to have a child. The child is probabilistically either a boy or a girl and either homosexual or heterosexual. The parents then probabilistically decide to have another child and so on and so on.

Such a family formation data generating process that respects the 51.5% percent probability of a boy, the 2% odds of homosexuality, the 2.19 average siblings of homosexuals, and 3.31 average siblings of heterosexuals would be more realistic and may help assess other claims made in the paper.

4. Part III: Multiverse analysis

Regarding the ten researcher degree of freedom factors (what you label the “Which” factors), you agree that what specific factors and what levels of those factors should be analyzed in a multiverse analysis ought to be based on subject matter considerations (see, e.g., Line 925).

You also note in your response note that you are not subject matter experts.

Therefore, you need to provide evidence that these ten factors and the levels you have picked for them are ones that subject matter experts (e.g., Blanchard and other FBOE researchers) would use. This could come from personal communication. Alternatively, you could justify it by citing papers where such researchers have made these choices (this is what Steegen et al did in that original multiverse paper). I realize you have done this for some of the factors but you have not for all of them.

As an example, you write on Lines 925-926 that “other researchers could disagree on substantive grounds and argue that same-sex marriage constitutes a less error-prone indicator of homosexual orientation than changes in penile blood volume upon exposure to certain stimuli (as is the case in phallometric testing).” Have they? You and I certainly are not subject matter experts on this. What do the subject matter experts have to say?

A nice reference discussing these issues (i.e., how multiverse-style analyses can produce misleading results without such justification) that you might find helpful is:

Del Giudice and Gangested (2012). “A Traveler’s Guide to the Multiverse: Promises, Pitfalls, and a Framework for the Evaluation of Analytic Decisions.” Advances in Methods and Practices in Psychological Science. Vol. 4, No. 1, 1–15.
https://journals.sagepub.com/doi/pdf/10.1177/2515245920954925

In addition, your single “How” factor is random versus fixed effects meta-analysis. You repeat the old canard that the random effects model is “estimating the mean and variance of an assumed distribution of population effects” (Line 938) This is surely one interpretation of the random effects model. However, it is not the only one. Again, the random effects model can be used for principled shrinkage estimation without believing anything about any population of studies.

Further, given the enormous estimates of heterogeneity, the use of a fixed effects model simply is not appropriate here. Yes, I am well-acquainted with the arguments of Hedges & Vevea, 1998 and Rice et al., 2018 but they do not seem to apply here.

The best argument for including the fixed effects model here is that it is a potential researcher degree of freedom. But, as noted above, what specific factors and what levels of those factors should be analyzed in a multiverse analysis ought to be based on subject matter considerations. Any subject matter expert on meta-analysis would agree the fixed effects model is not appropriate for this kind of data. Therefore, it does not seem reasonable to use for the reasons given above.

Therefore, I would remove the fixed effects model. If you insist on keeping it, please provide separate versions of Figures 5 and 6 (one for fixed and one for random).

5. Coda

The Coda is simply unnecessary as it is orthogonal to the points made in the manuscript. Further, it reads like an oblique attempt to libel Blanchard and other FBOE researchers as Nazis. If you want to accuse them of being such, do it directly and provide evidence for that assertion. Or, simply cut this Coda.

6. More Focused Comments

a. Lines 76-77: The authors write “The reason why the FBOE is interpreted causally seems to go back to the types of statistical analyses used in the context of FBOE research.” This is false: the FBOE is fundamentally a causal claim and one with a specified causal mechanism.

b. Lines 156-159: Despite the bombastic and inappropriate title claim that the FBOE is a “statistical artefact,” the authors seem to admit evidence consistent with the FBOE (just smaller than previously documented) when they write “the specific association between the number of older brothers and homosexual orientation is (a) much smaller than previously claimed, (b) highly variable across different samples, (c) not specific to men, and (d) possibly exaggerated due to the influence of publication bias.” Therefore, they should retitle their paper to something more accurate.

c. Line 225: Is not P(OB, Older) simply P(OB)?

d. Footnote 6: In English, we say “only children” not “single children.”

e. Lines 501-504. Same comment as a.

f. Lines 622-623: Something is awry. You have 18 combinations and 1,000 replications per combination and so should have 18,000 not 180,000 replications in total (or 10,000 replications per combination).

g. Line 1070: Why do you use rma.mv() and its accompanying profile likelihood intervals for tau? I understand you would need to use this for the three-level models you fit. However, I believe you also fit a variety of two-level models. These could be fit via rma.uni() which provides Q-profile based intervals which perform better than profile likelihood intervals.

h. Line 1123 / Table 8: I am confused why you use OR for Homo and OBOR for Hetero in the table. Also, I think it would be better separate out OR from OBOR rather than having them mingled since they are estimating different quantities. Finally, why are some of the PIs missing?

i. Line 1127-1131: I think this discussion of PIs and \hat \tau should be moved above and be together with the discussion of CIs.

j. Line 1143 / Figure 3: Sorting by effect size rather than weight would be better.

k. Line 1396: You write “The lack of available primary data highlights another peculiarity of FBOE research.” There is nothing peculiar to FBOE research about this. Lack of data availability is widespread in psychological research as a whole. This false statement should be removed.

·

Basic reporting

no comment

Experimental design

no comment

Validity of the findings

no comment

Additional comments

I read the response to reviewers and re-read the manuscript. I was satisfied with the authors responses to my queries and Dr. Rohrer’s queries. I was surprised by the novel section on the historical roots of the BO research. I would make this a separate paper, as it has no strong connection with the rest of the manuscript, which is already very long. It’s certainly interesting, but who knows, maybe the controversial/“political” aspect will overshadow the methodological aspect and provide something for detractors to focus on, even though the case against OBOR is clear no matter where you stand on citing Nazi science. My recommendation is that the authors separate this out instead of “tacking it on”, but it’s their manuscript. It’s also worth noting that Ray Blanchard has written a rambling reply to the preprint of the manuscript under review here. https://link.springer.com/article/10.1007/s10508-022-02362-z I think he confused several key issues, perhaps in part because the manuscript was very long and a bit hard to read, perhaps in part because he is very invested in his theory. I don’t think the authors need to reply to Blanchard 2022 here, as it needlessly makes the debate even more confusing to explain the confusions related to a previous version before rebutting his other points. I am quite hopeful that the revised version is a slightly easier read.
I hope that we will soon see better research on the important topic of the causes of homosexuality thanks to this manuscript. I made some minor responses within the text.

Reviewer 4 ·

Basic reporting

The reviewed manuscript critically evaluates the current evidence on the fraternal birth-order effect. I commend the authors with this thorough piece of work. They very extensively discussed some important issues with studies on the fraternal birth-order effect. However, the extensiveness is actually also a drawback of the manuscript. The manuscript is very lengthy. I am wondering whether the same points could not have been communicated in a shorter manuscript. I also believe that the writing could be improved at multiple places. Below I list my comments/suggestions to improve the manuscript.

1. From time to time, I had a hard time understanding what was explained (probably due to me not having any prior knowledge on the fraternal birth-order effect). I experienced especially difficulties with the so-called “Part I”. This might well be caused by the many abbreviations that were used in this section. I had to look up definitions multiple times when reading this section.
This is an overview of the difficulties I had with Part I:
- Lines 283-284: There is stated that “… the homosexual group must show a greater number (or proportion or probability)”. I think it is not about the absolute number in the homosexual group. It is about the average. Anyway, referring to proportion or probability is probably better here.
- Line 287: The link to Simpson’s paradox was not really clear to me. The main issue in Simpson’s paradox is the aggregation of data. I do not see any aggregation here, so I would not link this to Simpson’s paradox.
- Lines 337-356: This paragraph was hard to understand for me. Probably, because of the definitions. I can imagine that this is not the case if you are really familiar with this type of research.
- Line 347: There is stated that weighting was used, but it is unclear how the weights were supposed to look like.
- Lines 390-393: MROB and MPOB are currently not defined in the text. This made it impossible for me to understand this part.

2. The manuscript is currently very lengthy. I suggest to take a careful look whether it can be shortened. More concretely, I am not sure whether a simulation study is really needed in Part II. If I understood it correctly, the authors are arguing that the parameters that are estimated in these different models are actually not estimating the fraternal birth-order effect. That is, the \beta in the models are not estimating the \theta. If this is the case, you do not need simulations to prove this point. A theoretical explanation is needed here, and I would not use any simulations. This theoretical explanation can also be pretty brief, so this will substantially shorten the manuscript.
If the authors want to keep the simulation in the manuscript, Figure 4 can be improved in two ways:
- I would use as figure titles the notation in Table 4 rather than the formula argument.
- Can the axis labels be added? These are currently missing and make it difficult to interpret the figure.

3. The writing of the manuscript can be improved at various places. I was especially surprised by paragraphs that were sometimes very length (e.g., lines 1301-1349) and sometimes only a single sentence (e.g., lines 1192-1194, 1234-1235, 1236-1237). The readability of the manuscript will be improved if more attention is devoted to writing proper paragraphs. I understand that this is not very clear advice, but I suppose that the authors can quite easily improve the manuscript by carefully reading it again and making changes where needed.

Experimental design

4. The factors Lab and Archives are used in the multiverse analysis. I agree with the authors that these are interesting aspects to study. However, are these really factors of the multiverse. Where these factors also used in some of the meta-analyses? If not, I would not include it as factors in the multiverse.

Validity of the findings

No comment

Additional comments

1. Lines 85-88: There is stated: “For all we know, odds rations do not warrant a causal interpretation…”. It would be stronger if literature is cited here that makes this point and remove the “for all we know” phrase.

2. Lines 113: What is meant by “retrofitted”?

3. Lines 222-228: These lines are explaining really basic probability calculus. I would omit this and refer to a textbook on this.

4. Line 409: There is a word missing between “the” and “of”.

5. Line 540: This does not seem correct to me, because the vector is not multiplied by \mu. \lambda appears to be the vector multipled by \mu, but it is currently only the vector.

6. Lines 668-675: Simulations are used here to calibrate the sample size. Is this really needed? Can you not do a power analysis to determine this sample size?

7. Lines 794-837: Contain a lot of information that might not be necessary in the manuscript. Can this not be moved to the supplements? This will also shorten the manuscript a bit, which is a good thing.

8. Lines 1109-1112: I had difficulties understanding these results. I could not find these in Table 8. What kind of analysis was exactly conducted here? Can this be better explained?

9. Line 1129: Not all readers will be familiar with how to interpret the prediction intervals. I would add one sentence explaining the meaning of these intervals.

10. Lines 1268-1288: I am not sure whether this paragraph really adds value. I leave it up to the authors to decide on removing it or not.

11. Lines 1374: Small-study effects are here interpreted as evidence of publication bias. This could be the case, but there are also other causes of small-study effects (e.g., Egger et al., 1997). Hence, it is better to be a bit more careful here with interpreting small-study effects as evidence in favor of publication bias.

Egger et al. (1997). Bias in meta-analysis detected by a simple, graphical test.

12. Lines 1409-1517: First of all, I do understand the importance of conveying the message of this section. However, I am not sure whether it is related enough to be included in the manuscript. It does not question the papers that were more recently conducted on the fraternal birth-order effect. Given the current length of the manuscript, it can probably be omitted.

13. There is sometimes LaTeX code in the descriptions of the tables (e.g., Table 4).

---

## Round 0.3 · Major Revisions

With the Reviewer's responses in hand, I recommend further revising your manuscript, as there are remaining major issues that need to be addressed.

In particular, I think that you should engage with the most recent literature, as pointed out in comments #1 and #2. You can take advantage of this discussion to further clarify why you think that, for instance, the critique raised by Blanchard and Skorska is out of target. But I don't think you can ignore that critique, and I am sympathetic to the Reviewer's argument on this.

I also agree with the reviewer about removing the fixed-effect model in the presence of a great amount of heterogeneity (comment #5).

However, I disagree with Reviewer point #3, as I think that the authors provide a compelling argument, which was acknowledged also by the other reviewers, for conditioning on the number of older siblings. In fact, not doing that would mean chasing a confounded effect. Therefore, I do not think you need to address that.

As for point #4, I personally see some value in retaining Part II, but it would be great if you could tidy up your manuscript a bit further.

Finally, I recommend addressing point #6 either by rebutting the argument or by implementing it in your manuscript.

Reviewer 2 ·

Basic reporting

See below.

Experimental design

See below.

Validity of the findings

See below.

Additional comments

This paper assesses the empirical foundations of the Fraternal Birth Order Effect (FBOE) from three separate perspectives:
(a) a mathematical assessment of the relevant dependent variable
(b) a simulation study
(c) a multiverse meta-analysis
This third round manuscript is in most substantive respects highly similar to the original submission as well as the second submission. Therefore, because the authors have not been responsive to comments made in prior rounds, it continues to have many of the same problems as those identified in the original and second round submissions.

Given this, I will point the authors back to my comments from prior rounds and make some focused comments here:

1. The decision to not discuss Blanchard and Skorska’s published piece is a grave omission of the literature. Your reasoning for it is also quite poor. You justify not to discussing Blanchard and Skorska because Ruben Arslan finds it “rambly.” This is not a justification but even were it, it is important to note that Ruben Arslan also credits the rambliness and confusion in Blanchard and Skorska at least in part to the fact that your own “manuscript was very long and a bit hard to read.”
Further, your comment in the response note that discussing Blanchard and Skorska “has a high potential of confusing readers” is not persuasive. It is for you to write in a manner such that readers are not confused. You can, for example, reference previous versions of your manuscript if Blanchard and Skorska’s criticisms pertain to those but not the current version.

While I agree with you that “a journal response to a preprint appears highly unusual,” your decision to post to a preprint server (quite a good thing generally) created this situation. It therefore cannot be used as a reason for not discussing relevant literature.

2. The decision to brush aside the Ablaza et al piece by discussing it so briefly in the Discussion is also unjustified. This is an important study given its size and it really ought to be incorporated into the analyses.

Perhaps worse is your justification for omitting it. You write in your response note “[t]heir [Ablaza et al’s] work appeared after our preprint and therefore after the completion of our work.”

To be clear, posting a file to a preprint server does not constitute the completion of a work. Your work is obviously still in the process of revision and therefore it is not complete. In fact, you seem to acknowledge this in your reasoning for not responding to Blanchard and Skorska when you write in your response note that Blanchard and Skorska’s “response addressed a preprint study (ours) which since then has changed. Thus, Blanchard and Skorska’s response is outdated.”

While your lack of responsiveness to review team feedback is consistent with your view that your work is “complete,” it appears you want to have both ways, claiming it is complete regarding Ablaza et al but work in progress regarding Blanchard and Skorska.

3. You claim in your response note that “[t]he FBOE implies a greater number of older brothers *conditional on the number of older siblings*” [emphasis yours]. Where does the condition in emphasis come from? Again, the FBOE as you write is:
“a research claim which states that each older brother increases the odds of homosexual orientation in men via an immunoreactivity process known as the maternal immune hypothesis. Importantly, older sisters supposedly either do not affect these odds, or affects them to a lesser extent.”
This makes a causal statement and provides a mechanism but makes no comment about conditioning on number of older siblings. Now, it could be that your statement is an implication of the FBOE but it is not clear to me that it need be.

4. The manuscript remains much too long as everyone on the review team agrees. Part II really should be omitted and Parts I and III tightened up. They are, to use a word, “rambly.”

5. I would omit the fixed effects results. Heterogeneity is massive.

6. Line 839: A univariate three-level model is one way to accommodate male and female samples jointly. However, it (at least as implemented) relies on very strong assumptions about the variance-covariance matrix of the random effects. Your data is better viewed as bivariate (male, female) and two-level in nature. Your univariate three-level model assumes that the variance of the random effects for males and females from the same study are equal and that the covariance is positive. However, you need not make these strong assumptions: you can estimate two variances and an remove the sign constraint from the covariance. This is what a bivariate two-level model would do.

---

## Round 0.4 · Minor Revisions

I read the authors rebuttal letter and the tracked changes myself. I now feel that the authors have successfully addressed most of the major concerns previously raised by the Reviewers.

I believe that the manuscript is now suitable for publication, pending minor changes. In particular, there seem to be some formatting issues (e.g. lines 209-222), and some minor language issues that I recommend the authors go through carefully.

---

## Round 0.5 · accepted · Accept

I read the paper and I noticed that the remaining minor issues have been addressed. I am glad to inform you that I now consider the manuscript suitable for publication.